**Biological science practices**  

**Cite this article:** van der Wal JEM, Thorogood R, Horrocks NPC. 2021 Collaboration enhances career progression in academic science, especially for female researchers. *Proc. R. Soc. B* **288**: 20210219.

behaviour, ecology

bibliometrics, collaboration networks, gender gap, academic survival, sociality

**Authors for correspondence:**
Nicholas P. C. Horrocks
e-mail: nh415@cam.ac.uk
Jessica E. M. van der Wal
e-mail: jessicavanderwal1@gmail.com

†Present address: Cambridge Institute of Therapeutic Immunology and Infectious Disease, University of Cambridge, Cambridge, UK.

# Collaboration enhances career progression in academic science, especially for female researchers

Jessica E. M. van der Wal[1,2,3], Rose Thorogood[1,2,4] and
Nicholas P. C. Horrocks[4,†]

[1]Helsinki Institute of Life Science, and [2]Research Programme in Organismal and Evolutionary Biology, Faculty of Biological and Environmental Sciences, University of Helsinki, Helsinki, Finland
[3]Fitzpatrick Institute of African Ornithology, University of Cape Town, Cape Town, South Africa
[4]Department of Zoology, University of Cambridge, Cambridge, UK

JEMvdW, 0000-0002-6441-3598; RT, 0000-0001-5010-2177; NPCH, 0000-0003-0762-4142

Collaboration and diversity are increasingly promoted in science. Yet how collaborations influence academic career progression, and whether this differs by gender, remains largely unknown. Here, we use co-authorship ego networks to quantify collaboration behaviour and career progression of a cohort of contributors to biennial International Society of Behavioral Ecology meetings (1992, 1994, 1996). Among this cohort, women were slower and less likely to become a principal investigator (PI; approximated by having at least three last-author publications) and published fewer papers over fewer years (i.e. had shorter academic careers) than men. After adjusting for publication number, women also had fewer collaborators (lower adjusted network size) and published fewer times with each co-author (lower adjusted tie strength), albeit more often with the same group of collaborators (higher adjusted clustering coefficient). Authors with stronger networks were more likely to become a PI, and those with less clustered networks did so more quickly. Women, however, showed a stronger positive relationship with adjusted network size (increased career length) and adjusted tie strength (increased likelihood to become a PI). Finally, early-career network characteristics correlated with career length. Our results suggest that large and varied collaboration networks are positively correlated with career progression, especially for women.

## 1. Introduction

Persisting in academic science is a major challenge, with increasing competition for positions across career stages resulting in prolonged job insecurity and substantial declines in career length compared to 50 years ago [1,2]. At the same time, the persistence of gender disparities in science [3] means that female researchers continue to leave academia more often, and earlier, than male researchers [4]. Lack of diversity hampers novel insight and scientific advances [5], but addressing these inequalities remains challenging. Collaboration among diverse groups of academics is now promoted intensively by institutions and funding agencies [6], particularly in the biological sciences [7], and may also provide a form of mentoring and support [8]. However, while emphasis is often placed on collaboration in funding, hiring and promotion decisions (e.g. [9]), whether collaborations enhance career progression, and how any effects might differ according to gender, remains largely unexplored [1,10,11].

Positive associations between collaboration and academic research metrics, such as citation impact and *h*-index, have been demonstrated previously across academic disciplines [12–16]. Having a larger research network is correlated with obtaining more funding and being more highly cited [15,17], while articles

with higher Altmetric scores have more citations [18]. Furthermore, the reputation of co-authors can facilitate publication in career-defining journals (i.e. the chaperone effect [19]). By shaping funding, citation rates and reputation, collaboration behaviour could have a considerable impact on career progression in science [20,21]. Collaboration patterns, however, differ between genders: female researchers generally have smaller [22] (but see [23]) and more clustered networks [24], publish fewer papers as 'high prestige' authors (single-author, or in first or last authorship positions) [25], and publish fewer papers [10,22] overall than their male counterparts. Nevertheless, differences in productivity have rarely been accounted for in analyses of the consequences of collaboration behaviour (e.g. [13,14,23] but see [22,24]).

Here we borrow concepts from studies testing how sociality relates to fitness and survival in non-human animals to investigate how collaboration patterns may influence career progression. For many species, interacting with peers is a crucial component of lifetime fitness [26,27] and lifespan [27]. Group-level cooperation improves acquisition of resources (i.e. research funding), social learning helps individuals acquire new skills (i.e. social knowledge transfer), and attaining social status (i.e. citations) enhances lifespan (i.e. career length) and corresponds with increased opportunities to breed [28]. Analyses of social interactions and fitness typically use a cohort approach (e.g. [29]), where groups of individuals are followed through time, and combine social network analyses with survival analyses to explore how social behaviours shape, for example, lifespan [27]. We used social network analyses to quantify the structural properties of egocentric co-authorship networks constructed from more than 52 000 papers published over almost four decades by over 900 gender-assigned international researchers. Although collaboration is multi-faceted [8], co-authorship of published papers provides a useful proxy and is now used widely to estimate patterns in collaboration (e.g. [22,24]). In contrast with previous studies, however, we made no restrictions in terms of the journal [13,30] or institute [22], and included active researchers as well as those who subsequently left science [10]. Furthermore, our use of a cohort approach—identifying relevant authors from within a particular time period, rather than dredging all publications from a field of science—allowed us to account for changes in publishing habits over time (e.g. from predominantly single-author to multi-authored papers [15]) and to adjust our measures of collaboration behaviour for differences in productivity between genders.

Our cohort consisted of contributors to any of three consecutive biennial conferences of the International Society for Behavioral Ecology (ISBE) held in the 1990s. This cohort was ideal because: (i) the field includes researchers working on a diverse range of topics in ecology and evolutionary biology; (ii) multi-authored papers, necessary for the construction of egocentric networks, are the norm [13]; (iii) numerous studies of gender biases within ecology and evolutionary biology suggest that, while gender equality is still lacking [31–33], gender bias in publishing is a less significant issue [34,35] (although see [32]); and (iv) a previous study using papers published in behavioural ecology journals found that having larger and stronger networks increased the $h$-index, while collaborating with scientists that themselves were more connected decreased citation impact [13]. However, it remains unknown whether gender differences exist in collaboration networks, or if collaboration behaviour has an impact on careers. If

collaboration behaviour (calculated across academic careers) influences career progression and length, we expected that researchers who published with more collaborators (i.e. unique co-authors) multiple times (i.e. published repeatedly with co-authors), and in more unique constellations of collaborators (i.e. were less 'cliquey'), would: (i) be more likely to become principal investigators (PIs); (ii) become PIs more quickly; and (iii) publish for longer in science. A positive correlation between authors' collaboration behaviour and career length might arise because long-established or senior researchers are more attractive to collaborate with (the Matthew effect [36]), or have simply had more time to establish successful and enduring collaborations. Therefore, we also conducted a longitudinal analysis to investigate whether a focal authors' propensity to collaborate in their early career correlated with their eventual academic career length.

## 2. Methods

### (a) Identification of focal authors

Our subjects (1469 unique names, electronic supplementary material, S1) were contributors to any of the ISBE biennial meetings in 1992 (Princeton, USA, 766 contributors), 1994 (Nottingham, UK, 521 contributors) and 1996 (Canberra, Australia, 565 contributors). ISBE conferences attract researchers in the fields of animal behaviour, evolutionary biology, population biology, physiology and molecular biology (www.behavecol. com). We chose these three conferences because: (i) they spanned three continents, maximizing the geographical coverage of our sampling; (ii) ISBE conferences at this time attracted most of the active researchers in behavioural ecology and related fields (Prof. Lotta Sundström and Prof. Nick Davies, February 2020, personal communication); and (iii) three decades should allow sufficient time for participants starting their career in the early 1990s to reach PI status (should they have wanted to and been successful in doing so).

After inferring the gender (wo/man) of conference contributors from their first name and/or online profile, and recording the continent that they were associated with at the time of their first conference registration, we used R v. 3.5.0 [37] for all further data processing, analyses and visualizations. We used the 'rscopus' package [38] to identify contributors who had published at least one peer-reviewed scientific article, as listed in the academic database Scopus (www.scopus.com) as of September 2018. We manually excluded duplicated records and included only contributors who had published at least one paper on a subject relevant to an ISBE conference (see the electronic supplementary material, S2, for more details). Using the 'bibliometrix' package [39] in conjunction with Scopus API keys (www.api.elsevier.com; see the electronic supplementary material, S2), we downloaded the author list, title, publication year and journal of all publications for all remaining contributors in our dataset. Finally, we applied a series of further restrictions to this dataset to account for variation in publishing practices over time and to allow the construction of meaningful social networks based on a homogeneous cohort (see the electronic supplementary material, S3 for details). None of our restrictions disproportionately excluded women or men from our final dataset (electronic supplementary material, S3). Lastly, we used the 'asnipe' [40], 'igraph' [41], 'intergraph' [39] and 'network' [42] packages to transform publication records into a data structure from which network metrics could be calculated. Our final dataset contained 52 698 papers published by 935 focal authors (298 women: 32%, 637 men: 68%). Focal authors came from 39 different countries, with the majority from Europe (47%) or North America (35%).

## (b) Egocentric network metrics and accounting for number of publications

For each focal author, we constructed egocentric social networks [43] based on the author's full list of publications and co-authors, and derived three metrics: (i) network size, which describes the total number of unique co-authors; (ii) mean tie strength ('tie strength', hereafter) which describes how many times, on average, a focal author published with each co-author (i.e. high tie strength indicates that a focal author published repeatedly with each co-author); and (iii) 'global clustering coefficient' (using the transitivity function in the 'igraph' package [41]) which ranges from 0 to 1 and describes the average connection of co-authors to each other, via the focal author [44] (see the electronic supplementary material, S4 and figure S4). High connectedness (i.e. clustering coefficient close to 1) indicates the same co-authors appear together on many of the focal author's publications (but does not indicate how many papers co-authors may have published together *without* the focal author). A clustering coefficient could only be calculated for focal authors with more than one unique co-author, but this excluded only five focal authors (two women and three men) from the dataset.

If the author position on papers in our dataset varied by gender [45], then it could bias interpretations about collaboration behaviour and career progression. Therefore, we checked whether female and male focal authors differed in their proportion of either first or last author papers, after accounting for the first publication year, given that number of authors on publications has increased with time [15]. We found no significant difference (prestige authorship position: 62% of female focal authors' multi-authored papers, 63% of male focal authors' multi-authored papers; generalized linear model (GLM) (using the maximum-likelihood method) with a binomial error distribution and identity link function, $n = 935$ focal authors, gender: estimate ± s.e. = $-0.02 ± 0.02$, $z = -0.91$, $p = 0.36$; first publication year: estimate ± s.e. = $-0.02 ± 0.002$, $z = -8.87$, $p < 0.001$). Therefore, we did not include authorship position in our further analyses.

As expected, productivity differed according to gender (see Results), and all three metrics of collaboration behaviour were significantly correlated with the number of publications (all $R^2 > 0.39$ and $p < 0.001$; electronic supplementary material, figure S5.1). To account for these strong and gender-biased relationships between publication number and collaboration behaviour metrics, we regressed each metric in a non-hierarchical model including a main effect of number of publications and its interaction with the focal authors' gender (see the electronic supplementary material, table S5), and then used the mean-centred, standardized model residuals as measures of collaboration behaviour adjusted for publication number (henceforth referred to as 'adjusted' metrics) in all further analyses. For example, a focal author with a positive 'adjusted network size' score would have more unique co-authors than expected, given their number of publications. Note that this residual regression approach [26,46] is not the same as the much criticized 'regression of residuals' practice [47], since here we regress the predictors, rather than the response variable. For comparison, analyses using unadjusted metrics of collaboration behaviour are included in the electronic supplementary material, appendix 1.

## (c) Measures of academic career progression and accounting for authorship name changes

We measured the career progression of focal authors in terms of their PI status and career length. The last authorship position usually denotes senior leadership of an independent project in our field [48], so we classified only those focal authors with at least three last-author (multi-authored) papers as 'being a PI' [49]. Similarly, we approximated 'time to become a PI' as the difference in years between a focal author's first publication and their third last-author (multi-authored) publication (based on the methods of [49]). This approach is robust to variations such as using the time to publish two or four last-author publications (e.g. SI in [41]), although we acknowledge that it remains a crude estimate of PI status. Career length was measured in years from first until last publication, as in [1]. Bibliometric definitions of academic career progression are necessarily limited and we recognize that there are other equally valid and important means of contributing to science that do not produce publications; however, these are not easily quantified [50].

Our definition of career length relied on correctly identifying publications of focal authors, which can become complicated if an author changed their publishing name during their career (e.g. owing to marriage). Unfortunately, there is little published work on how name changes affect indexing services and citation accuracy (but see [51]), so we checked whether author name changes could have generated false-negative records of leaving science. Using Google Scholar and ResearchGate, we searched for professional and personal webpages containing publication lists for 123 female focal authors in our cohort who stopped publishing. We looked for any change in publishing name and checked the number of publications that a focal author had produced, and the date of their last publication, against our main dataset. Only two female focal authors were incorrectly assigned in our dataset as having left science owing to a change in publishing name, and we updated our records accordingly.

## (d) Statistical analyses
### (i) Data independence

Metrics derived from egocentric networks can suffer from non-independence if individuals occur repeatedly across multiple networks [52]. We determined the proportion of networks in which focal authors also appeared as co-authors (less than 1% of networks on average, with one focal author appearing in a maximum of 5.5% of networks, electronic supplementary material, figure S6A), and the proportion of papers that occurred in multiple networks (less than 1% of all networks, electronic supplementary material, figure S6B). Given that the egocentric networks in our study were almost entirely independent, we followed the approach of [52] and used parametric statistical analyses.

### (ii) Testing for gender differences in productivity, collaboration behaviour and career progression

We used GLMs ('lme4' package [53]) using the maximum-likelihood method to test for gender differences in: publication number (Poisson error distribution and log link function), unadjusted social network measures (network size: Poisson error distribution and log link function; tie strength: Gaussian error distribution and identity link function, log10-transformed; global clustering coefficient: binomial error distribution and logit link function), and adjusted measures of collaboration behaviour (adjusted network size: Gaussian error distribution; adjusted tie strength: Gaussian error distribution; adjusted clustering coefficient: Gaussian error distribution). In all GLMs, we set alpha to 0.05 and 'man' was the reference gender. We assessed the assumptions of normality and homoscedasticity for GLMs by visual inspection of residuals and normal probability plots (using the 'DHARMa' package [54]), and derived model outputs using the 'jtools' package [55].

We then used parametric accelerated failure time (AFT) survival models using the functions survreg (in package 'survival' [56]) and flexsurvreg (in 'flexsurv' [57]), to model whether gender explained variation in (i) time to become a PI and (ii) career length. Unlike Cox proportional hazards models that

estimate a hazard function, AFT models provide a 'deceleration factor' which is an intuitive summary measure that describes the extent to which the survival curve is shifted forwards (deceleration factor greater than 1.0) or backwards (deceleration factor less than 1.0) by the variable of interest [58] (see the electronic supplementary material, S7). Estimates of the deceleration factor are significantly different (alpha = 0.05) when the 95% confidence interval (CI) does not include the value 1.0. We ran all AFT models (with 'man' as reference gender) using a lognormal distribution, following a visual inspection of residuals to determine model fit and comparison of log-likelihood and Akaike information criterion (AIC) values for models with different distributions.

In (i) (analysis of time to become a PI), we excluded 163 subjects who left science—i.e. had stopped publishing two or more years before the end of the study period—without ever becoming a PI, and thus had no future prospect of achieving this status (leaving 772 focal authors; of which 219 were women and 553 men). In (i), the event was being a PI, and 64 of 772 focal authors (8%; 50% of whom were women) were censored because they were still in science at the end of the study period yet did not become a PI during this time. One focal individual that achieved PI status in the same year as their first publication was assigned a dummy value of 0.5, as survival models ignore zero values. In (ii) (analysis of career length), the event was cessation of publishing, and 642 of 935 focal authors (69%; 27% of whom were women) were censored because they were still publishing within 2 years of the end of the study period. A 2-year cut-off was chosen because the majority (80%) of publication gaps (i.e. years when no papers were published) were shorter than 2 years (electronic supplementary material, figure S8). For the subset of focal authors that did become a PI ($n = 708$), we ran a further AFT analysis to see how achieving PI status affected future career length (measured here in years from the time a focal author became a PI until the time they left science) and whether this differed between genders. We assigned a dummy value of 0.5 to eighteen focal authors (eight women and 10 men) that left science in the same year that they became a PI. Excluding these 18 authors from the analysis did not alter conclusions.

### (iii) Relationships between collaboration behaviour and likelihood of being a principal investigator, time to become a principal investigator and career length

Given that all three metrics of collaboration behaviour were significantly correlated with one another (electronic supplementary material, figure S5.1), we used separate GLMs with a binomial error distribution [53] to determine whether any of our three metrics of collaboration behaviour correlated with the likelihood of focal authors being a PI ($n = 935$ focal authors), and whether this was dependent on gender. We included gender and its interaction with the metric of interest in these GLMs, and when the interaction was significant, we ran models for each gender separately. When not significant, we re-ran GLMs without the interaction term to derive estimates for the main effects.

Using AFT models as described above, we then tested whether these metrics correlated with (i) time to become a PI and (ii) career length. We included gender and its interaction with the metric of interest in all AFT models. If the interaction was significant, we re-ran AFT models for each gender separately. When not significant, we re-ran AFT models without the interaction term to derive estimates for the main effects.

We investigated the effects of our censoring decisions for all survival analyses. Excluding 93 focal authors (27 women, 66 men) who were classified as PIs according to our definition, but who had a gap between their first and third last-author papers of longer than 10 years (mean + 1 s.d. for all PIs = 5.6 +

4.4 years), did not qualitatively change results. Considering focal authors to still be active if their last publication was 4 or 6 years before the end of our sampling period, or ignoring publication gaps of up to 6 years, yielded highly comparable results (electronic supplementary material, table S8) to our main analysis, where a 2-year publication gap was used as our cut-off for assuming a focal author had stopped publishing.

### (iv) Early effects of collaboration behaviour

Lastly, we conducted an analysis to investigate whether a focal authors' collaboration behaviour in their early career was related to their overall academic career length. To produce as consistent a cohort as possible, we only included focal authors who published their first paper within 1 year of first participating at an ISBE conference and calculated collaboration behaviour metrics using papers published during the first 10 years after their first publication [1] ($n = 390$ authors; 40% women; 4821 publications), a period of time that encompasses the median time to become a PI for both genders. We used GLMs to test for gender differences in adjusted measures of early-career collaboration behaviour and used AFT models to test how collaboration behaviour metrics, gender and their interaction related to career length, as described above.

## 3. Results

### (a) Gender differences in productivity, collaboration behaviour and career progression

Publication number and patterns of collaboration differed significantly according to gender (table 1). Female focal authors published significantly fewer papers, and almost half as many, as male focal authors, and this result held even when correcting for career length as an offset in the model (table 1a). Female focal authors had significantly fewer co-authors (i.e. smaller network size; electronic supplementary material, figure S5.2A), published with their co-authors less frequently (i.e. weaker tie strength; electronic supplementary material, figure S5.2B), and published with co-authors in more connected groups (i.e. larger global clustering coefficient; electronic supplementary material, figure S5.2C), compared to male focal authors (table 1b). These differences persisted once we adjusted for gender differences in productivity (table 1c): when compared to male co-authors and given their number of publications, female focal authors had significantly fewer co-authors (adjusted network size; figure 1d), published with the same co-authors less frequently (adjusted tie strength; figure 1e) and had more connected co-authors (adjusted clustering coefficient; figure 1f).

Gender differences were also apparent in all measures of career progression. Female focal authors were significantly less likely to be a PI than were men (GLM, $n = 935$ focal authors, estimate ± s.e. = $-0.98 \pm 0.16$, $z = -6.21$, $p < 0.001$; figure 1a): in total 708 focal authors became PI; 63% of women ($n = 187$) and 82% of men ($n = 521$). Those female focal authors that did become PIs took 27% more time to do so than did their male counterparts (table 1d and figure 1b). Overall, female focal authors were almost half as likely to remain in academic science as were male focal authors (table 1d and figure 1c) and even after becoming a PI were still more likely to leave academic science than were male PIs (table 1d). Of the 708 focal authors that became PI, 130 subsequently stopped publishing before the

**Table 1.** Gender differences in productivity, collaboration behaviour and career progression. Results from GLMs (*a*–*c*) and AFT models (*d*) testing for gender differences in: (*a*) productivity (total publication number); (*b*) unadjusted metrics of collaboration behaviour; (*c*) metrics of collaboration behaviour adjusted for publication number; and (*d*) career progression metrics. For GLMs, significant relationships are indicated by *p* < 0.05; for AFT models, estimates of the deceleration factor are significantly different from 1.0 (alpha = 0.05) when the 95% CI does not include the value 1.0.

| | *n* (F, M) | mean ± s.e. (range) | | GLM est. ± s.e. | test (z/t) | *p*-value |
| --- | --- | --- | --- | --- | --- | --- |
| | | F | M | | | |
| *(a) productivity* | | | | | | |
| publication number | 935 (298, 637) | 38 ± 2 (3–233)[a] | 65 ± 2 (3–326)[a] | −0.53 ± 0.01 | *z* = −50.21 | <0.001 |
| publication number offset: career length | 935 (298, 637) | 38 ± 2 (3–233)[a] | 65 ± 2 (3–326)[a] | −0.38 ± 0.01 | *z* = −35.50 | <0.001 |
| *(b) unadjusted metrics of collaboration behaviour* | | | | | | |
| network size | 935 (298, 637) | 56 ± 3 (1–450) | 93 ± 4 (1–628) | −0.52 ± 0.01 | *z* = −58.69 | <0.001 |
| tie strength | 935 (298, 637) | 1.88 ± 0.04 (1–5.8) | 2.05 ± 0.03 (1–7) | −0.08 ± 0.02 | *t* = −3.89 | <0.001 |
| clustering coefficient | 930 (296, 634) | 0.48 ± 0.01 (0–1) | 0.40 ± 0.01 (0–1) | 0.30 ± 0.01 | *z* = 171.68 | <0.001 |
| *(c) metrics of collaboration behaviour adjusted for publication number* | | | | | | |
| adjusted network size | 935 (298, 637) | −0.26 ± 0.05 (−3.56–3.41) | 0.12 ± 0.04 (−4.76–4.29) | −0.38 ± 0.07 | *z* = −5.49 | <0.001 |
| adjusted tie strength | 935 (298, 637) | −0.12 ± 0.05 (−1.95–4.37) | 0.06 ± 0.04 (−2.39–5.12) | −0.17 ± 0.07 | *t* = −2.49 | 0.01 |
| adjusted clustering coefficient | 930 (296, 634) | 0.14 ± 0.06 (−2.55–3.39) | −0.07 ± 0.05 (−5.53–4.18) | 0.21 ± 0.07 | *z* = 3.01 | 0.003 |
| *(d) career progression metrics* | | | | AFT estimate (95% CI) | | |
| time to become a PI (years) | 772 (219, 553) | 11.03 ± 0.45 (1–31) | 9.68 ± 0.23 (0–33) | 1.27 (1.13, 1.42) | | |
| career length (years) | 935 (298, 637) | 22.16 ± 0.55 (2–39) | 25.90 ± 0.34 (2–39) | 0.59 (0.49, 0.72) | | |
| career length after becoming PI (years) | 708 (187, 521) | 14.76 ± 0.58 (0–35) | 17.57 ± 0.33 (0–36) | 0.51 (0.32, 0.82) | | |

[a]To enable the construction of meaningful social networks, only focal authors who had published at least three papers were included in analyses (see the electronic supplementary material, S3).

end of our study period; 24% of women (*n* = 44) and 17% of men (*n* = 86).

## (b) Relationships between collaboration behaviour and likelihood of being a principal investigator, time to become a principal investigator and career length

### (i) Likelihood of being a principal investigator

Focal authors were more likely to be a PI if they had more co-authors (i.e. higher adjusted network size), but this relationship was not significantly different between genders (table 2*a* and figure 2*a*). Female focal authors were also more likely to be a PI if they had published with the same co-authors more frequently (i.e. higher adjusted tie strength). However, this relationship was non-significant for men (table 2*a* and figure 2*a*). Focal authors were more likely to be a PI if they had less connected co-authors (i.e. lower adjusted clustering coefficient), but this relationship was not significantly different between genders (table 2*a* and figure 2*a*).

### (ii) Time to become a principal investigator

There was no significant relationship between the number of co-authors a focal author had and how long they took to become a PI for either gender (adjusted network size; table 2*b* and figure 2*b*; electronic supplementary material, figure S9.1A,B). However, focal authors who published more frequently with the same co-authors took significantly less time to become a PI, although the effect size was again not significantly different between genders (adjusted tie strength; table 2*b* and figure 2*b*; electronic supplementary material, figure S9.1C,D). An increase of one unit in adjusted tie strength score decreased the time to become a PI by approximately 18%. Having more connected co-authors increased the time taken to reach PI status for both

genders (adjusted clustering coefficient; table 2*b* and figure 2*b*; electronic supplementary material, figure S9.1E,F). An increase of one unit in the adjusted clustering coefficient score increased the time to become a PI by approximately 6%.

### (iii) Career length

All three measures of collaboration behaviour significantly correlated with career length, but in different ways. Focal authors with more co-authors were more likely to continue publishing than focal authors with fewer co-authors (adjusted network size; table 2*c* and figure 2*c*; electronic supplementary material, figure S9.2A,B), with the effect size being 62% larger for women compared to men. Publishing with the same co-authors more frequently significantly decreased career length (adjusted tie strength; table 2*c* and figure 2*c*; electronic supplementary material, figure S9.2C,D), but the effect size was not significantly different between genders. An increase of one unit in adjusted tie strength decreased career length by 17%. Having many co-authors repeatedly appear together on a focal author's papers also significantly shortened career length and was also not significantly different between the genders (table 2*c* and figure 2*c*; electronic supplementary material, figure S9.2E,F). An increase of one unit in the adjusted clustering coefficient score decreased the chance of continuing to publish by approximately 23%.

## (c) Early-career collaboration behaviour and career length

When we recalculated the collaboration behaviour metrics of a subset of focal authors based only on publications from the first 10 years of their career, we found very similar patterns between collaboration behaviour metrics and

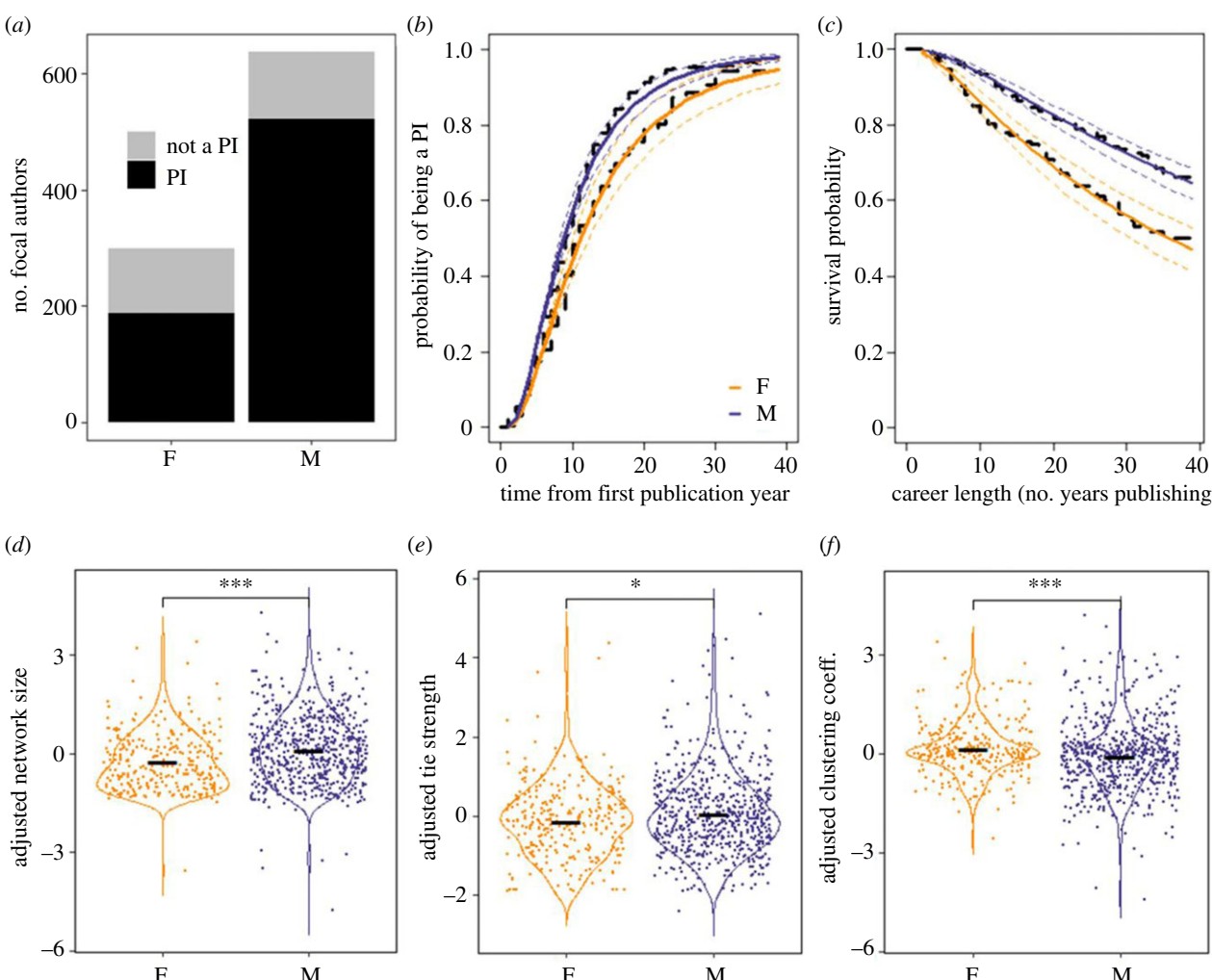

**Figure 1.** Gender differences for (F)emale and (M)ale focal authors in (*a*) the likelihood of being a PI, (*b*) time taken to become a PI, (*c*) career length and (*d–f*) adjusted collaboration behaviour metrics (network size, tie strength and clustering coefficient). See table 1 for sample sizes. In (*b,c*), black dashed lines indicate Kaplan–Meier survival functions, while the solid coloured lines represent model predictions with 95% CIs (dotted lines). The further apart the lines are, the greater the effect of gender on the response variable at that time point. In (*d,e,f*), violin plots visualize the distribution of the data and its probability density. All raw (scaled) data points are displayed, and the horizontal black line indicates the mean. ***$p < 0.001$; *$p < 0.01$. (Online version in colour.)

career length (table 2*d*). Publishing with more unique co-authors, with the same co-authors more frequently, and with less connected co-authors in early career all corresponded with a longer career overall. However, unlike in the complete dataset, there were no gender differences in either the collaboration behaviour metrics (adjusted network size: GLM, $n = 390$ focal authors, estimate ± s.e. = 0.05 ± 0.10, $t = 0.48$, $p = 0.63$; adjusted tie strength: GLM, $n = 390$ focal authors, estimate ± s.e. = 0.10 ± 0.10, $t = 0.95$, $p = 0.35$; adjusted clustering coefficient: GLM, $n = 379$ focal authors, estimate ± s.e. = −0.14 ± 0.10, $t = −1.37$, $p = 0.17$), or in their relationship with career length (table 2*d*).

## 4. Discussion

It is now well established that collaboration behaviour correlates with scientific impact and funding success [12–15,59], including in our focal field of behavioural ecology [13]. Our results extend this work by suggesting that variation in the number of co-authors, how often authors publish with those co-authors, and how connected their co-authors are, may also correlate with academic career progression. However, we found important differences in these relationships for women

and men, as well as stark contrasts in productivity and career progression. Compared to men, women published almost half the number of papers, were almost 25% less likely to become a PI, did so 27% more slowly and had 41% shorter careers. While publishing with more unique co-authors was positively correlated with a researcher's likelihood to be a PI, and their career length, both these relationships were stronger for women than men, although significantly so only in the latter case. Publishing repeatedly with co-authors was associated with researchers taking less time to become a PI, regardless of gender, and also with female focal authors having a higher likelihood to be a PI. Publishing repeatedly within the same clusters of co-authors was negatively correlated with all aspects of career progression, regardless of gender. Therefore, our results are consistent with suggestions that persisting with 'in-house' or 'niche topic' collaborations can become detrimental over time [13,60]. If this is the correct causal interpretation, then our results also suggest that women may benefit more than men from having large collaboration networks characterized by strong relationships (measured here in terms of the average number of publications with each co-author).

If collaborative behaviours have causal effects on academic career trajectories, why might the effects be stronger for women than men? One explanation may be that collaborators

**Table 2.** Output from GLMs (*a*) and AFT models (*b–d*) testing the relationship between collaboration behaviour, gender and (*a*) likelihood of being a PI; (*b*) time to become a PI; (*c,d*) career length. (*a–c*) Are the results based on the full dataset, and (*d*) presents the results based on collaboration behaviour metrics calculated over the first 10 years of career (early career), for those focal authors that started publishing in the year of their first ISBE conference, or later. Because the relationships between collaboration metrics and gender are already presented separately (table 1), results are presented for (F)emale and (M)ale focal authors separately only when the interaction between gender and the metric of interest was significant. For GLMs, significant relationships are indicated by $p < 0.05$; for AFT models, estimates of the deceleration factor are significantly different from 1.0 (alpha = 0.05) when the 95% CI does not include the value 1.0. Significant outputs are highlighted in italics.

| | *n* (F, M) | model | term | model outputs | | |
| | | | | GLM est. ± s.e. | *z*-test | *p*-value |
| --- | --- | --- | --- | --- | --- | --- |
| *(a) likelihood of being a PI* | | | | | | |
| adjusted network size | 935 (298, 637) | metric × gender | interaction | 0.05 ± 0.30 | 0.16 | 0.87 |
| | | metric + gender | metric | *1.69 ± 0.14* | *11.66* | *<0.001* |
| adjusted tie strength | 935 (298, 637) | metric × gender | interaction | *0.35 ± 0.17* | *2.05* | *0.04* |
| | | | F | *0.52 ± 0.15* | *3.37* | *<0.001* |
| | | | M | −0.001 ± 0.09 | −0.01 | 0.99 |
| adjusted clustering coefficient | 930 (296, 634) | metric × gender | interaction | −0.14 ± 0.19 | −0.75 | 0.45 |
| | | metric + gender | metric | *−0.52 ± 0.08* | *−6.10* | *<0.001* |
| *(b) time to become a PI* | | | | AFT estimate (95% CI) | | |
| adjusted network size | 772 (219, 553) | metric × gender | interaction | 1.02 (0.90, 1.17) | | |
| | | metric + gender | metric | 0.98 (0.93, 1.04) | | |
| adjusted tie strength | 772 (219, 553) | metric × gender | interaction | 0.88 (0.77, 1.01) | | |
| | | metric + gender | metric | *0.82 (0.77, 0.86)* | | |
| adjusted clustering coefficient | 772 (219, 553) | metric × gender | interaction | 1.10 (0.97, 1.25) | | |
| | | metric + gender | metric | *1.06 (1.01, 1.12)* | | |
| *(c) career length* (full dataset) | | | | | | |
| adjusted network size | 935 (298, 637) | metric × gender | interaction | *1.62 (1.26, 2.09)* | | |
| | | metric : gender | F | *3.63 (2.91, 4.53)* | | |
| | | | M | *2.45 (2.12, 2.83)* | | |
| adjusted tie strength | 935 (298, 637) | metric × gender | interaction | 1.15 (0.96, 1.38) | | |
| | | metric + gender | metric | *0.83 (0.77, 0.91)* | | |
| adjusted clustering coefficient | 930 (296, 634) | metric × gender | interaction | 1.03 (0.82, 1.29) | | |
| | | metric + gender | metric | *0.77 (0.69, 0.85)* | | |
| *(d) career length* (early-career subset) | | | | | | |
| adjusted network size | 390 (156, 234) | metric × gender | interaction | 1.09 (0.81, 1.46) | | |
| | | metric + gender | metric | *1.30 (1.12, 1.50)* | | |
| adjusted tie strength | 390 (156, 234) | metric × gender | interaction | 1.16 (0.90, 1.50) | | |
| | | metric + gender | metric | *0.78 (0.69, 0.89)* | | |
| adjusted clustering coefficient | 379 (152, 227) | metric × gender | interaction | 1.02 (0.78, 1.33) | | |
| | | metric + gender | metric | *0.78 (0.68, 0.89)* | | |

represent a more important resource for female researchers than they do for male researchers. Women can often be judged more than men on the basis of whom they work with [36], and while authorship positions assigned to women often undervalue their contributions to research, having a female senior author on a manuscript can increase both the overall proportion of female authors, and the probability of a female first author [61]. Researchers publish more often with colleagues of the same gender than is expected by chance [4,62], and high-performing academics (who are more often

men) employ relatively fewer women [63]. This means that gender homophily may influence associated collaboration opportunities and therefore reinforce gender differences in collaboration networks and career length.

A second explanation is that large, strong co-authorship networks with low connectedness may allow at least some projects to continue during periods of hardship for individual team members [64], and these benefits may be especially important for women (e.g. [65]). Constraints on research time are often greater for female researchers than men [65],

*Proc. R. Soc. B* **288**: 20210219

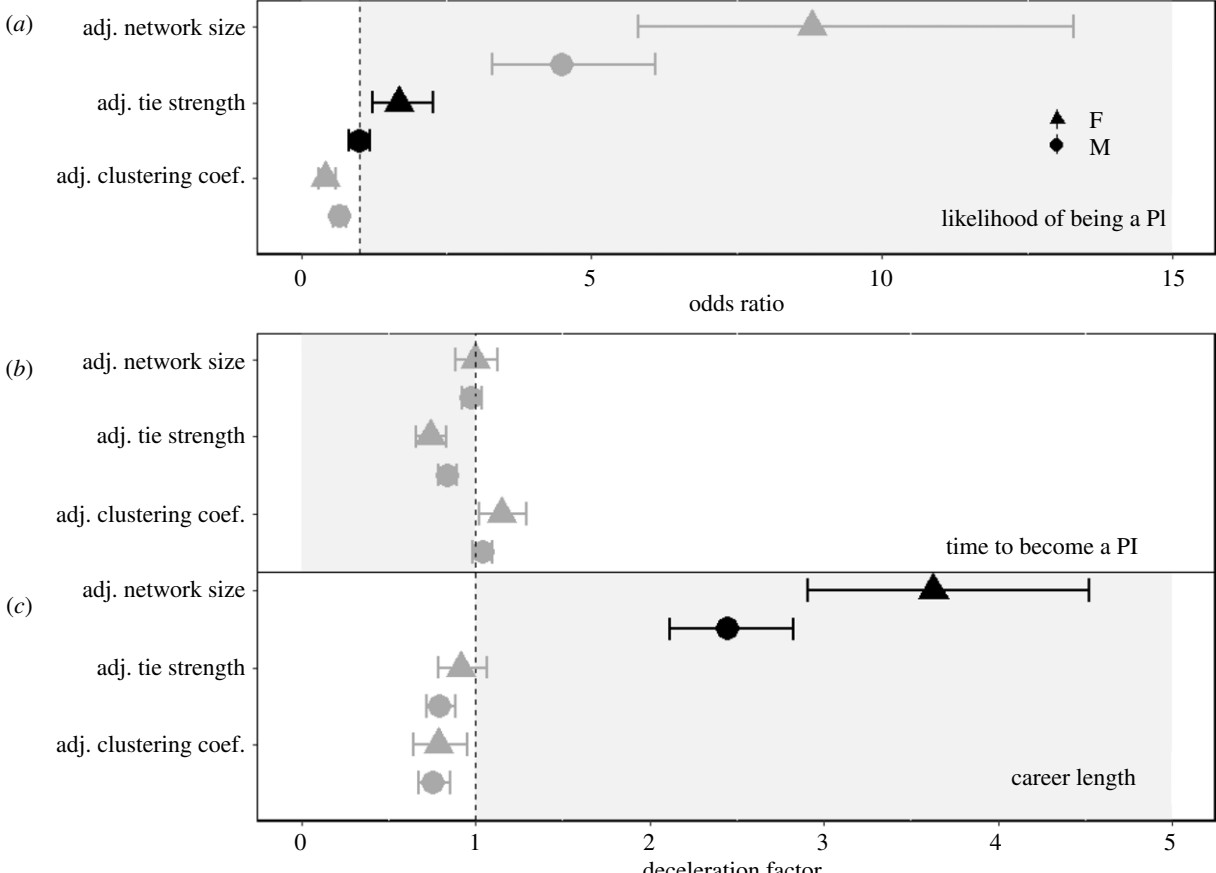

**Figure 2.** The relative effect sizes of publication number-adjusted collaboration behaviour metrics from ego networks on (*a*) the likelihood of being a PI, (*b*) the time taken to become a PI and (*c*) career length, for (F)emale and (M)ale focal authors. See table 2 for sample sizes. Symbols indicate the estimate, with error bars representing 95% CIs. Estimates within the shaded areas indicate a beneficial relationship of the collaboration behaviour on career progression (i.e. greater likelihood of being a PI, reduced time taken to become a PI, greater career length). Estimates with CIs overlapping 1.0 are considered to show non-significant effects. Black symbols indicate that the effect size for a particular collaboration metric differs significantly between female and male focal authors; grey symbols indicate a non-significant gender difference.

and strong collaborator relationships may be important for buffering temporary declines in productivity. For example, collaboration may play a large role in reducing burn-out and buffering productivity from increasing administrative duties [6] and is frequently mentioned as an important factor in researchers' own reflections on their success [66]. Large, diverse, yet strong collaborator networks are also likely to be more resilient to the loss of collaborators (collaborator network resilience, e.g. [64]), which could be especially valuable in networks with high gender homophily, and if female researchers are more likely to leave science than are men.

Our study was, by necessity, correlational and it is possible that collaboration behaviour and career progression are not linked causally. For example, putative causes of gender differences in collaboration behaviour may also explain differences in career progression. Women can have less confidence or self-esteem than male academics [67], which could reduce their propensity to collaborate [68] and affect the securing of faculty positions [63]. Family obligations may also fall more heavily on female researchers, limiting opportunities to travel and make new connections for collaborations [33], as well as raising barriers to job mobility [67]. While we detected similar gender differences overall to other studies of collaboration behaviour [24,25] and career progression [69,70], our statistical approach accounted for differences in productivity. Our results therefore indicate that women who are relatively more

collaborative, given their productivity, are also more likely to persist in academic science. Moreover, these relationships were stronger than those that we detected in men. This suggests that gross differences among genders are unlikely to explain our results, and rather that there are nuances in collaboration behaviour itself. Further studies are, however, needed to pinpoint the characteristics or conditions that underpin both collaboration behaviour and career progression to help disentangle causality.

Although collaboration has been suggested to have only minor benefits for citation impact in ecology compared to the physical sciences ([60] but see [13]), our study provides evidence that, in behavioural ecology at least, collaboration behaviour is associated with career progression. Collaborative working has similarly positive associations with scientific success in computer science [24], and for cell biologists and physicists [16], and our results also show striking parallels with studies across a range of species and contexts that investigate the role of sociality in fitness and longevity [27,71]. Furthermore, our longitudinal analysis suggested that collaboration behaviour established in a researcher's first 10 years of publishing could be formative for career progression (e.g. [49,72]), although we cannot exclude that the characteristics which shape both careers and collaborations are present at an early stage. Compared to our overall dataset, however, we did not find gender differences in collaboration behaviour metrics or their relationships with career length. One reason

could be that this analysis was limited to a smaller cohort of researchers (390 focal authors, one-third of our overall sample), potentially reducing our ability to detect any gender differences (although the proportions of men and women were similar, 40 : 60 versus 32 : 68 in the overall dataset). Alternatively, gender effects on careers may manifest more strongly after the early career stage.

The gender-biased challenges of transitioning to mid-career are now receiving increasing attention as the 'leaky pipeline' continues to leak in biology [73]. Indeed, in our dataset, we found that achieving PI status (i.e. reaching scientific independence) was no guarantee of remaining in science, with female researchers more likely to leave academic science than their male colleagues, especially if they had smaller and weaker collaboration networks. Institutions, funding bodies and programmes are increasingly fostering collaboration [9,72] and while the focus must always be on supporting good science, such collaborative programmes should be encouraged, particularly for women, and across career stages. A quantitative analysis of the importance of collaboration behaviour at different career stages is essential to assess whether the investment in collaboration training will have the desired outcomes for retaining academics in the pipeline [6]. Nonetheless, our study provides empirical evidence of the positive correlations between collaboration and career progression in academic science.

Data accessibility. The data and computational scripts used in this study are available via OSF: osf.io/7v4ep (doi:10.17605/OSF.IO/7V4EP). For purposes of confidentiality, author names and Scopus Author Identifiers (SAI) will not be made available. To illustrate the calculation of social network metrics, we provide a dummy dataset of our own publication records.

Authors' contributions. N.P.C.H. and R.T. conceived the study and all authors developed the concept. All authors collected the data. J.E.M.vdW. prepared the data for analysis and ran the social network analyses and regression models. N.P.C.H. and J.E.M.vdW. carried out the AFT survival analyses. N.P.C.H. and J.E.M.vdW. wrote the initial draft and all authors edited and approved the final manuscript. All authors gave final approval for publication and agreed to be held accountable for the work performed therein.

Competing interests. The authors declare that no competing interests exist.

Funding. This study was supported by Association for the Study of Animal Behaviour (Editorial grant), Natural Environment Research Council (grant no. NE/K00929X/1), Helsinki Institute of Life Science (HiLIFE) and Leverhulme Trust.

Acknowledgements. We thank Neeltje Boogert, Lauren Brent, Alecia Carter, Nick Davies, Chris Duncan, Damien Farine, Rahia Mashoodh, Hannah Rowland, Lotta Sundström and the Evolution, Sociality and Behaviour group at the University of Helsinki for valuable discussions and advice. Wendy King provided access to the ISBE archives, and Luc St-Pierre provided logistical support. We thank four anonymous reviewers for their valuable and comprehensive comments.

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
