## [Peer Review File · Proceedings of the Royal Society B: Biological Sciences]

Review History

RSPB-2020-1024.R0 (Original submission)

Review form: Reviewer 1

Recommendation

Accept with minor revision (please list in comments)

Scientific importance: Is the manuscript an original and important contribution to its field?

Good

General interest: Is the paper of sufficient general interest?

Good

Quality of the paper: Is the overall quality of the paper suitable?

Good

Is the length of the paper justified?

Yes

Should the paper be seen by a specialist statistical reviewer?

No

Do you have any concerns about statistical analyses in this paper? If so, please specify them explicitly in your report.

Yes

It is a condition of publication that authors make their supporting data, code and materials available - either as supplementary material or hosted in an external repository. Please rate, if applicable, the supporting data on the following criteria.

Is it accessible?

Yes

Is it clear?

Yes

Is it adequate?

Yes

Do you have any ethical concerns with this paper?

No

Comments to the Author

Although I can't say I enjoyed reading this paper. I do think it was a very interesting and well executed study. It is actually a study I have personally thought of conducting, but without experience in social network analysis had never gotten around to it. So I am glad that someone finally has.

In their study the authors conduct an ego centric social network analysis using data obtained from authorship of scientific papers within the field of behavioural ecology. The authors delineate their study group by using only people that were listed as participants in conferences run by the International Society for behavioural ecology in the 1990's - although this of course limits the conclusions the authors can make from their study, it is a clever way to ensure a manageable dataset while simultaneously ensuring they obtained data from active scientists and that these scientist would have had time to see the career progression/decline that the authors are interested in.

I hope that my comments, help to improve the manuscript.

Major comments:

1. To be honest, I'm not sure how I feel about the use of the term sociality in the context of this paper - particularly in the abstract where you have not had a chance to explain your definition of the term. Equating tendency to collaborate with being more social is I think problematic. While I can see why this is appealing in terms of "selling" the paper, I think it is a little misleading for several reasons. One, I think people can be very social but not very collaborative - do you have any data to suggest that more social people are more collaborative? Two I think even if I were translating collaboration into an animal context I don't think collaboration and sociality are interchangeable - I would put collaboration more in the realms of alliances or cooperating to achieve a common goal, even at a species level I'm not convinced these are related necessarily related to sociality. This is something that I think needs to be changed throughout the manuscript, and which may change the tone of the introduction considerably.

2. While I agree with the conclusion that encouraging people to invest in collaborations with women will help to address the gender gap in science, I'm not sure I agree with the motivation - ie collaborate to aid career progression. One day I would like to see conducting good science as the motivation. Perhaps the end of the abstract could be rephrased at the moment it sounds a little like benefits career progression is more important that the science, which I'm sure isn't the intention.

3. I am also not convinced that women collaborate less than men - but rather suspect that part of the problem is that women's contributions don't receive the same level of recognition. I have seen this over and over again, particularly for ECR's, so perhaps it's not just about investing in these collaborations but also recognising the work that women do. I know this comment errs

toward personal opinion – I have no concrete data to back it up, so if you have good reason to disagree that's fine, but it might be worth considering further.

Line 56: While people are obviously social animals I think adding this phrase “which scientists surely are” is a little strange as it implies scientists are under similar selection pressures as social animals. Most of the evidence you provide in the next sentences suggests that the advantages of collaborating are more related to status than they are to learning, so I think the similarity might not be as strong as implied.

Line 71: I don't think you really look at “social strategies”. I would reword to reflect more accurately what you did.

Line 85: I thought this in a few places throughout the manuscript, but this is the first place. At what stage of the focal authors career did you “measure” sociality. I think from reading further into the manuscript you did this across the authors entire career and that you did a check to see whether just looking at early career data changed things. I think it is worth making this clear earlier

Line 103: Ok so I had been wondering about this... I think it would be worth saying briefly how you do this here.

Line 118-119: This sentence assumes that everyone reaches PI status if they want to... I'm not sure this is the case. I know plenty of people who have tried to obtain PI status and not succeeded.

Line 126: Do you mean gender or sex here?

Line 166: I'm not sure you mean “further compounded by” here. That would mean that gender made the differences even bigger. I think what you mean is that the difference in number of papers that women publish compounded/generated gender differences in these metrics (depending on the metric in question)

Line 177: I got confused about this when reading the supplement because in the figure axes you refer to these measures as degrees, strength and clustering. I would define the terms collaborativeness, consistency and connectedness as early as possible and then use these terms throughout.

Line 185-189: Excuse me if I'm being dumb, but I've read this sentence 3 times and can't see the difference between your definition for consistency and connectdeness.

Line 194: In your introduction you kind of equate career progression in academia with fitness and survival, which made me wonder here whether your definition of a PI (ie three papers as last author) is really a very good definition of longterm success, this also stood out when you mentioned that some people became PI in the same year they first published. How are single author papers treat here? What if someone had a last author paper every 10 years. What if the focal author was on a fellowship and then went back to being a postdoc? I understand that figuring out when people got permanent jobs or made tenure etc might be possible for your dataset, and I think that this measure is probably reasonable, but perhaps it is possible to validate it further, or provide further acknowledgement that your measure does not necessarily imply someone has obtained a permanent job.

Line 223: I'm surprised by this, I thought behavioural ecology was much more incestuous than that □

Line 246: You control for number of papers but I think behaviour could change with career progression in a way that is not related to number of papers, so when in the career were these metrics calculated? This still isn't clear yet.

Line 246: Why run individual models for each of the social networks, and not put them all in one model? Are they correlated? I know in the supplement you show correlations for these prior to controlling for number of papers, but what about afterwards?

Line 257: Do you mean if someone didn't become a PI they were essentially considered as never having been born? what about people who continued to publish throughout the career but never acheived PI status... or what about people who had 3 last author papers but they were all really far apart? Do they still count as PIs?

Line 265: were longer gaps mostly attributed to women? I know you have this in your supplement, but I think its worth also stating here, since this might be expected as a result of women being more likely to take time out due to caring responsibilities. Given that 20% (which seems like a considerable amount) had a great than 2 year gap and looking at figure esm9a I would have thought 4 years would have been a better choice. Although given you show the

results are comparable across all time frames, I don't suppose it's a big deal.

Line 277: add "for" between published and a

Line 279: Why did you use 10 years here? I think you said above that the median time till becoming a PI was 8 years, so would that be a better cutoff?

Line 293: I know you discuss other papers that do this, but I wondered whether it was possible to look at the sex ratio of coauthors with your data and whether it differs between the sexes ie is their sexual segregation and are focal authors that collaborate more with males more likely to do well in their careers? And does that differ depending on the sex of the focal author.

Line 306: How do you know they identify as a male?

Line 317: again I wondered how these measures correlate with each other, if they are highly correlated and you are conducting separate models for each, they might not really be saying anything different...

Line 366-367: should it be and/or? They didn't have to do all these things to become more likely to become a PI.

Line 370: this made me wonder whether people who are more collaborative early on are also more collaborative later? Does this differ for men and women? I feel like the things driving collaboration are different early and late in your career - early on it might be driven by confidence or having a specific skill, whereas later in your career it might be driven by how many students you take on. Likewise I guess early behaviour can go on to shape later behaviour and also drive long lasting changes in perception...

Line 372-377: I'm not sure I get the logic here

Line 379-383: You say you found differences in career productivity, but also that women had shorter careers. If you correct for time in academia do you still see differences in productivity?

Line 388-389: I think this is almost certainly true. I think women are more likely to be questioned about whether the work presented was theirs or not. However, I'm not sure its who the collaborators are but rather the number that matters - potentially having multiple different collaborators is important, not necessarily the identity of those collaborators. I wonder if it is possible to tease this out with your analysis by comparing the association between variance in male or female success with you metrics.

Line 391: I would argue that authorship full stop is allocated to the detriment of women eg men more likely to be put on papers for doing less and that PIs should be encouraged to reflect on how they allocate authorship and whether it is equitable.

Figure 1B: interesting although I suppose not surprising that once women get to 15-20year career length the rate of decline appears to be equal for men and women

Figure 2: I find these figures difficult to interpret, maybe add info on how to interpret to the legend. eg the further the lines are apart the stronger the effects??? Is it possible to use response surface type figures or contour plots instead?

Table 1: Instead of the shading a more standard approach for highlighting significant results would be to bold the significant ones or highlight them with asterisks

Supplement line 132: Is the degree supposed to be 4? I only see 4 unique coauthors? I does this number include the focal author?

Supplement line 161: delete "to those of" I think in the previous part of the sentence the subject is the authors, not the papers of the authors

Supplement Line 164: are these metrics highly correlated once you control for the number of papers? Also can you give them the same labels as what you use in the main paper

Supplement line 168: are consistency and connectedness the same as strength and clustering respectively, would be better to keep terms consistent throughout and with paper

Review form: Reviewer 2

Recommendation

Major revision is needed (please make suggestions in comments)

Scientific importance: Is the manuscript an original and important contribution to its field?

Good

General interest: Is the paper of sufficient general interest?

Good

Quality of the paper: Is the overall quality of the paper suitable?

Marginal

Is the length of the paper justified?

Yes

Should the paper be seen by a specialist statistical reviewer?

Yes

Do you have any concerns about statistical analyses in this paper? If so, please specify them explicitly in your report.

Yes

It is a condition of publication that authors make their supporting data, code and materials available - either as supplementary material or hosted in an external repository. Please rate, if applicable, the supporting data on the following criteria.

Is it accessible?

Yes

Is it clear?

No

Is it adequate?

No

Do you have any ethical concerns with this paper?

No

Comments to the Author

I found this manuscript interesting to think about. I appreciate the authors have already put in a considerable amount of work (and might be disappointed by the length of this review). Although substantial, all my concerns can be addressed in a revision. Below I raise three concerns that will require the most effort to address, followed by quicker fixes. First, I have misgivings about how gender differences in productivity were controlled for. In addition to some statistical concerns, I question the validity of the current approach, given that gender differences exist not only in the quantity of published papers, but also the prevalence of author positions. Second, I disagree with the manuscript's current 'pitch' (relating sociality and survival to authorship networks and academic careers), and recommend the authors re-write these aspects of the article with more focus on meta/social-science. Third, I question the interpretation of three aspects of the study: the definition of 'principal investigator', claims of directionality of effects, and implications for reducing the gender gap in science.

#--MAJOR COMMENTS--#

****1. Accounting for gender differences in productivity****

There are well-established gender differences in publication outputs, which inevitably create gender differences in co-author networks. Therefore, interpreting gender differences in co-author networks requires careful consideration of how that 'productivity gap' is controlled for. I am concerned that in its current form, the manuscript does not adequately explain and justify the

chosen methods. While there is likely no ‘perfect’ solution, it would be worthwhile to test whether alternative analysis have a qualitative impact on the findings.

Currently, the authors have analysed three types of residual statistics, taken from generalized linear regressions where the covariate is the number of papers per author:

(1) ‘Collaborativeness’ are the z-scaled residuals taken from a Poisson regression, where the response is the number of unique co-authors per author.

(2) ‘Connectedness’ are the z-scaled residuals from a logistic regression, where the ‘successes’ are the number of ‘triangles’, and the ‘failures’ are the number of triangles divided by the global clustering coefficient.

(3) ‘Consistency’ are the residuals from a model of two z-scaled residuals: the number of unique co-authors per author (‘collaborativeness’, above) is the covariate (fixed effect), and the response variable comes from the Gaussian model of the natural logarithm of $1 +$ the number of papers divided by the total number of co-authors (tie strength).

Details of these analyses in the methods section (both main text and supplement) are sparse. For example, error structures and the $\ln(x+1)$ transformations are not specified (and even within the analysis script, there is no justification for why $\ln(x+1)$ was used, rather than $\ln(x)$, given the data made available do not show any zeroes in x)

I am not a statistician (and I hope the editor invites one to review this manuscript), but intuitively I suspect analysing these types of residuals could be problematic. Other than the seeming difficult of interpreting effect sizes from these models, I’ve thought of the following potential concerns (listed in no particular order):

(1) errors in the residual estimates are not propagated to later analyses. Analysing the posterior distribution of residuals from Bayesian models could be a solution. Or just including number of papers as a covariate in the main models, rather than extracting the residuals?

(2) the authors do not report any indices of model fit (e.g. R^2 values), and residuals from poorly-fit regression lines could lead to biased conclusions. E.g. see “Forstmeier, W. (2011), ‘Women have Relatively Larger Brains than Men: A Comment on the Misuse of General Linear Models in the Study of Sexual Dimorphism’, *The Anatomical Record*, 294:1856-1863. <http://doi.org/10.1002/ar.21423>”.

(3) Is there something strange about the ‘Consistency’ residuals, where residuals from closely related metrics (Fig. ESM6.1; $r = 0.95$) are regressed against each other? Do multiple levels of correlation cause statistical issues here? If not, how does one interpret one “unit” of this resulting response variable?

Perhaps more importantly, I’ve thought about a potential problem with correcting for the total number of publications, without considering the focal author’s position on each publication. Currently, the residual correction assumes that X number of publications is the same thing for female and males. But as the authors mention in the discussion, female and male authors differ in both the magnitude and types of publications. For example, women might be more likely to be middle authors (and have carried out the research, rather than ‘leading’ it).

See the following references:

West, J.D., Jacquet, J., King, M.M., Correll, S.J., Bergstrom, C.T. (2013). The Role of Gender in Scholarly Authorship. *PLOS One*, 8(7): e66212. <https://doi.org/10.1371/journal.pone.0066212>

Macaluso, B., Larivière, V., Sugimoto, T., Sugimoto, C.R. (2016). Is Science Built on the Shoulders of Women? A Study of Gender Differences in Contributorship. *Academic Medicine*, 91(8) p 1136-

1142. <http://doi.org/10.1097/ACM.0000000000001261>

(But this effect could differ with career stage, with male PhD students more likely to be credited with publications:

David, F.F., Peugh, J., Maher, M.A., Roksa, J., Tofel-Grehl, C. (2017). Time-to-Credit Gender Inequities of First-Year PhD Students in the Biological Sciences. *CBE – Life Sciences Education*, 16:1. <http://doi.org/10.1187/cbe.16-08-0237>)

Imagine that for a given number of papers, female authors have a higher proportion of middle author papers. We can assume that, compared to first and senior-author papers, middle author papers contribute less to facilitating future research opportunities (i.e. collaborations), career longevity, and seniority. We know that the dataset contains more male than female authors, therefore the modelled relationship between the number of publications, and co-author network indices, will tend to reflect the relationship between the ‘male’ proportion of middle-authored papers. Therefore, for the same number of papers, the female estimate of the ‘impact’ of that productivity on her career might be below the regression line (resulting in a negative residual). Indeed, in the available dataset, the average residual value for the number of unique co-authors, and publications-per-co-authors, is negative for females and positive for males, whereas the pattern for the clustering coefficient (‘cliqueyness’?) is reversed.

All this is to say, given the hidden gender differences in the ‘prestige’ of publications, I think it is simplistic to correct for the number of publications alone. I recommend the authors distinguish between first, middle, and last-author publications in their analyses. It appears this information has already been extracted in section 3 of ‘Social authors code_part1.Rmd’, although these data are not made available for subsequent analyses.

2. Article pitch

The manuscript equates persistence in scientific publishing with ‘survival’, and co-authorship collaborations with sociality. To me, this pitch – which is set up in the second paragraph of the introduction – is an overreach. (A more minor point: in the opening line of that paragraph, which is in the context of non-human animals, perhaps ‘gender’ should be changed to ‘sex’).

Case in point: Lines 65-69 state “Furthermore, increased sociality is associated with greater ability to tolerate stress[12,19]. If a larger social network helps researchers to better cope with the stresses of a career in science and academia[20], then we might also expect a positive correlation between career lifespan and sociality, with less sociable individuals leaving science earlier.” Breaking this down, “increased sociality is associated with reduced stress” cites “Social relationships and mortality risk: a meta-analytic review” (<http://doi.org/10.1371/journal.pmed.1000316>; reporting social relationships correlate with longer human lifespans), and “Social buffering: relief from stress and anxiety” (<http://doi.org/10.1098/rstb.2006.1941>; where social buffering is defined as “when conspecific animals are together, they show a better recovery from aversive experiences”). I do not think the types of social relationships described by these papers are equivalent to the social relationships in a co-author network. Researchers frequently publish with people they have never even met. Quite simply, co-authorship does not evoke the type of interpersonal relationship that buffers people from the stresses and anxieties of life. It is entirely conceivable that there is an inverse relationship between co-author network scores and ‘sociality’ (as traditionally measured), whereby academics who devote more time to co-authorship have less time to spare for maintaining their personal friendships and familial relationships. It therefore does not follow that “If a larger social network helps researchers to better cope with the stresses of a career in science and academia[20], then we might also expect a positive correlation between career lifespan and sociality, with less sociable individuals leaving science earlier.” (also, citation 20 is not clearly relevant – “Staff Wellbeing in Higher Education” is a report based on 25 staff members at higher education institutes, not specific to the sciences).

I recommend the authors replace ‘sociality’ with a word or phrase that better describes what is actually measured in this study (e.g. ‘co-author network indices’). Alternatively, the metrics could simply be referred to separately (as is done intermittently as ‘collaborativeness’, ‘consistency’ and ‘connectedness’, although I think ‘cliquey’, as used by citation 22, is more descriptive?). Less importantly, the use of ‘publishing career length’ instead of the snappier ‘survival’ has the advantage of not implying the mortal failure of ceasing to publish (e.g. citation 45). In general, the manuscript would be improved by focusing on what the co-author network metrics specifically imply and confer (e.g. nepotism, Matthew effects), and less on the nebulous links to the non-human literature, such as the associations between reproductive success and sociality.

3. Study interpretation

~~Definition of ‘Principal investigator’~~

I was surprised by the definition of a Principal investigator given at lines 192-193: “following the definitions in[44], we classified any focal author with at least three last-author publications as being a PI”.

I vaguely consider the status of ‘Principal investigator’ to confer more career stability than simply being the last author on three publications (e.g. a post-doc might publish with students in the laboratory they work for, attaining last-author publications without attaining the ‘power’ inferred by ‘PI’). Indeed, the Wikipedia page for ‘Principal investigator’ says that, in Canada and the US, PI “is also often used as a synonym for “head of the laboratory” or “research group leader.”” (PI is also defined this way in <https://doi.org/10.1371/journal.pcbi.1007448>). The citation given (44: van Dijk et al. 2014, <http://doi.org/10.1016/j.cub.2014.04.039>), does not define ‘PI’ in the main text, and the supplemental methods suggest that this definition was chosen arbitrarily and for convenience, without being validated (“To ensure a robust estimation of whether someone becomes PI, we consider as becoming PI only those authors that have at least three last author publications and measure the time to PI as the time between that person’s first publication and the time of the second last-author publication. We note that our results are highly robust to variations on this method, such as using the first two or four last author publications.”).

Therefore, the current justification for the definition of PI amounts to “someone else did it this way”. To continue using ‘PI’ throughout the manuscript, I think the authors should validate whether having three last-author publications is strongly associated with being a lab leader. For this historical dataset this task seems, at best, tedious. An easier approach would be to simply replace the use of ‘PI’ with ‘at least three senior-author publications’. While less elegant, this definition won’t mislead readers about what the study actually measured. I think this point is important, given the aforementioned research on male authors being more likely to secure that last-author position. The authors could alternatively use the time taken to publish one senior-author paper (and simply define this as “time to senior author”).

~~Assessing directionality of the effects of sociality~~

The methods paragraph from lines 273-281 says that directionality was assessed by seeing “whether a focal authors’ sociality in their early career predicted their future academic survival”, where ‘early career’ is defined as the first ten years of publishing, for the cohort of researchers who had not published long before attending ISBE. The discussion (lines 369-371) says “Importantly, the extent of sociality during a focal authors’ early career predicted their future survival, suggesting that establishing collaborate behaviour early drives career success, rather than the other way around.” I don’t follow this logic. Early career researchers who express traits that are linked to success (e.g. having a specialised and useful skill, being a good networker, etc) could foster more collaborations, but that does not mean that those collaborations are what caused their career success (i.e. the unmeasured researcher traits could be causing the indirect correlation between the network indices and career success).

~~'Fixing' the gender gap~~

There is a repeated message throughout the manuscript that the study's results can help address the gender gap in science. I don't think this conclusion is warranted; the authors describe an underlying pattern, but do not provide evidence of the causes. There could be something systematic that prevents women from collaborating more, in which case an individual approach (suggesting women seek out more collaborations) is a bit like suggesting women long-jump athletes, to close the pay gap, simply jump as far as their male counterparts.

Specific examples of this messaging are:

Last sentence of abstract: "Encouraging researchers at all levels to invest in collaborations, particularly with female researchers, will help to close the gender gap in science and academia."

Introduction, Lines 72-75: "Understanding the connections between sociality and career progression and duration, and particularly how this might differ between genders, could shed light on strategies that researchers may wish to follow, and provide greater understanding of the underlying causes of gender disparities in the sciences."

Discussion, lines 413-44 and 417-418 "Our findings suggest that for scientists early in their career, seeking out collaborations has long term-benefits", and "all researchers, regardless of gender, could benefit from seeking out collaborations early on in their career."

Concluding statements, lines 436-441: "In conclusion, our results suggest that all researchers – but particularly those that are women – can enhance their career progression and survival in science by collaborating widely and repeatedly. Creating research environments that encourage collaboration – across disciplines and institutes, among career levels, and especially between genders – will lead to greater and more rapid scientific advances[3], and could assist in reducing the gender gap in science."

#--MINOR COMMENTS--#

****Computational reproducibility****

I thank the authors for making analysis scripts available, in addition to processed data, but have some simple suggestions for improvements.

~~Data availability~~

Pre-processed data has not been made available, with the justification that "For purposes of confidentiality, author names and Scopus Author Identifiers (SAI) will not be made available." Is this protection of author identity necessary, given that SAI's only contain publicly available information? If it is, then perhaps the authors could include a hypothetical dataset of SAI's (e.g. with their own identifiers), in the same style as the real dataset, so that the code 'Social authors code_part1.Rmd' can still be run (currently, it is hard to really tell how the script works, without having real data to play with).

~~Reproducibility of code~~

In 'Social authors code_part1.Rmd', section 8 currently calls the wrong columns. Columns 17 and 21 are being renamed, but only 13 columns are presented in the available dataset. To make this code more robust, the column names could be called directly with their names, rather than relying on their number (e.g. `names(new.corr)[names(new.corr) == "indegree"] <- "degree"`).

In 'Social authors code_part2.Rmd', the wrong file name was used for importing the data (i.e. doesn't match the name of the data file that was uploaded).

For plotting results figures, the numbers are manually entered. To minimise mistakes, it would be best to call them directly from the model (e.g. 'plotM6a' numbers were slightly different to what was manually entered). For Figure 4 it was not clear which models were used to extract the

plotted numbers. It was also not clear in the code which models were used for the results tables.

~~Missing version numbers for R packages~~
 ‘rscopus’ (line 128)
 ‘bibliometrix’ (line 132)
 ‘asnipe’, ‘igraph’, ‘itergraph’, and ‘network’ (line 140)
 ‘lmvar’ (line 183)

****Wording of the abstract****

In the opening sentence of the abstract - “Intense competition for limited opportunities means the career path of a scientist is a challenging one, and female scientists in particular are less likely to survive in academia.” - the use of ‘in particular’ implies that gender is the biggest inequity in career length, rather than being one of many factors (e.g. racial identity, socioeconomic status).

“We built authorship social networks from publication records to test how sociality predicts career progression and survival in biologists contributing to three international conferences in the 1990s”. Had I only read the abstract, I’d think the study had taken three different conferences, covering different aspects of biology, in the 1990s, rather than three iterations of the same conference. A more accurate summary would be something like “To test how sociality predicted career progression, we built authorship social networks (corrected for productivity) from publication records of behavioral ecologists who had attended a meeting of the International Society of Behavioral Ecology in 1992, 1994, or 1996.”

Abstract line 37, “publishing with many diverse co-authors”, Discussion lines 368: “Being more collaborative with a diverse set of co-authors”. Many readers will associate ‘Diverse’ with diversity initiatives, e.g. gender, race, field of study, socio-economic background. Perhaps better to use “unique” or “distinct”, to convey that the co-authors are different people, rather than the more ambiguous “diverse”.

****Other comments****

Lines 62-63: “articles mentioned on social media gain more citations[16]”. I think this statement is premature, given the limited evidence on this topic. The paper cited as [16], ‘Tweeting birds: online mentions predict future citations in ornithology’ (<http://doi.org/10.1098/rsos.171371>) looks at only a subset of the scientific literature, and tests for a correlation between Altmetric scores (which includes both traditional and social media) and citation counts from the year 2014. Even if there is a robust correlation between social media attention and citation counts, this does not prove that social media attention increases citations (e.g. more citable papers could be more tweetable - see also <https://doi.org/10.1371/journal.pone.0183551>).

Line 87-88:

“Our cohort consisted of contributors to three consecutive conference of the International Society for Behavioral Ecology in the 1990s” □ I misunderstood this to mean that the cohort was people who went to three consecutive ISBEs, not people who had been to any of the three

Line 92-93:

“gender bias in publishing is unlikely to be an issue[31,32] (although see[29])” □ this claim is too broad to be supported by the given evidence. ‘Gender bias in publishing’ could refer to any stage in the publication process, from manuscript conception to appearing in a journal. Citations 31(<https://doi.org/10.1371/journal.pone.0201725>) and 32 (<https://doi.org/10.1016/j.biocon.2009.06.021>) do not find gender bias in the editorial decisions at two journals, but this does not discount other types of biases (e.g. citation 31 suggests women are under-represented as corresponding authors).

Misplaced parenthesis lines 169-170: “published with co-authors that were less connected lower (global clustering coefficient; Fog.ESM6.2C). □ ‘lower’ should be inside the parenthesis

In the main text, the same description – “published more often with the same co-authors” – is given for interpreting the ‘consistency score’ and ‘connectedness score’ (lines 185-189). I think this is a typo, because the two indices must have slightly different interpretations? I found the supplementary Figure ESM5 helpful for understanding the indices, and think it could be usefully included in the main text.

Discussion, Line 306: “Identifying as male” □ given that gender was inferred from “first name and/or online profile”, rather than asking authors to identify their gender, perhaps this should instead be “Being identified as male”

Discussion, Lines 382-386: “The shorter careers and higher dropout rates of female scientists explain a large proportion of their reduced productivity and impact[7]. However, these factors likely also restrict opportunities for enhancing sociality by establishing new collaborations, thereby further contributing to less favourable career outcomes for female scientists.” What is the connection between these two sentences? (And aren’t “shorter careers” and “higher dropout rates” the same thing?) The second sentence says “these factors” (i.e. having a shorter career) reduce opportunities for collaborations, but this is trivial: women who stop publishing do not form new collaborations. That pattern should not affect the outcome of this study, though, given that the analysis tried to control for the number of publications (which would be correlated with career length)?

Discussion, lines 411-413: “Junior scientists are typically pursuers of collaborations, while senior researchers are net attractors, having new collaborations proposed to them [59]” □ Citation 59 offers this interpretation for the pattern that the duration of collaborations decreases with career age (“Quantifying the impact of weak, strong, and super ties in scientific careers”: <https://doi.org/10.1073/pnas.1501444112>), but this seems simplistic. Senior researchers can easily propose opportunistic collaborations with ERCs who have the time or motivation for the time-consuming aspect of the project (e.g. students of collaborators, visitors), resulting in the same pattern without those senior researchers being ‘pursued’.

Alternative explanation for why “the effects of author sociality on career progression and length were consistently stronger for women than for men” (Discussion, lines 386-388) □ time use data from around the world consistently show that women have less free time than men. Stronger co-author networks, which can keep balls in the air when the focal author drops the ball, could therefore be more important for women, who simply cannot attend to those balls as often (e.g. look at productivity data during COVID-19).

Review form: Reviewer 3

Recommendation

Major revision is needed (please make suggestions in comments)

Scientific importance: Is the manuscript an original and important contribution to its field?

Marginal

General interest: Is the paper of sufficient general interest?

Marginal

Quality of the paper: Is the overall quality of the paper suitable?

Marginal

Is the length of the paper justified?

Yes

Should the paper be seen by a specialist statistical reviewer?

No

Do you have any concerns about statistical analyses in this paper? If so, please specify them explicitly in your report.

No

It is a condition of publication that authors make their supporting data, code and materials available - either as supplementary material or hosted in an external repository. Please rate, if applicable, the supporting data on the following criteria.

Is it accessible?

Yes

Is it clear?

Yes

Is it adequate?

Yes

Do you have any ethical concerns with this paper?

No

Comments to the Author

This paper explores collaboration differences between men and women authors relate to PI status and career length. While the question is of great interest to a broad interdisciplinary scientific community, several questions with the chosen methodology suggest a revision is necessary before publication in Royal Society B.

~ I would like to see Fig ESM3 broken down by gender (say using a stacked box plot) and with the appropriate log-binning (Fig ESM3 A,C are heavy-tailed distributions).

~ many effects are driven by the heavy-tailed nature of publications and citations — thus, unless there is reason to suspect name disambiguation errors, the 6 focal authors with more than 400 papers should be included. Were any of these women?

~ since the number of publications displays strong gender differences, the correction procedure is very important. It looks like linear and normality assumptions were used to build the adjusted measures of sociality. Please provide justification that the residuals are normally distributed, and this standardization procedure did not actually exacerbate the gender differences in productivity.

~ the definition of PI as time to 2 last-author papers is consistent with other measures of seniority and the disciplinary norms of this field.

~ The authors attempt to control for female author name changes is applauded. However, in practice, we have found that contemporary male authors are also likely to change their last names (typically hyphenated), and the rate of real author name changes is much smaller than the rate of mis-identified authors, last-name spelling differences (particularly with latinized eastern-european names and east-asian names), name order errors (last & first names flipped), missing prefixes or suffices, and missing punctuation — all of which contribute to noise in the reconstruction of publication careers.

~ are the survival results consistent if the dependent variable is switched to publication sequence from real-world time?

~ A similar analysis was conducted in:

Jadidi et al (2018) Gender disparities in science? Dropout, Productivity, Collaborations and Success of Male and Female Computer Scientists. *Advances in Complex Systems*. 21. 1750011
Could the authors put their findings in the context of these results?

~ Can the authors offer a more specific interpretation for these findings in terms of ecology field norms and practices?

Decision letter (RSPB-2020-1024.R0)

04-Aug-2020

Dear Dr van der Wal:

I am writing to inform you that your manuscript RSPB-2020-1024 entitled "Sociality enhances survival in science, especially for female researchers" has, in its current form, been rejected for publication in *Proceedings B*.

This action has been taken on the advice of referees, who have recommended that substantial revisions are necessary. With this in mind we would be happy to consider a resubmission, provided the comments of the referees are fully addressed. However please note that this is not a provisional acceptance.

Sincerely,
Dr Robert Barton
mailto: proceedingsb@royalsociety.org

Associate Editor

Board Member: 1

Comments to Author:

I have received three reviews for your manuscript entitled “Sociality enhances survival in science, especially for female researchers”. Two reviewers thought your paper was overall interesting and the analysis generally well done. Reviewer #3 pointed out that the methodology is not entirely novel: another study published in 2018 already investigated the same issue using a similar methodology, albeit studying a different researcher community (computer scientists). I would recommend adding the mentioned paper (Jadidi et al) to your reference list and relate to this study in a revision. Your study, I believe, will nevertheless still be interesting for the Proc B readership (even if indeed methodologically not entirely novel) because it investigates data from within the biology researcher community. That said, please also make more obvious in your abstract that the data is drawn from a single conference but different years – this is indeed confusing now. I also hope that you may find inspiration from the Jadidi paper as well as the reviewers’ comments for how to attend to several remaining statistical issues (in particular the correction procedure) and issues of interpretation (in particular concerning causality). Those issues really need to be convincingly addressed and resolved for the paper to be considered for publication in Proc B. All three reviewers also questioned your methods to identify PI’s – please also take more care here. Finally, two reviewers saw problems with the conceptual embedding of the paper: they did not really see the need for a close connection between the concepts of scientific collaboration and (biological) sociality. Reviewer #2 made several good suggestions for how to reframe your study. I also whole-heartedly agree that in the end, what we should strive for is not collaboration per se. Instead, as reviewer # 1 so nicely put it, “I would like to see conducting good science as the motivation”. I hope you find the reviewers’ suggestions helpful to revise your manuscript, and I would be looking forward to see your paper in a revised form. In agreement with at least two of the reviewers, I think it holds a lot of potential.

Reviewer(s)' Comments to Author:

Referee: 1

Comments to the Author(s)

Although I can't say I enjoyed reading this paper. I do think it was a very interesting and well executed study. It is actually a study I have personally thought of conducting, but without experience in social network analysis had never gotten around to it. So I am glad that someone finally has.

In their study the authors conduct an ego centric social network analysis using data obtained from authorship of scientific papers within the field of behavioural ecology. The authors delineate their study group by using only people that were listed as participants in conferences run by the International Society for behavioural ecology in the 1990's - although this of course limits the conclusions the authors can make from their study, it is a clever way to ensure a manageable dataset while simultaneously ensuring they obtained data from active scientists and that these scientist would have had time to see the career progression/decline that the authors are interested in.

I hope that my comments, help to improve the manuscript.

Major comments:

1. To be honest, I'm not sure how I feel about the use of the term sociality in the context of this paper – particularly in the abstract where you have not had a chance to explain your definition of the term. Equating tendency to collaborate with being more social is I think problematic. While I can see why this is appealing in terms of “selling” the paper, I think it is a little misleading for several reasons. One, I think people can be very social but not very collaborative – do you have any data to suggest that more social people are more collaborative? Two I think even if I were translating collaboration into an animal context I don't think collaboration and sociality are interchangeable – I would put collaboration more in the realms of alliances or cooperating to achieve a common goal, even at a species level I'm not convinced these are related necessarily

related to sociality. This is something that I think needs to be changed throughout the manuscript, and which may change the tone of the introduction considerably.

2. While I agree with the conclusion that encouraging people to invest in collaborations with women will help to address the gender gap in science, I'm not sure I agree with the motivation – ie collaborate to aid career progression. One day I would like to see conducting good science as the motivation. Perhaps the end of the abstract could be rephrased at the moment it sounds a little like benefits career progression is more important than the science, which I'm sure isn't the intention.

3. I am also not convinced that women collaborate less than men – but rather suspect that part of the problem is that women's contributions don't receive the same level of recognition. I have seen this over and over again, particularly for ECR's, so perhaps it's not just about investing in these collaborations but also recognising the work that women do. I know this comment errs toward personal opinion – I have no concrete data to back it up, so if you have good reason to disagree that's fine, but it might be worth considering further.

Line 56: While people are obviously social animals I think adding this phrase “which scientists surely are” is a little strange as it implies scientists are under similar selection pressures as social animals. Most of the evidence you provide in the next sentences suggests that the advantages of collaborating are more related to status than they are to learning, so I think the similarity might not be as strong as implied.

Line 71: I don't think you really look at “social strategies”. I would reword to reflect more accurately what you did.

Line 85: I thought this in a few places throughout the manuscript, but this is the first place. At what stage of the focal authors career did you “measure” sociality. I think from reading further into the manuscript you did this across the authors entire career and that you did a check to see whether just looking at early career data changed things. I think it is worth making this clear earlier

Line 103: Ok so I had been wondering about this... I think it would be worth saying briefly how you do this here.

Line 118-119: This sentence assumes that everyone reaches PI status if they want to... I'm not sure this is the case. I know plenty of people who have tried to obtain PI status and not succeeded.

Line 126: Do you mean gender or sex here?

Line 166: I'm not sure you mean “further compounded by” here. That would mean that gender made the differences even bigger. I think what you mean is that the difference in number of papers that women publish compounded/generated gender differences in these metrics (depending on the metric in question)

Line 177: I got confused about this when reading the supplement because in the figure axes you refer to these measures as degrees, strength and clustering. I would define the terms collaborativeness, consistency and connectedness as early as possible and then use these terms throughout.

Line 185-189: Excuse me if I'm being dumb, but I've read this sentence 3 times and can't see the difference between your definition for consistency and connectdeness.

Line 194: In your introduction you kind of equate career progression in academia with fitness and survival, which made me wonder here whether your definition of a PI (ie three papers as last author) is really a very good definition of longterm success, this also stood out when you mentioned that some people became PI in the same year they first published. How are single author papers treat here? What if someone had a last author paper every 10 years. What if the focal author was on a fellowship and then went back to being a postdoc? I understand that figuring out when people got permanent jobs or made tenure etc might be possible for your dataset, and I think that this measure is probably reasonable, but perhaps it is possible to validate it further, or provide further acknowledgement that your measure does not necessarily implice someone has obtained a permanent job.

Line 223: I'm surprised by this, I thought behavioural ecology was much more incestuous than that □

Line 246: You control for number of papers but I think behaviour could change with career progression in a way that is not related to number of papers, so when in the career were these metrics calculated? This still isn't clear yet.

Line 246: Why run individual models for each of the social networks, and not put them all in one model? Are they correlated? I know in the supplement you show correlations for these prior to controlling for number of papers, but what about afterwards?

Line 257: Do you mean if someone didn't become a PI they were essentially considered as never having been born? what about people who continued to publish throughout the career but never achieved PI status... or what about people who had 3 last author papers but they were all really far apart? Do they still count as PIs?

Line 265: were longer gaps mostly attributed to women? I know you have this in your supplement, but I think its worth also stating here, since this might be expected as a result of women being more likely to take time out due to caring responsibilities. Given that 20% (which seems like a considerable amount) had a great than 2 year gap and looking at figure esm9a I would have thought 4 years would have been a better choice. Although given you show the results are comparable across all time frames, I don't suppose it's a big deal.

Line 277: add "for" between published and a

Line 279: Why did you use 10 years here? I think you said above that the median time till becoming a PI was 8 years, so would that be a better cutoff?

Line 293: I know you discuss other papers that do this, but I wondered whether it was possible to look at the sex ratio of coauthors with your data and whether it differs between the sexes ie is their sexual segregation and are focal authors that collaborate more with males more likely to do well in their careers? And does that differ depending on the sex of the focal author.

Line 306: How do you know they identify as a male?

Line 317: again I wondered how these measures correlate with each other, if they are highly correlated and you are conducting separate models for each, they might not really be saying anything different...

Line 366-367: should it be and/or? They didn't have to do all these things to become more likely to become a PI.

Line 370: this made me wonder whether people who are more collaborative early on are also more collaborative later? Does this differ for men and women? I feel like the things driving collaboration are different early and late in your career - early on it might be driven by confidence or having a specific skill, whereas later in your career it might be driven by how many students you take on. Likewise I guess early behaviour can go on to shape later behaviour and also drive long lasting changes in perception...

Line 372-377: I'm not sure I get the logic here

Line 379-383: You say you found differences in career productivity, but also that women had shorter careers. If you correct for time in academia do you still see differences in productivity?

Line 388-389: I think this is almost certainly true. I think women are more likely to be questioned about whether the work presented was theirs or not. However, I'm not sure its who the collaborators are but rather the number that matters - potentially having multiple different collaborators is important, not necessarily the identity of those collaborators. I wonder if it is possible to tease this out with your analysis by comparing the association between variance in male or female success with you metrics.

Line 391: I would argue that authorship full stop is allocated to the detriment of women eg men more likely to be put on papers for doing less and that PIs should be encouraged to reflect on how they allocate authorship and whether it is equitable.

Figure 1B: interesting although I suppose not surprising that once women get to 15-20year career length the rate of decline appears to be equal for men and women

Figure 2: I find these figures difficult to interpret, maybe add info on how to interpret to the legend. eg the further the lines are apart the stronger the effects??? Is it possible to use response surface type figures or contour plots instead?

Table 1: Instead of the shading a more standard approach for highlighting significant results would be to bold the significant ones or highlight them with asterisks

Supplement line 132: Is the degree supposed to be 4? I only see 4 unique coauthors? I does this number include the focal author?

Supplement line 161: delete “to those of” I think in the previous part of the sentence the subject is the authors, not the papers of the authors

Supplement Line 164: are these metrics highly correlated once you control for the number of papers? Also can you give them the same labels as what you use in the main paper

Supplement line 168: are consistency and connectedness the same as strength and clustering respectively, would be better to keep terms consistent throughout and with paper

Referee: 2

Comments to the Author(s)

I found this manuscript interesting to think about. I appreciate the authors have already put in a considerable amount of work (and might be disappointed by the length of this review). Although substantial, all my concerns can be addressed in a revision. Below I raise three concerns that will require the most effort to address, followed by quicker fixes. First, I have misgivings about how gender differences in productivity were controlled for. In addition to some statistical concerns, I question the validity of the current approach, given that gender differences exist not only in the quantity of published papers, but also the prevalence of author positions. Second, I disagree with the manuscript’s current ‘pitch’ (relating sociality and survival to authorship networks and academic careers), and recommend the authors re-write these aspects of the article with more focus on meta/social-science. Third, I question the interpretation of three aspects of the study: the definition of ‘principal investigator’, claims of directionality of effects, and implications for reducing the gender gap in science.

#--MAJOR COMMENTS--#

****1. Accounting for gender differences in productivity****

There are well-established gender differences in publication outputs, which inevitably create gender differences in co-author networks. Therefore, interpreting gender differences in co-author networks requires careful consideration of how that ‘productivity gap’ is controlled for. I am concerned that in its current form, the manuscript does not adequately explain and justify the chosen methods. While there is likely no ‘perfect’ solution, it would be worthwhile to test whether alternative analysis have a qualitative impact on the findings.

Currently, the authors have analysed three types of residual statistics, taken from generalized linear regressions where the covariate is the number of papers per author:

- (1) ‘Collaborativeness’ are the z-scaled residuals taken from a Poisson regression, where the response is the number of unique co-authors per author.
- (2) ‘Connectedness’ are the z-scaled residuals from a logistic regression, where the ‘successes’ are the number of ‘triangles’, and the ‘failures’ are the number of triangles divided by the global clustering coefficient.
- (3) ‘Consistency’ are the residuals from a model of two z-scaled residuals: the number of unique co-authors per author (‘collaborativeness’, above) is the covariate (fixed effect), and the response variable comes from the Gaussian model of the natural logarithm of $1 +$ the number of papers divided by the total number of co-authors (tie strength).

Details of these analyses in the methods section (both main text and supplement) are sparse. For example, error structures and the $\ln(x+1)$ transformations are not specified (and even within the analysis script, there is no justification for why $\ln(x+1)$ was used, rather than $\ln(x)$, given the data made available do not show any zeroes in x)

I am not a statistician (and I hope the editor invites one to review this manuscript), but intuitively I suspect analysing these types of residuals could be problematic. Other than the seeming difficult

of interpreting effect sizes from these models, I've thought of the following potential concerns (listed in no particular order):

- (1) errors in the residual estimates are not propagated to later analyses. Analysing the posterior distribution of residuals from Bayesian models could be a solution. Or just including number of papers as a covariate in the main models, rather than extracting the residuals?
- (2) the authors do not report any indices of model fit (e.g. R2 values), and residuals from poorly-fit regression lines could lead to biased conclusions. E.g. see "Forstmeier, W. (2011), 'Women have Relatively Larger Brains than Men: A Comment on the Misuse of General Linear Models in the Study of Sexual Dimorphism', *The Anatomical Record*, 294:1856-1863. <http://doi.org/10.1002/ar.21423>".
- (3) Is there something strange about the 'Consistency' residuals, where residuals from closely related metrics (Fig. ESm6.1; $r = 0.95$) are regressed against each other? Do multiple levels of correlation cause statistical issues here? If not, how does one interpret one "unit" of this resulting response variable?

Perhaps more importantly, I've thought about a potential problem with correcting for the total number of publications, without considering the focal author's position on each publication.

Currently, the residual correction assumes that X number of publications is the same thing for female and males. But as the authors mention in the discussion, female and male authors differ in both the magnitude and types of publications. For example, women might be more likely to be middle authors (and have carried out the research, rather than 'leading' it).

See the following references:

West, J.D., Jacquet, J., King, M.M., Correll, S.J., Bergstrom, C.T. (2013). The Role of Gender in Scholarly Authorship. *PLOS One*, 8(7): e66212. <https://doi.org/10.1371/journal.pone.0066212>

Macaluso, B., Larivière, V., Sugimoto, T., Sugimoto, C.R. (2016). Is Science Built on the Shoulders of Women? A Study of Gender Differences in Contributorship. *Academic Medicine*, 91(8) p 1136-1142. <http://doi.org/10.1097/ACM.0000000000001261>

(But this effect could differ with career stage, with male PhD students more likely to be credited with publications:

David, F.F., Peugh, J., Maher, M.A., Roksa, J., Tofel-Grehl, C. (2017). Time-to-Credit Gender Inequities of First-Year PhD Students in the Biological Sciences. *CBE – Life Sciences Education*, 16:1. <http://doi.org/10.1187/cbe.16-08-0237>)

Imagine that for a given number of papers, female authors have a higher proportion of middle author papers. We can assume that, compared to first and senior-author papers, middle author papers contribute less to facilitating future research opportunities (i.e. collaborations), career longevity, and seniority. We know that the dataset contains more male than female authors, therefore the modelled relationship between the number of publications, and co-author network indices, will tend to reflect the relationship between the 'male' proportion of middle-authored papers. Therefore, for the same number of papers, the female estimate of the 'impact' of that productivity on her career might be below the regression line (resulting in a negative residual). Indeed, in the available dataset, the average residual value for the number of unique co-authors, and publications-per-co-authors, is negative for females and positive for males, whereas the pattern for the clustering coefficient ('cliqueyness?') is reversed.

All this is to say, given the hidden gender differences in the 'prestige' of publications, I think it is simplistic to correct for the number of publications alone. I recommend the authors distinguish between first, middle, and last-author publications in their analyses. It appears this information

has already been extracted in section 3 of 'Social authors code_part1.Rmd', although these data are not made available for subsequent analyses.

2. Article pitch

The manuscript equates persistence in scientific publishing with 'survival', and co-authorship collaborations with sociality. To me, this pitch – which is set up in the second paragraph of the introduction – is an overreach. (A more minor point: in the opening line of that paragraph, which is in the context of non-human animals, perhaps 'gender' should be changed to 'sex').

Case in point: Lines 65-69 state "Furthermore, increased sociality is associated with greater ability to tolerate stress[12,19]. If a larger social network helps researchers to better cope with the stresses of a career in science and academia[20], then we might also expect a positive correlation between career lifespan and sociality, with less sociable individuals leaving science earlier." Breaking this down, "increased sociality is associated with reduced stress" cites "Social relationships and mortality risk: a meta-analytic review" (<http://doi.org/10.1371/journal.pmed.1000316>; reporting social relationships correlate with longer human lifespans), and "Social buffering: relief from stress and anxiety" (<http://doi.org/10.1098/rstb.2006.1941>; where social buffering is defined as "when conspecific animals are together, they show a better recovery from aversive experiences"). I do not think the types of social relationships described by these papers are equivalent to the social relationships in a co-author network. Researchers frequently publish with people they have never even met. Quite simply, co-authorship does not evoke the type of interpersonal relationship that buffers people from the stresses and anxieties of life. It is entirely conceivable that there is an inverse relationship between co-author network scores and 'sociality' (as traditionally measured), whereby academics who devote more time to co-authorship have less time to spare for maintaining their personal friendships and familial relationships. It therefore does not follow that "If a larger social network helps researchers to better cope with the stresses of a career in science and academia[20], then we might also expect a positive correlation between career lifespan and sociality, with less sociable individuals leaving science earlier." (also, citation 20 is not clearly relevant – "Staff Wellbeing in Higher Education" is a report based on 25 staff members at higher education institutes, not specific to the sciences).

I recommend the authors replace 'sociality' with a word or phrase that better describes what is actually measured in this study (e.g. 'co-author network indices'). Alternatively, the metrics could simply be referred to separately (as is done intermittently as 'collaborativeness', 'consistency' and 'connectedness', although I think 'cliquey', as used by citation 22, is more descriptive?). Less importantly, the use of 'publishing career length' instead of the snappier 'survival' has the advantage of not implying the mortal failure of ceasing to publish (e.g. citation 45). In general, the manuscript would be improved by focusing on what the co-author network metrics specifically imply and confer (e.g. nepotism, Matthew effects), and less on the nebulous links to the non-human literature, such as the associations between reproductive success and sociality.

3. Study interpretation

~~Definition of 'Principal investigator'~~

I was surprised by the definition of a Principal investigator given at lines 192-193: "following the definitions in[44], we classified any focal author with at least three last-author publications as being a PI".

I vaguely consider the status of 'Principal investigator' to confer more career stability than simply being the last author on three publications (e.g. a post-doc might publish with students in the laboratory they work for, attaining last-author publications without attaining the 'power' inferred by 'PI'). Indeed, the Wikipedia page for 'Principal investigator' says that, in Canada and the US, PI "is also often used as a synonym for "head of the laboratory" or "research group leader."" (PI

is also defined this way in <https://doi.org/10.1371/journal.pcbi.1007448>). The citation given (44: van Dijk et al. 2014, <http://doi.org/10.1016/j.cub.2014.04.039>), does not define 'PI' in the main text, and the supplemental methods suggest that this definition was chosen arbitrarily and for convenience, without being validated ("To ensure a robust estimation of whether someone becomes PI, we consider as becoming PI only those authors that have at least three last author publications and measure the time to PI as the time between that person's first publication and the time of the second last-author publication. We note that our results are highly robust to variations on this method, such as using the first two or four last author publications.").

Therefore, the current justification for the definition of PI amounts to "someone else did it this way". To continue using 'PI' throughout the manuscript, I think the authors should validate whether having three last-author publications is strongly associated with being a lab leader. For this historical dataset this task seems, at best, tedious. An easier approach would be to simply replace the use of 'PI' with 'at least three senior-author publications'. While less elegant, this definition won't mislead readers about what the study actually measured. I think this point is important, given the aforementioned research on male authors being more likely to secure that last-author position. The authors could alternatively use the time taken to publish one senior-author paper (and simply define this as "time to senior author").

~~Assessing directionality of the effects of sociality~~

The methods paragraph from lines 273-281 says that directionality was assessed by seeing "whether a focal authors' sociality in their early career predicted their future academic survival", where 'early career' is defined as the first ten years of publishing, for the cohort of researchers who had not published long before attending ISBE. The discussion (lines 369-371) says "Importantly, the extent of sociality during a focal authors' early career predicted their future survival, suggesting that establishing collaborate behaviour early drives career success, rather than the other way around." I don't follow this logic. Early career researchers who express traits that are linked to success (e.g. having a specialised and useful skill, being a good networker, etc) could foster more collaborations, but that does not mean that those collaborations are what caused their career success (i.e. the unmeasured researcher traits could be causing the indirect correlation between the network indices and career success).

~~'Fixing' the gender gap~~

There is a repeated message throughout the manuscript that the study's results can help address the gender gap in science. I don't think this conclusion is warranted; the authors describe an underlying pattern, but do not provide evidence of the causes. There could be something systematic that prevents women from collaborating more, in which case an individual approach (suggesting women seek out more collaborations) is a bit like suggesting women long-jump athletes, to close the pay gap, simply jump as far as their male counterparts.

Specific examples of this messaging are:

Last sentence of abstract: "Encouraging researchers at all levels to invest in collaborations, particularly with female researchers, will help to close the gender gap in science and academia."

Introduction, Lines 72-75: "Understanding the connections between sociality and career progression and duration, and particularly how this might differ between genders, could shed light on strategies that researchers may wish to follow, and provide greater understanding of the underlying causes of gender disparities in the sciences."

Discussion, lines 413-44 and 417-418 "Our findings suggest that for scientists early in their career, seeking out collaborations has long term-benefits", and "all researchers, regardless of gender, could benefit from seeking out collaborations early on in their career."

Concluding statements, lines 436-441: "In conclusion, our results suggest that all researchers – but particularly those that are women – can enhance their career progression and survival in science by collaborating widely and repeatedly. Creating research environments that encourage

collaboration – across disciplines and institutes, among career levels, and especially between genders – will lead to greater and more rapid scientific advances[3], and could assist in reducing the gender gap in science.”

#--MINOR COMMENTS--#

****Computational reproducibility****

I thank the authors for making analysis scripts available, in addition to processed data, but have some simple suggestions for improvements.

~~Data availability~~

Pre-processed data has not been made available, with the justification that “For purposes of confidentiality, author names and Scopus Author Identifiers (SAI) will not be made available.” Is this protection of author identity necessary, given that SAI’s only contain publicly available information? If it is, then perhaps the authors could include a hypothetical dataset of SAI’s (e.g. with their own identifiers), in the same style as the real dataset, so that the code ‘Social authors code_part1.Rmd’ can still be run (currently, it is hard to really tell how the script works, without having real data to play with).

~~Reproducibility of code~~

In ‘Social authors code_part1.Rmd’, section 8 currently calls the wrong columns. Columns 17 and 21 are being renamed, but only 13 columns are presented in the available dataset. To make this code more robust, the column names could be called directly with their names, rather than relying on their number (e.g. `names(new.corr)[names(new.corr) == "indegree"] <- "degree"`).

In ‘Social authors code_part2.Rmd’, the wrong file name was used for importing the data (i.e. doesn’t match the name of the data file that was uploaded).

For plotting results figures, the numbers are manually entered. To minimise mistakes, it would be best to call them directly from the model (e.g. ‘plotM6a’ numbers were slightly different to what was manually entered). For Figure 4 it was not clear which models were used to extract the plotted numbers. It was also not clear in the code which models were used for the results tables.

~~Missing version numbers for R packages~~

‘rscopus’ (line 128)

‘bibliometrix’ (line 132)

‘asnipe’, ‘igraph’, ‘itergraph’, and ‘network’ (line 140)

‘lmvar’ (line 183)

****Wording of the abstract****

In the opening sentence of the abstract - “Intense competition for limited opportunities means the career path of a scientist is a challenging one, and female scientists in particular are less likely to survive in academia.” - the use of ‘in particular’ implies that gender is the biggest inequity in career length, rather than being one of many factors (e.g. racial identity, socioeconomic status).

“We built authorship social networks from publication records to test how sociality predicts career progression and survival in biologists contributing to three international conferences in the 1990s”. Had I only read the abstract, I’d think the study had taken three different conferences, covering different aspects of biology, in the 1990s, rather than three iterations of the same conference. A more accurate summary would be something like “To test how sociality predicted career progression, we built authorship social networks (corrected for productivity) from publication records of behavioral ecologists who had attended a meeting of the International Society of Behavioral Ecology in 1992, 1994, or 1996.”

Abstract line 37, “publishing with many diverse co-authors”, Discussion lines 368: “Being more collaborative with a diverse set of co-authors”. Many readers will associate ‘Diverse’ with

diversity initiatives, e.g. gender, race, field of study, socio-economic background. Perhaps better to use “unique” or “distinct”, to convey that the co-authors are different people, rather than the more ambiguous “diverse”.

****Other comments****

Lines 62-63: “articles mentioned on social media gain more citations[16]”. I think this statement is premature, given the limited evidence on this topic. The paper cited as [16], “Tweeting birds: online mentions predict future citations in ornithology” (<http://doi.org/10.1098/rsos.171371>) looks at only a subset of the scientific literature, and tests for a correlation between Altmetric scores (which includes both traditional and social media) and citation counts from the year 2014. Even if there is a robust correlation between social media attention and citation counts, this does not prove that social media attention increases citations (e.g. more citable papers could be more tweetable – see also <https://doi.org/10.1371/journal.pone.0183551>).

Line 87-88:

“Our cohort consisted of contributors to three consecutive conference of the International Society for Behavioral Ecology in the 1990s” □ I misunderstood this to mean that the cohort was people who went to three consecutive ISBEs, not people who had been to any of the three

Line 92-93:

“gender bias in publishing is unlikely to be an issue[31,32] (although see[29])” □ this claim is too broad to be supported by the given evidence. ‘Gender bias in publishing’ could refer to any stage in the publication process, from manuscript conception to appearing in a journal. Citations 31(<https://doi.org/10.1371/journal.pone.0201725>) and 32 (<https://doi.org/10.1016/j.biocon.2009.06.021>) do not find gender bias in the editorial decisions at two journals, but this does not discount other types of biases (e.g. citation 31 suggests women are under-represented as corresponding authors).

Misplaced parenthesis lines 169-170: “published with co-authors that were less connected lower (global clustering coefficient; Fog.ESM6.2C). □ ‘lower’ should be inside the parenthesis

In the main text, the same description – “published more often with the same co-authors” – is given for interpreting the ‘consistency score’ and ‘connectedness score’ (lines 185-189). I think this is a typo, because the two indices must have slightly different interpretations? I found the supplementary Figure ESM5 helpful for understanding the indices, and think it could be usefully included in the main text.

Discussion, Line 306: “Identifying as male” □ given that gender was inferred from “first name and/or online profile”, rather than asking authors to identify their gender, perhaps this should instead be “Being identified as male”

Discussion, Lines 382-386: “The shorter careers and higher dropout rates of female scientists explain a large proportion of their reduced productivity and impact[7]. However, these factors likely also restrict opportunities for enhancing sociality by establishing new collaborations, thereby further contributing to less favourable career outcomes for female scientists.” What is the connection between these two sentences? (And aren’t “shorter careers” and “higher dropout rates” the same thing?) The second sentence says “these factors” (i.e. having a shorter career) reduce opportunities for collaborations, but this is trivial: women who stop publishing do not form new collaborations. That pattern should not affect the outcome of this study, though, given that the analysis tried to control for the number of publications (which would be correlated with career length)?

Discussion, lines 411-413: “Junior scientists are typically pursuers of collaborations, while senior researchers are net attractors, having new collaborations proposed to them[59]” □ Citation 59 offers this interpretation for the pattern that the duration of collaborations decreases with career

age (“Quantifying the impact of weak, strong, and super ties in scientific careers”: <https://doi.org/10.1073/pnas.1501444112>), but this seems simplistic. Senior researchers can easily propose opportunistic collaborations with ERCs who have the time or motivation for the time-consuming aspect of the project (e.g. students of collaborators, visitors), resulting in the same pattern without those senior researchers being ‘pursued’.

Alternative explanation for why “the effects of author sociality on career progression and length were consistently stronger for women than for men” (Discussion, lines 386-388) □ time use data from around the world consistently show that women have less free time than men. Stronger co-author networks, which can keep balls in the air when the focal author drops the ball, could therefore be more important for women, who simply cannot attend to those balls as often (e.g. look at productivity data during COVID-19).

Referee: 3

Comments to the Author(s)

This paper explores collaboration differences between men and women authors relate to PI status and career length. While the question is of great interest to a broad interdisciplinary scientific community, several questions with the chosen methodology suggest a revision is necessary before publication in Royal Society B.

~ I would like to see Fig ESM3 broken down by gender (say using a stacked box plot) and with the appropriate log-binning (Fig ESM3 A,C are heavy-tailed distributions).

~ many effects are driven by the heavy-tailed nature of publications and citations – thus, unless there is reason to suspect name disambiguation errors, the 6 focal authors with more than 400 papers should be included. Were any of these women?

~ since the number of publications displays strong gender differences, the correction procedure is very important. It looks like linear and normality assumptions were used to build the adjusted measures of sociality. Please provide justification that the residuals are normally distributed, and this standardization procedure did not actually exacerbate the gender differences in productivity.

~ the definition of PI as time to 2 last-author papers is consistent with other measures of seniority and the disciplinary norms of this field.

~ The authors attempt to control for female author name changes is applauded. However, in practice, we have found that contemporary male authors are also likely to change their last names (typically hyphenated), and the rate of real author name changes is much smaller than the rate of mis-identified authors, last-name spelling differences (particularly with latinized eastern-european names and east-asian names), name order errors (last & first names flipped), missing prefixes or suffices, and missing punctuation – all of which contribute to noise in the reconstruction of publication careers.

~ are the survival results consistent if the dependent variable is switched to publication sequence from real-world time?

~ A similar analysis was conducted in:

Jadidi et al (2018) Gender disparities in science? Dropout, Productivity, Collaborations and Success of Male and Female Computer Scientists. *Advances in Complex Systems*. 21. 1750011

Could the authors put their findings in the context of these results?

~ Can the authors offer a more specific interpretation for these findings in terms of ecology field norms and practices?

Author's Response to Decision Letter for (RSPB-2020-1024.R0)

See Appendix A.

RSPB-2021-0219.R0

Review form: Reviewer 2

Recommendation

Accept with minor revision (please list in comments)

Scientific importance: Is the manuscript an original and important contribution to its field?

Good

General interest: Is the paper of sufficient general interest?

Good

Quality of the paper: Is the overall quality of the paper suitable?

Good

Is the length of the paper justified?

Yes

Should the paper be seen by a specialist statistical reviewer?

Yes

Do you have any concerns about statistical analyses in this paper? If so, please specify them explicitly in your report.

Yes

It is a condition of publication that authors make their supporting data, code and materials available - either as supplementary material or hosted in an external repository. Please rate, if applicable, the supporting data on the following criteria.

Is it accessible?

Yes

Is it clear?

Yes

Is it adequate?

Yes

Do you have any ethical concerns with this paper?

No

Comments to the Author

I appreciate the author's responses to the reviewer comments, and think the revised manuscript is much improved.

I still think it would be worthwhile having a specialist statistical reviewer comment on the analysis methods. I didn't quite understand the justification for keeping the residual analyses (which do not propagate error, and can lead to erroneous conclusions when model fit is poor), rather than controlling for productivity by including number of papers as a covariate. The authors replied "Number of publications could not be entered into survival models as a covariate or an offset because it was significantly correlated with all metrics". But isn't the correlation the thing we're trying to control? Again, this is not my specialty so I might be missing something obvious.

Otherwise, my comments are minor and can be easily addressed:

Abstract line 29-32: recommend revising or splitting this long sentence.

Abstract line 32-33: "women were less likely, and slower, to become a principal investigator" and line 40-41 "increased likelihood and reduced time to become a PI". One of my concerns about the original manuscript was the unvalidated measure of 'principal investigator' being three last author publications. The authors have chosen to stick with the term PI throughout the manuscript. I would still like the abstract to define how this was measured, because many readers won't look into the details. For example: "women were less likely, and slower, to become a principal investigator (approximated by having three last-author publications)".

Introduction, lines 53-55. The first sentence of the introduction is a bit hard to read. Recommend splitting into two after 'challenge', or adding a comma after 'insecurity'.

Introduction, line 177: "if collaboration behaviour impacts on careers" ... Should be "impacts careers" or "has an impact on careers".

Introduction, line 199: "published with more collaborators (i.e. unique co-authors), multiple times" ... stray comma

Methods, lines 208-211: "...any focal author with at least three last-author publications was classified as being a PI, and ii) we approximated the time taken ('time to become a PI') as the difference in years between their first publication and their second last-author (multi-authored) publication (i.e. requiring at least three publications)". If a PI requires three last-author publications, shouldn't the time taken to reach PI be the time between their first publication and *third* last-author paper, not the second?

Results, lines 328-332: "female focal authors had significantly fewer co-authors than expected (adjusted network size; Fig.1D), published with the same co-authors less frequently than expected (adjusted tie strength; Fig.1E), and had more connected co-authors than expected (adjusted clustering coefficient; Fig.1F) when compared to male co-authors." The use of 'than expected' in this sentence reinforces the 'male default'. I recommend moving "when compared to male co-authors" to the start of the sentence and deleting the three "than expected"s.

Results, line 387: "publishing with the same co-authors more frequently significantly reduced career length". Is this meant to say 'increased' rather than 'reduced'? If not, the results are a bit confusing, because lines 399-401 say "Publishing with more unique co-authors, with the same co-authors more frequently, and with less connected co-authors in early career all predicted a longer career overall."

Discussion, lines 418-419: "Compared to men, women were also more likely to be a PI, and to become one more quickly, if they published repeatedly with their co-authors." Recommend re-writing this sentence. I think the most likely interpretation is "women who publish repeatedly with their co-authors become a PI more than men", but instead I think it's trying to say that the effect of repeated collaborations is higher for women than men.

Discussion line 494: “comparable positive effects” ... comparably?

Discussion line 502: “Researchers can enhance their careers in academic science by collaborating widely and repeatedly, and this is likely to be of even greater benefit for women” ... Up until this concluding paragraph of the revised manuscript, the authors have been fairly careful not to overreach the causal implications of the data. I recommend toning this down.

Review form: Reviewer 4

Recommendation

Major revision is needed (please make suggestions in comments)

Scientific importance: Is the manuscript an original and important contribution to its field?

Good

General interest: Is the paper of sufficient general interest?

Excellent

Quality of the paper: Is the overall quality of the paper suitable?

Acceptable

Is the length of the paper justified?

Yes

Should the paper be seen by a specialist statistical reviewer?

Yes

Do you have any concerns about statistical analyses in this paper? If so, please specify them explicitly in your report.

Yes

It is a condition of publication that authors make their supporting data, code and materials available - either as supplementary material or hosted in an external repository. Please rate, if applicable, the supporting data on the following criteria.

Is it accessible?

N/A

Is it clear?

N/A

Is it adequate?

N/A

Do you have any ethical concerns with this paper?

No

Comments to the Author

I have attached a file called: van der wal et al.pdf

Decision letter (RSPB-2021-0219.R0)

25-Mar-2021

Dear Dr van der Wal:

Your manuscript has now been peer reviewed and the reviews have been assessed by an Associate Editor. The reviewers' comments (not including confidential comments to the Editor) and the comments from the Associate Editor are included at the end of this email for your reference. As you will see, the reviewers and the Editors have raised some concerns with your manuscript and we would like to invite you to revise your manuscript to address them.

Research ethics:

Use of animals and field studies:

It is a condition of publication that you make available the data and research materials supporting the results in the article (<https://royalsociety.org/journals/authors/author-guidelines/#data>). Datasets should be deposited in an appropriate publicly available repository and details of the associated accession number, link or DOI to the datasets must be included in the Data Accessibility section of the article (<https://royalsociety.org/journals/ethics->

policies/data-sharing-mining/). Reference(s) to datasets should also be included in the reference list of the article with DOIs (where available).

Please submit a copy of your revised paper within three weeks. If we do not hear from you within this time your manuscript will be rejected. If you are unable to meet this deadline please let us know as soon as possible, as we may be able to grant a short extension.

Best wishes,
Dr Maurine Neiman
mailto: proceedingsb@royalsociety.org

Associate Editor Board Member

Comments to Author:

Many thanks for sending us your carefully revised manuscript, which is a substantial improvement over your first submission. In particular, you have done a good job in reframing your analysis (now more in relation to other collaboration analyses). However, there are still some important issues remaining, in particular concerning the statistical analysis and interpretations of causality. We have invited a special reviewer (reviewer #2) to have a look at the statistical analysis. This reviewer really had a careful look at your analysis and made a lot of important suggestions that you should take very serious when revising your manuscript. In particular, please entirely remove the residuals analysis as it can lead to erroneous conclusions with your data (this was also seen as potentially very problematic by reviewer #1). Furthermore, you should be more careful when describing your results: you should frame them more in terms of statistical relationships and patterns rather than making (potentially erroneous) conclusions about causality. This is also an issue that is pointed out (again) by reviewer # 1; in particular for the last discussion section. The data, even without any over-interpretation, are interesting enough

to publish. I also agree that it would be premature to generally conclude that women should collaborate more to be more successful in academia. And as reviewer #1 suggested, please be upfront with how you identify a PI already in the abstract. Finally, also please have a look at the other issues that the two reviewers have raised. I do hope that you will not be too disappointed by this review as I realize you have done a lot of work with the resubmitted manuscript already – but as reviewer # 2 also pointed out, no new analyses will be needed and it is mostly a matter of rephrasing/ toning down your results and the interpretation. I am looking forward to receive your revised paper in due time.

Reviewer(s)' Comments to Author:

Referee: 2

Comments to the Author(s).

I appreciate the author's responses to the reviewer comments, and think the revised manuscript is much improved.

I still think it would be worthwhile having a specialist statistical reviewer comment on the analysis methods. I didn't quite understand the justification for keeping the residual analyses (which do not propagate error, and can lead to erroneous conclusions when model fit is poor), rather than controlling for productivity by including number of papers as a covariate. The authors replied "Number of publications could not be entered into survival models as a covariate or an offset because it was significantly correlated with all metrics". But isn't the correlation the thing we're trying to control? Again, this is not my specialty so I might be missing something obvious.

Otherwise, my comments are minor and can be easily addressed:

Abstract line 29-32: recommend revising or splitting this long sentence.

Abstract line 32-33: "women were less likely, and slower, to become a principal investigator" and line 40-41 "increased likelihood and reduced time to become a PI". One of my concerns about the original manuscript was the unvalidated measure of 'principal investigator' being three last author publications. The authors have chosen to stick with the term PI throughout the manuscript. I would still like the abstract to define how this was measured, because many readers won't look into the details. For example: "women were less likely, and slower, to become a principal investigator (approximated by having three last-author publications)".

Introduction, lines 53-55. The first sentence of the introduction is a bit hard to read. Recommend splitting into two after 'challenge', or adding a comma after 'insecurity'.

Introduction, line 177: "if collaboration behaviour impacts on careers" ... Should be "impacts careers" or "has an impact on careers".

Introduction, line 199: "published with more collaborators (i.e. unique co-authors), multiple times" ... stray comma

Methods, lines 208-211: "...any focal author with at least three last-author publications was classified as being a PI, and ii) we approximated the time taken ('time to become a PI') as the difference in years between their first publication and their second last-author (multi-authored) publication (i.e. requiring at least three publications)". If a PI requires three last-author publications, shouldn't the time taken to reach PI be the time between their first publication and *third* last-author paper, not the second?

Results, lines 328-332: "female focal authors had significantly fewer co-authors than expected (adjusted network size; Fig.1D), published with the same co-authors less frequently than expected (adjusted tie strength; Fig.1E), and had more connected co-authors than expected (adjusted clustering coefficient; Fig.1F) when compared to male co-authors." The use of 'than

expected' in this sentence reinforces the 'male default'. I recommend moving "when compared to male co-authors" to the start of the sentence and deleting the three "than expected"s.

Results, line 387: "publishing with the same co-authors more frequently significantly reduced career length". Is this meant to say 'increased' rather than 'reduced'? If not, the results are a bit confusing, because lines 399-401 say "Publishing with more unique co-authors, with the same co-authors more frequently, and with less connected co-authors in early career all predicted a longer career overall."

Discussion, lines 418-419: "Compared to men, women were also more likely to be a PI, and to become one more quickly, if they published repeatedly with their co-authors." Recommend re-writing this sentence. I think the most likely interpretation is "women who publish repeatedly with their co-authors become a PI more than men", but instead I think it's trying to say that the effect of repeated collaborations is higher for women than men.

Discussion line 494: "comparable positive effects" ... comparably?

Discussion line 502: "Researchers can enhance their careers in academic science by collaborating widely and repeatedly, and this is likely to be of even greater benefit for women" ... Up until this concluding paragraph of the revised manuscript, the authors have been fairly careful not to overreach the causal implications of the data. I recommend toning this down.

Referee: 4

Comments to the Author(s).

I have attached a file called: van der wal et al.pdf

Author's Response to Decision Letter for (RSPB-2021-0219.R0)

See Appendix B.

RSPB-2021-0219.R1 (Revision)

Review form: Reviewer 2

Recommendation

Accept with minor revision (please list in comments)

Scientific importance: Is the manuscript an original and important contribution to its field?

Good

General interest: Is the paper of sufficient general interest?

Good

Quality of the paper: Is the overall quality of the paper suitable?

Good

Is the length of the paper justified?

Yes

Should the paper be seen by a specialist statistical reviewer?

Yes

Do you have any concerns about statistical analyses in this paper? If so, please specify them explicitly in your report.

No

It is a condition of publication that authors make their supporting data, code and materials available - either as supplementary material or hosted in an external repository. Please rate, if applicable, the supporting data on the following criteria.

Is it accessible?

Yes

Is it clear?

No

Is it adequate?

No

Do you have any ethical concerns with this paper?

No

Comments to the Author

I thank the authors for their responses to the reviewer comments, and to the editor for finding a statistical reviewer.

I still find the definition of PI and time taken to reach PI odd. I now understand that authors were classified as being a PI when they had three last-author publications, but the "time to reach PI status" was taken as the time to have two last-author publications. This is not clear in the abstract, which says (lines 31-34) "Among this cohort, women were slower and less likely to become a principal investigator (PI; approximated by having three last-author publications), and published fewer papers over fewer years (i.e. had shorter academic careers) than men." Shouldn't it instead say "Among this cohort, women were less likely to become a principal investigator (PI; approximated by having three last-author publications), and were slower to do so (approximated by the time taken to have two last-author publications)"? Or, the analyses could be re-done so that time taken to reach PI is congruent with the stated definition of a PI.

On the same topic, lines 313-317 read "Following the methods of[48], we approximated the 'time to become a PI' as the difference in years between a focal author's first publication and their second last-author (multi-authored) publication. Last authorship position usually denotes senior leadership of an independent project in our field[49] and this approach is robust to variations such as using time to publish two or four last-author publications (e.g. SI in[41])." Shouldn't this say "robust to variations such as using time to publish three or four last-author publications", given that time to publish two is presented as the main analysis?

Another semantic quibble: "The likelihood that a female focal author left science after becoming a PI was higher than for men". This could be slightly changed to "left academic science", in recognition that some of those female authors might have left academic publishing but still be conducting scientific research elsewhere (e.g. industry, government).

Also, please make all code available on the OSF (including for the supporting information - currently it says "The computational scripts used for these Supplements are available upon request").

Finally, I recommend the authors do a careful proof-read before submitting any minor revisions. (I really sympathise with the tedium of doing so on a manuscript that has been so long in the making! I find the MS word “read aloud” feature helpful for this).

For example, some small things introduced by tracked changes:

“Overall, authors with stronger and less clustered networks were more likely to become a PI, did so more quickly” - missing “and”

“Persisting in academic science a major challenge” - missing “is”

“By shaping funding, citation rates, and reputation, collaboration behaviour could have considerable impact on career progression in science” - not sure but I think this is missing “a”, or else “could considerably”

“Collaboration patterns, however, differ between genders: female researchers generally have smaller[20] (but see[21]) and more clustered networks[22], publish fewer papers as ‘high prestige’ authors (single-author, or in first or last authorship positions)[23], and typically publish fewer papers than their male counterparts [10,20].” - superfluous “typically”, given there is already a “generally” in this sentence

“Therefore, we conducted a logitudinalanalysis to investigate whether a” - typo and missing space

“Thereofor, we did not include authorship position in further analyses.” -- typo

“Unfortunately, there is little published work on how name changes affect indexing services and citation accuracy (but see[51]) so checked whether author name changes could have generated false negative records of leaving science.” - missing ‘we’

Overly confusing sentence:

“In i), event = being a PI, and 64 of 772 focal authors (8%; 50% of whom were women) who were still in science at the end of the study period yet did not become a PI during this time, were censored. In ii), event = cessation of publishing, and 642 of 935 focal authors (69%; 27% of whom were women) who published within two years of the end of the study period were censored as still active.”

Stray parenthesis:

Collaborative working has similarly positive effects on scientific success in computer science[22]), and for cell biologists and physicists[16],

Review form: Reviewer 4

Recommendation

Major revision is needed (please make suggestions in comments)

Please rate the overall quality of this paper:

Good

Is it accessible?

N/A

Is it clear?

N/A

Is it adequate?

N/A

Do you have any concerns about statistical analyses in this paper? If so, please specify them explicitly in your report.

Yes

It is a condition of publication that authors make their supporting data, code and materials available - either as supplementary material or hosted in an external repository. Please rate, if applicable, the supporting data on the following criteria.

Comments to the Author

I am attaching a file with my comments to the authors

Decision letter (RSPB-2021-0219.R1)

29-Jun-2021

Dear Dr van der Wal:

Your manuscript has now been peer reviewed and the reviews have been assessed by an Associate Editor. The reviewers' comments (not including confidential comments to the Editor) and the comments from the Associate Editor are included at the end of this email for your reference. As you will see, the reviewers and the Editors have raised some concerns with your manuscript and we would like to invite you to revise your manuscript to address them.

Research ethics:

Use of animals and field studies:

It is a condition of publication that you make available the data and research materials supporting the results in the article (<https://royalsociety.org/journals/authors/author-guidelines/#data>). Datasets should be deposited in an appropriate publicly available repository and details of the associated accession number, link or DOI to the datasets must be included in the Data Accessibility section of the article (<https://royalsociety.org/journals/ethics-policies/data-sharing-mining/>). Reference(s) to datasets should also be included in the reference list of the article with DOIs (where available).

Please submit a copy of your revised paper within three weeks. If we do not hear from you within this time your manuscript will be rejected. If you are unable to meet this deadline please let us know as soon as possible, as we may be able to grant a short extension.

Best wishes,
 Dr Maurine Neiman
 Editor, Proceedings B
 mailto: proceedingsb@royalsociety.org

Associate Editor
 Board Member: 1
 Comments to Author:

Many thanks for resubmitting your manuscript, which has now been re-evaluated by the two previous reviewers. While both reviewers think the manuscript to be valuable, there are still some issues that need to be carefully addressed before potential publication. I believe a revision will be relatively straightforward because both reviewers provide clear guidance.

Reviewer(s)' Comments to Author:

Referee: 2

Comments to the Author(s)

I thank the authors for their responses to the reviewer comments, and to the editor for finding a statistical reviewer.

I still find the definition of PI and time taken to reach PI odd. I now understand that authors were classified as being a PI when they had three last-author publications, but the "time to reach PI status" was taken as the time to have two last-author publications. This is not clear in the abstract, which says (lines 31-34) "Among this cohort, women were slower and less likely to become a principal investigator (PI; approximated by having three last-author publications), and published fewer papers over fewer years (i.e. had shorter academic careers) than men." Shouldn't it instead say "Among this cohort, women were less likely to become a principal investigator (PI; approximated by having three last-author publications), and were slower to do so (approximated by the time taken to have two last-author publications)"? Or, the analyses could be re-done so that time taken to reach PI is congruent with the stated definition of a PI.

On the same topic, lines 313-317 read "Following the methods of[48], we approximated the 'time to become a PI' as the difference in years between a focal author's first publication and their second last-author (multi-authored) publication. Last authorship position usually denotes senior leadership of an independent project in our field[49] and this approach is robust to variations such as using time to publish two or four last-author publications (e.g. SI in[41])." Shouldn't this say "robust to variations such as using time to publish three or four last-author publications", given that time to publish two is presented as the main analysis?

Another semantic quibble: "The likelihood that a female focal author left science after becoming a PI was higher than for men". This could be slightly changed to "left academic science", in recognition that some of those female authors might have left academic publishing but still be conducting scientific research elsewhere (e.g. industry, government).

Also, please make all code available on the OSF (including for the supporting information - currently it says "The computational scripts used for these Supplements are available upon request.").

Finally, I recommend the authors do a careful proof-read before submitting any minor revisions. (I really sympathise with the tedium of doing so on a manuscript that has been so long in the making! I find the MS word "read aloud" feature helpful for this).

For example, some small things introduced by tracked changes:

"Overall, authors with stronger and less clustered networks were more likely to become a PI, did so more quickly" - missing "and"

“Persisting in academic science a major challenge” – missing “is”

“By shaping funding, citation rates, and reputation, collaboration behaviour could have considerable impact on career progression in science” – not sure but I think this is missing “a”, or else “could considerably”

“Collaboration patterns, however, differ between genders: female researchers generally have smaller[20] (but see[21]) and more clustered networks[22], publish fewer papers as ‘high prestige’ authors (single-author, or in first or last authorship positions)[23], and typically publish fewer papers than their male counterparts [10,20].” – superfluous “typically”, given there is already a “generally” in this sentence

“Therefore, we conducted a logitudinalanalysis to investigate whether a” – typo and missing space

“Therefor, we did not include authorship position in further analyses.” -- typo

“Unfortunately, there is little published work on how name changes affect indexing services and citation accuracy (but see[51]) so checked whether author name changes could have generated false negative records of leaving science.” – missing ‘we’

Overly confusing sentence:

“In i), event = being a PI, and 64 of 772 focal authors (8%; 50% of whom were women) who were still in science at the end of the study period yet did not become a PI during this time, were censored. In ii), event = cessation of publishing, and 642 of 935 focal authors (69%; 27% of whom were women) who published within two years of the end of the study period were censored as still active.”

Stray parenthesis:

Collaborative working has similarly positive effects on scientific success in computer science[22]), and for cell biologists and physicists[16],

Referee: 4

Comments to the Author(s)

I am attaching a file with my comments to the authors

Author's Response to Decision Letter for (RSPB-2021-0219.R1)

See Appendix C.

Decision letter (RSPB-2021-0219.R2)

13-Aug-2021

Dear Dr van der Wal

I am pleased to inform you that your manuscript entitled "Collaboration enhances career progression in academic science, especially for female researchers" has been accepted for publication in Proceedings B.

Data Accessibility section

Open Access

Paper charges

Sincerely,

Dr Maurine Neiman

Associate Editor:

Board Member: 1

Comments to Author:

I think you have done a great job to revise your paper. I think it has improved majorly in the process, congratulations!

Appendix A

Associate Editor

Board Member: 1

Comments to Author:

I have received three reviews for your manuscript entitled “Sociality enhances survival in science, especially for female researchers”. Two reviewers thought your paper was overall interesting and the analysis generally well done. Reviewer #3 pointed out that the methodology is not entirely novel: another study published in 2018 already investigated the same issue using a similar methodology, albeit studying a different researcher community (computer scientists). I would recommend adding the mentioned paper (Jadidi et al) to your reference list and relate to this study in a revision. Your study, I believe, will nevertheless still be interesting for the Proc B readership (even if indeed methodologically not entirely novel) because it investigates data from within the biology researcher community. That said, please also make more obvious in your abstract that the data is drawn from a single conference but different years – this is indeed confusing now. I also hope that you may find inspiration from the Jadidi paper as well as the reviewers’ comments for how to attend to several remaining statistical issues (in particular the correction procedure) and issues of interpretation (in particular concerning causality). Those issues really need to be convincingly addressed and resolved for the paper to be considered for publication in Proc B. All three reviewers also questioned your methods to identify PI’s – please also take more care here. Finally, two reviewers saw problems with the conceptual embedding of the paper: they did not really see the need for a close connection between the concepts of scientific collaboration and (biological) sociality. Reviewer #2 made several good suggestions for how to reframe your study. I also whole-heartedly agree that in the end, what we should strive for is not collaboration per se. Instead, as reviewer # 1 so nicely put it, “I would like to see conducting good science as the motivation”. I hope you find the reviewers’ suggestions helpful to revise your manuscript, and I would be looking forward to see your paper in a revised form. In agreement with at least two of the reviewers, I think it holds a lot of potential.

Dear Editor,

Thank you for your positive response to our paper, and for the very comprehensive and thoughtful reviews that your referees have provided. We appreciate the time and effort they have taken in reading and critiquing our manuscript and these have been extremely helpful.

We have responded to all the reviewers’ comments, but in particular we have now included the Jadidi paper, addressed the issues surrounding the correction procedure, and made our explanation of interpretation and causality clearer. We have amended the abstract to make it clear that the data is drawn from a single conference but different years, and have provided more information on our methods for identifying PIs. We have also reframed the conceptual basis of the paper: while we still acknowledge that our study takes methodological inspiration from studies of non-human animals’ sociality and survival, we have toned down links between the concepts of scientific collaboration and biological sociality. Finally, we have removed previous references to collaboration as a strategy that researchers may wish to pursue for academic survival and focus on explaining why the patterns we detect between co-authorship and careers might arise.

Reviewer(s)’ Comments to Author:

Referee: 1

Comments to the Author(s)

Although I can’t say I enjoyed reading this paper. I do think it was a very interesting and well executed study. It is actually a study I have personally thought of conducting, but without experience in social network analysis had never gotten around to it. So I am glad that someone finally has. In their study the authors conduct an ego centric social network analysis using data obtained from authorship of scientific papers within the field of behavioural ecology. The authors delineate their study group by using only people that were listed as participants in conferences run by the International Society for behavioural ecology in the 1990’s - although this of course limits the conclusions the authors can make from their study, it is a clever way to ensure a manageable dataset while simultaneously ensuring they obtained data from active scientists and that these scientist would have had time to see the career progression/decline that the authors are interested in.

I hope that my comments, help to improve the manuscript.

Thank you for your very useful comments on our manuscript.

Major comments:

1. To be honest, I'm not sure how I feel about the use of the term sociality in the context of this paper – particularly in the abstract where you have not had a chance to explain your definition of the term. Equating tendency to collaborate with being more social is I think problematic. While I can see why this is appealing in terms of “selling” the paper, I think it is a little misleading for several reasons. One, I think people can be very social but not very collaborative – do you have any data to suggest that more social people are more collaborative? Two I think even if I were translating collaboration into an animal context I don't think collaboration and sociality are interchangeable – I would put collaboration more in the realms of alliances or cooperating to achieve a common goal, even at a species level I'm not convinced these are related necessarily related to sociality. This is something that I think needs to be changed throughout the manuscript, and which may change the tone of the introduction considerably.

This point highlights a key issue in the literature - what is 'sociality'? The definition of this term varies according to field e.g. Ashley Ward and Mike Webster provide a succinct history of this in the introduction to their recent book 'Sociality: the behaviour of group living' (Springer, 2018). In behavioural ecology (following Tinbergen 1951), sociality typically refers to the tendency of animals to be in proximity of others when expressing the behaviour of interest (Ward & Webster, p. 2). Therefore, collaboration is a measure of sociality in terms of scientific publishing behaviour. To avoid ambiguity, however, we now refer to 'collaboration behaviour' or 'collaboration patterns' rather than 'sociality', and we have substantially toned down links between the concepts of scientific collaboration and biological sociality. Instead, we now focus on how our cohort approach and use of survival analysis takes inspiration from studies of sociality and survival in human and non-human animals (Introduction).

2. While I agree with the conclusion that encouraging people to invest in collaborations with women will help to address the gender gap in science, I'm not sure I agree with the motivation – ie collaborate to aid career progression. One day I would like to see conducting good science as the motivation. Perhaps the end of the abstract could be rephrased at the moment it sounds a little like benefits career progression is more important than the science, which I'm sure isn't the intention.

Thank you for pointing this out. We have reworded the last sentence of the abstract to make our intention clearer: "Our results suggest, therefore, that large and varied collaboration networks improve career progression, especially for women."

3. I am also not convinced that women collaborate less than men – but rather suspect that part of the problem is that women's contributions don't receive the same level of recognition. I have seen this over and over again, particularly for ECR's, so perhaps it's not just about investing in these collaborations but also recognising the work that women do. I know this comment errs toward personal opinion – I have no concrete data to back it up, so if you have good reason to disagree that's fine, but it might be worth considering further.

This is an important point. Co-authorship is a measurable output of collaboration, but there are many other facets to collaboration (e.g. see ref.9 in the manuscript). We now make it clear in the manuscript that we are using co-authorship as an imperfect, yet established, proxy, and that women's contributions may not receive the same level of recognition.

Line 56: While people are obviously social animals I think adding this phrase “which scientists surely are” is a little strange as it implies scientists are under similar selection pressures as social animals. Most of the evidence you provide in the next sentences suggests that the advantages of collaborating are more related to status than they are to learning, so I think the similarity might not be as strong as implied.

We have removed this phrase from the text.

Line 71: I don't think you really look at “social strategies”. I would reword to reflect more accurately what you did.

Changed to: “collaboration patterns”

Line 85: I thought this in a few places throughout the manuscript, but this is the first place. At what stage of the focal authors career did you “measure” sociality. I think from reading further into the manuscript you did this across the authors entire career and that you did a check to see whether just looking at early career data changed things. I think it is worth making this clear earlier

Thank you. We have clarified that collaboration behaviour was calculated across academic careers in science.

Line 103: Ok so I had been wondering about this... I think it would be worth saying briefly how you do this here.

Thank you. We have now stated this more clearly as: "A positive correlation between authors' collaboration behaviour and career length might arise because long-established or senior researchers are more attractive to collaborate with (the Matthew effect[39]), or have simply had more time to establish successful and enduring collaborations. Therefore, we conducted a final analysis to determine whether a focal authors' propensity to collaborate in their early career predicted overall academic career length."

Line 118-119: This sentence assumes that everyone reaches PI status if they want to... I'm not sure this is the case. I know plenty of people who have tried to obtain PI status and not succeeded.

Thank you for pointing this out. Our intention was not to imply that all researchers reach PI status if they want to. We have now changed the wording to reflect that not everyone is successful in obtaining PI status: "We chose these three conferences because participants starting their career in the early 1990s should by now have reached PI status, should they have wanted to and been successful in doing so."

Line 126: Do you mean gender or sex here?

We mean gender here. We inferred the gender of our focal authors from their name and/or online profile, since we have no information about their biological sex.

Line 166: I'm not sure you mean "further compounded by" here. That would mean that gender made the differences even bigger. I think what you mean is that the difference in number of papers that women publish compounded/generated gender differences in these metrics (depending on the metric in question)

Yes, you are correct. The text has been rewritten to better reflect that it is the gender differences in publication output that generated gender differences in our metrics of collaboration behaviour.

Line 177: I got confused about this when reading the supplement because in the figure axes you refer to these measures as degrees, strength and clustering. I would define the terms collaborativeness, consistency and connectedness as early as possible and then use these terms throughout.

We have changed how we present our metrics to be more transparent. Degree (now called network size), tie strength and the global clustering coefficient are the 'raw measures' derived directly from the ego networks. We then adjusted these measures for the number of publications, to get our final collaboration behaviour metrics. To avoid confusion and ensure full transparency, we now refer to these final measures as 'adjusted network size' (previously, 'collaborativeness') 'adjusted tie strength' (previously 'consistency' and for which we now use a more intuitive and established social network metric), and 'adjusted clustering coefficient' (previously co-author connectedness).

Line 185-189: Excuse me if I'm being dumb, but I've read this sentence 3 times and can't see the difference between your definition for consistency and connectedness.

Following reviewers' comments, we have chosen an alternative measure to describe the consistency of a focal author's collaborations: adjusted tie strength, and have revised this section substantially.

Line 194: In your introduction you kind of equate career progression in academia with fitness and survival, which made me wonder here whether your definition of a PI (ie three papers as last author) is really a very good definition of longterm success, this also stood out when you mentioned that some people became PI in the same year they first published. How are single author papers treat here? What if someone had a last author paper every 10 years. What if the focal author was on a fellowship and then went back to being a postdoc? I understand that figuring out when people got permanent jobs or made tenure etc might be possible for your dataset, and I think that this measure is probably reasonable, but perhaps it is possible to validate it further, or provide further acknowledgement that your measure does not necessarily implicate someone has obtained a permanent job.

Thank you for your comment. We now acknowledge in the text that this definition of a PI is an approximation based on existing studies using publication records (e.g. van Dijk et al., Current Biology 2014, cited 136 times), but that it is not perfect and nor does it imply job security. There will be exceptions, but in general this measure seems to be robust to different criteria (see Methods). We now provide more details about how we estimated when an author 'became a PI' and also include evaluations of PI status exceptions on our results: e.g. 18 focal authors became PI in the same year as their first publication (i.e. time to PI was 0 years), and so were assigned a dummy value of 0.5, since survival models ignore zero values. Re-running the survival analyses with these authors excluded did not qualitatively alter results.

Line 223: I'm surprised by this, I thought behavioural ecology was much more incestuous than that.

No comment!

Line 246: You control for number of papers but I think behaviour could change with career progression in a way that is not related to number of papers, so when in the career were these metrics calculated? This still isn't clear yet.

We have clarified that egocentric social networks were calculated across the entire career by adding the underlined section in the following sentence: "We constructed egocentric social networks[44] based on each focal author's full list of publications and co-authors (see ESM4 for details)."

Line 246: Why run individual models for each of the social networks, and not put them all in one model? Are they correlated? I know in the supplement you show correlations for these prior to controlling for number of papers, but what about afterwards?

It is common practice in social network analyses to analyse several different measures, even if correlated with one another, as they may reveal different information (e.g. Webber, Schneider and Vander Wal 2020 Animal Behaviour). In our case, some of the collaboration behaviour metrics are significantly, albeit weakly, correlated (adjusted network size and adjusted tie strength: $r = 0.15$, $p < 0.001$, adjusted network size and adjusted clustering coefficient: $r = 0.22$, $p < 0.001$) while adjusted tie strength and adjusted clustering coefficient are not (but there was a trend: $r = 0.06$, $p = 0.06$). We have expanded the correlation matrix in Figure ESM5.1 to include the collaboration behaviour metrics after adjusting for number of publications, and the statistics mentioned here, to explain why we could not include them all in one model.

Line 257: Do you mean if someone didn't become a PI they were essentially considered as never having been born? what about people who continued to publish throughout the career but never achieved PI status... or what about people who had 3 last author papers but they were all really far apart? Do they still count as PIs?

A survival analysis concerns time-to-event data, where the event in this case is becoming a PI (i.e. at least 3 last-author papers), and the time is quantified as the interval between first publication year and the publication year of the second last-author paper. Individuals that did not experience this event (i.e. that did not publish at least three last-author papers) before the end of our sampling period are censored, but they are still considered (as being 'born') in the dataset. The average gap between first and last multi-author papers for PIs (mean \pm SE = 3.14 ± 0.12 , median = 2, range = 0 – 21 years) was very similar to the gap for all focal authors (mean \pm SE = 3.36 ± 0.13 , median = 2, range = 0 – 25). Excluding 35 focal authors (13 women, 22 men) classified as PIs according to our definition but with a publication gap between their first and second last-author papers of longer than 11 years (mean + 1 SD = $5.4 + 5.9$) did not qualitatively change results. Therefore, we choose not to apply any cut-off on how far apart these PI-qualifying papers were. These evaluations are now reported in the Methods.

Line 265: were longer gaps mostly attributed to women? I know you have this in your supplement, but I think its worth also stating here, since this might be expected as a result of women being more likely to take time out due to caring responsibilities. Given that 20% (which seems like a considerable amount) had a great than 2 year gap and looking at figure esm9a I would have thought 4 years would have been a better choice. Although given you show the results are comparable across all time frames, I don't suppose it's a big deal.

Thank you, we added the following sentence to the Results: "In total 293 focal authors stopped publishing; 41% of women ($n = 123$) and 27% of men ($n = 170$)." The proportion of women assumed to have stopped publishing when using a more conservative censoring approach (still publishing within six years from the end of our sampling period, respectively, changed very little: 41% (compared to 42% at the 2-year cut-off) (see ESM8). We have also adapted Figure ESM8 so that values are specified per gender, in a stacked histogram.

Line 277: add "for" between published and a

Added.

Line 279: Why did you use 10 years here? I think you said above that the median time till becoming a PI was 8 years, so would that be a better cutoff?

Definitions of 'early-career' vary widely. Milojevic PNAS 2018 (ref.1) determines 'Junior level career dropout' to end at 10 years. We have made it clearer in the text that we follow Milojevic PNAS 2018 in our paper. Mean time to become a PI for women was 10.6 years, and 8.8 years for men, and so we feel that a value of 10 years provides an appropriate cut-off point for the end of the early career period in our dataset.

Line 293: I know you discuss other papers that do this, but I wondered whether it was possible to look at the sex ratio of coauthors with your data and whether it differs between the sexes ie is their sexual

segregation and are focal authors that collaborate more with males more likely to do well in their careers? And does that differ depending on the sex of the focal author.

Thank you for this suggestion. Unfortunately, it is not feasible for us to investigate sex ratio of co-authors: we inferred the gender (wo/man) of conference contributors from their first name given in the conference booklets, and/or online profile, but Scopus only provides the initials of co-authors and detecting their gender is beyond the scope of our study (n = 43,803 non-focal authors).

Line 306: How do you know they identify as a male?

You are right; we do not know this. We have changed this to “The likelihood that a focal author remained in science was also lower for women [...]”.

Line 317: again I wondered how these measures correlate with each other, if they are highly correlated and you are conducting separate models for each, they might not really be saying anything different...

Please see our response to your comment on L246.

Line 366-367: should it be and/or? They didn't have to do all these things to become more likely to become a PI.

We have changed the discussion substantially and this comment is no longer relevant.

Line 370: this made me wonder whether people who are more collaborative early on are also more collaborative later? Does this differ for men and women? I feel like the things driving collaboration are different early and late in your career – early on it might be driven by confidence or having a specific skill, whereas later in your career it might be driven by how many students you take on. Likewise I guess early behaviour can go on to shape later behaviour and also drive long lasting changes in perception...

Yes, these are very interesting questions. We considered looking at whether authors that are more collaborative (have a higher adjusted network size) early on are also more collaborative later on, but this is difficult to do with our dataset because of different career lengths and different time periods when careers started (with different collaboration norms). For a fair comparison the social network metrics would need to be calculated over the same period. It would have been possible to do this for authors with career lengths of at least 20 years, but then we would be introducing a bias due to the gender differences in career length that we identified. Addressing these questions therefore did not seem appropriate with our dataset.

Line 372-377: I'm not sure I get the logic here

We have substantially changed the discussion and this sentence no longer exists.

Line 379-383: You say you found differences in career productivity, but also that women had shorter careers. If you correct for time in academia do you still see differences in productivity?

Yes, if we correct for number of publication years as an offset in the model, we still find a difference in career productivity between genders. This has now been added to the Results: “[...] and this result held even when correcting for career length as an offset in the model (GLM with offset, n = 935 focal authors, estimate ± SE = -0.38 ± 0.01, z = -35.50, p < 0.001).”

Line 388-389: I think this is almost certainly true. I think women are more likely to be questioned about whether the work presented was theirs or not. However, I'm not sure its who the collaborators are but rather the number that matters - potentially having multiple different collaborators is important, not necessarily the identity of those collaborators. I wonder if it is possible to tease this out with your analysis by comparing the association between variance in male or female success with you metrics.

Unfortunately, we are unclear what the suggested analysis would look like here. To be able to assess variance between male and female focal authors, we would need multiple measures per individual, and we only have single values for each subject.

Line 391: I would argue that authorship full stop is allocated to the detriment of women eg men more likely to be put on papers for doing less and that PIs should be encouraged to reflect on how they allocate authorship and whether it is equitable.

We have changed the discussion a lot, and the relevant sentence now reads: “Women are often judged more than men on the basis of whom they work with[39], and while authorship positions assigned to women often undervalue their contributions to research, having a female senior author on a manuscript increases both the overall proportion of female authors, and the probability of a female first author[62].”

Figure 1B: interesting although I suppose not suprising that once women get to 15-20year career length the rate of decline appears to be equal for men and women

Thank you for making this valid point.

Figure 2: I find these figures difficult to interpret, maybe add info on how to interpret to the legend. eg the further the lines are apart the stronger the effects???. Is it possible to use response surface type figures or contour plots instead?

Survival plots are the most standard way to present results from survival analyses, however we agree that these were difficult to interpret. Therefore, we have now moved what were Figures 2 and 3 to the Electronic Supplementary Material, and provide a more intuitive set of Figures to accompany the main text.

Table 1: Instead of the shading a more standard approach for highlighting significant results would be to bold the significant ones or highlight them with asterisks

Thank you. We have changed the tables and legends accordingly.

Supplement line 132: Is the degree supposed to be 4? I only see 4 unique coauthors? I does this number include the focal author?

Yes, thank you for spotting this mistake. We have now corrected it.

Supplement line 161: delete "to those of" I think in the previous part of the sentence the subject is the authors, not the papers of the authors

Corrected.

Supplement Line 164: are these metrics highly correlated once you control for the number of papers? Also can you give them the same labels as what you use in the main paper

Our response to your comment on Line 246 describes how we have addressed this.

Supplement line 168: are consistency and connectedness the same as strength and clustering respectively, would be better to keep terms consistent throughout and with paper

We now refer to the terms in a more transparent manner. See our response to your comment on Line 177.

Referee: 2

Comments to the Author(s)

I found this manuscript interesting to think about. I appreciate the authors have already put in a considerable amount of work (and might be disappointed by the length of this review). Although substantial, all my concerns can be addressed in a revision. Below I raise three concerns that will require the most effort to address, followed by quicker fixes. First, I have misgivings about how gender differences in productivity were controlled for. In addition to some statistical concerns, I question the validity of the current approach, given that gender differences exist not only in the quantity of published papers, but also the prevalence of author positions. Second, I disagree with the manuscript's current 'pitch' (relating sociality and survival to authorship networks and academic careers), and recommend the authors re-write these aspects of the article with more focus on meta/social-science. Third, I question the interpretation of three aspects of the study: the definition of 'principal investigator', claims of directionality of effects, and implications for reducing the gender gap in science.

We were happy to hear that you thought it was interesting! Thank you for providing detailed feedback on our manuscript, and for providing great suggestions for addressing your concerns. These are very much appreciated!

#--MAJOR COMMENTS--#

****1. Accounting for gender differences in productivity****

There are well-established gender differences in publication outputs, which inevitably create gender differences in co-author networks. Therefore, interpreting gender differences in co-author networks requires careful consideration of how that 'productivity gap' is controlled for. I am concerned that in its current form, the manuscript does not adequately explain and justify the chosen methods. While there is likely no 'perfect' solution, it would be worthwhile to test whether alternative analysis have a qualitative impact on the findings.

Currently, the authors have analysed three types of residual statistics, taken from generalized linear regressions where the covariate is the number of papers per author:

(1) 'Collaborativeness' are the z-scaled residuals taken from a Poisson regression, where the response is the number of unique co-authors per author.

(2) 'Connectedness' are the z-scaled residuals from a logistic regression, where the 'successes' are the number of 'triangles', and the 'failures' are the number of triangles divided by the global clustering coefficient.

(3) 'Consistency' are the residuals from a model of two z-scaled residuals: the number of unique co-authors per author ('collaborativeness', above) is the covariate (fixed effect), and the response variable comes from the Gaussian model of the natural logarithm of 1 + the number of papers divided by the total number of co-authors (tie strength).

Details of these analyses in the methods section (both main text and supplement) are sparse. For example, error structures and the $\ln(x+1)$ transformations are not specified (and even within the analysis script, there is no justification for why $\ln(x+1)$ was used, rather than $\ln(x)$, given the data made available do not show any zeroes in x)

We have now added more details on the GLM error structures we used in the text. The $\ln(x+1)$ transformation is no longer included.

I am not a statistician (and I hope the editor invites one to review this manuscript), but intuitively I suspect analysing these types of residuals could be problematic. Other than the seeming difficult of interpreting effect sizes from these models, I've thought of the following potential concerns (listed in no particular order):

(1) errors in the residual estimates are not propagated to later analyses. Analysing the posterior distribution of residuals from Bayesian models could be a solution. Or just including number of papers as a covariate in the main models, rather than extracting the residuals?

Number of publications could not be entered into survival models as a covariate or an offset because it was significantly correlated with all metrics (all $p < 0.001$, network size: $r = 0.87$, tie strength: $r = 0.39$, clustering coefficient: $r = 0.63$). Instead, we decided to extract the residuals to obtain adjusted collaboration behaviour measures: adjusted network size, adjusted tie strength and adjusted clustering coefficient, which were used in further analyses. We have rewritten the Methods substantially to clarify why and how we have done this.

(2) the authors do not report any indices of model fit (e.g. R^2 values), and residuals from poorly-fit regression lines could lead to biased conclusions. E.g. see “Forstmeier, W. (2011), ‘Women have Relatively Larger Brains than Men: A Comment on the Misuse of General Linear Models in the Study of Sexual Dimorphism’, *The Anatomical Record*, 294:1856-1863. <http://doi.org/10.1002/ar.21423>”.

For the residual models, we now give McFadden Pseudo- R^2 as indications of good model fit in the ESM.

(3) Is there something strange about the ‘Consistency’ residuals, where residuals from closely related metrics (Fig. ESm6.1; $r = 0.95$) are regressed against each other? Do multiple levels of correlation cause statistical issues here? If not, how does one interpret one “unit” of this resulting response variable?

Thank you for this comment – we have replaced this measure with ‘tie strength’, a standard social network metric.

Perhaps more importantly, I’ve thought about a potential problem with correcting for the total number of publications, without considering the focal author’s position on each publication.

Currently, the residual correction assumes that X number of publications is the same thing for female and males. But as the authors mention in the discussion, female and male authors differ in both the magnitude and types of publications. For example, women might be more likely to be middle authors (and have carried out the research, rather than ‘leading’ it).

See the following references:

West, J.D., Jacquet, J., King, M.M., Correll, S.J., Bergstrom, C.T. (2013). The Role of Gender in Scholarly Authorship. *PLOS One*, 8(7): e66212. <https://doi.org/10.1371/journal.pone.0066212>

Macaluso, B., Larivière, V., Sugimoto, T., Sugimoto, C.R. (2016). Is Science Built on the Shoulders of Women? A Study of Gender Differences in Contributorship. *Academic Medicine*, 91(8) p 1136-1142. <http://doi.org/10.1097/ACM.0000000000001261>

(But this effect could differ with career stage, with male PhD students more likely to be credited with publications:

David, F.F., Peugh, J., Maher, M.A., Roksa, J., Tofel-Grehl, C. (2017). Time-to-Credit Gender Inequities of First-Year PhD Students in the Biological Sciences. *CBE—Life Sciences Education*, 16:1. <http://doi.org/10.1187/cbe.16-08-0237>)

Imagine that for a given number of papers, female authors have a higher proportion of middle author papers. We can assume that, compared to first and senior-author papers, middle author papers contribute less to facilitating future research opportunities (i.e. collaborations), career longevity, and seniority. We know that the dataset contains more male than female authors, therefore the modelled relationship between the number of publications, and co-author network indices, will tend to reflect the relationship between the ‘male’ proportion of middle-authored papers. Therefore, for the same number of papers, the female estimate of the ‘impact’ of that productivity on her career might be below the regression line (resulting in a negative residual). Indeed, in the available dataset, the average residual value for the number of unique co-authors, and publications-per-co-authors, is negative for females and positive for males, whereas the pattern for the clustering coefficient (‘cliqueyness’?) is reversed.

All this is to say, given the hidden gender differences in the ‘prestige’ of publications, I think it is simplistic to correct for the number of publications alone. I recommend the authors distinguish between first, middle, and last-author publications in their analyses. It appears this information has already been extracted in section 3 of ‘Social authors code_part1.Rmd’, although these data are not made available for subsequent analyses.

Thank you for your great suggestion to consider the focal author's position on each publication. Your clear explanation helped us to realise that this is indeed crucial (see for example Lerchenmueller & Sorenson 2018 Research Policy). We have added the following text:

“As in many fields of life science[24], first and last authorship in ecology and evolutionary biology publications represent the highest prestige positions[48]. If author position in our dataset varies by gender[49], then it could bias interpretations about collaboration behaviour and career progression. Therefore, we checked whether female and male focal authors differed in their proportion of either first or last author papers, after accounting for first publication year, given that number of authors on publications has increased with time[16] but we found no difference (prestige authorship position: 62% of female authors' multi-authored papers, 63% of male authors' multi-authored papers; generalised linear model with a binomial error distribution, $n = 935$ focal authors, gender: estimate \pm SE = 0.02 ± 0.02 , $z = 0.91$, $p = 0.36$; first publication year: estimate \pm SE = -0.02 ± 0.002 , $z = -8.87$, $p < 0.001$).”

Given that there was no significant difference between genders in the proportion of prestige papers published, we have chosen not to change our main analyses to take account of authorship position. However, we do now account for gender differences in productivity when calculating the adjusted metrics by including an interaction between number of papers and gender.

****2. Article pitch****

The manuscript equates persistence in scientific publishing with 'survival', and co-authorship collaborations with sociality. To me, this pitch – which is set up in the second paragraph of the introduction – is an overreach. (A more minor point: in the opening line of that paragraph, which is in the context of non-human animals, perhaps 'gender' should be changed to 'sex').

Case in point: Lines 65-69 state “Furthermore, increased sociality is associated with greater ability to tolerate stress[12,19]. If a larger social network helps researchers to better cope with the stresses of a career in science and academia[20], then we might also expect a positive correlation between career lifespan and sociality, with less sociable individuals leaving science earlier.” Breaking this down, “increased sociality is associated with reduced stress” cites “Social relationships and mortality risk: a meta-analytic review” (<http://doi.org/10.1371/journal.pmed.1000316>; reporting social relationships correlate with longer human lifespans), and “Social buffering: relief from stress and anxiety” (<http://doi.org/10.1098/rstb.2006.1941>; where social buffering is defined as “when conspecific animals are together, they show a better recovery from aversive experiences”). I do not think the types of social relationships described by these papers are equivalent to the social relationships in a co-author network. Researchers frequently publish with people they have never even met. Quite simply, co-authorship does not evoke the type of interpersonal relationship that buffers people from the stresses and anxieties of life. It is entirely conceivable that there is an inverse relationship between co-author network scores and 'sociality' (as traditionally measured), whereby academics who devote more time to co-authorship have less time to spare for maintaining their personal friendships and familial relationships. It therefore does not follow that “If a larger social network helps researchers to better cope with the stresses of a career in science and academia[20], then we might also expect a positive correlation between career lifespan and sociality, with less sociable individuals leaving science earlier.” (also, citation 20 is not clearly relevant – “Staff Wellbeing in Higher Education” is a report based on 25 staff members at higher education institutes, not specific to the sciences).

I recommend the authors replace 'sociality' with a word or phrase that better describes what is actually measured in this study (e.g. 'co-author network indices'). Alternatively, the metrics could simply be referred to separately (as is done intermittently as 'collaborativeness', 'consistency' and 'connectedness', although I think 'cliquey', as used by citation 22, is more descriptive?). Less importantly, the use of 'publishing career length' instead of the snappier 'survival' has the advantage of not implying the mortal failure of ceasing to publish (e.g. citation 45). In general, the manuscript would be improved by focusing on what the co-author network metrics specifically imply and confer (e.g. nepotism, Matthew effects), and less on the nebulous links to the non-human literature, such as the associations between reproductive success and sociality.

Thank you, we have edited our manuscript substantially to focus more on existing studies of collaboration behaviour in academic publishing and less on studies of non-human animals. Please see our response to Reviewer 1's Major comment 1.

Similarly, we have replaced 'survival' with 'career progression' to avoid any confusion and draw inferences with the effects that you mentioned (e.g. Matthew effect (L124)). Thank you for your very useful feedback!

****3. Study interpretation****

~~Definition of 'Principal investigator'~~

I was surprised by the definition of a Principal investigator given at lines 192-193: "following the definitions in[44], we classified any focal author with at least three last-author publications as being a PI".

I vaguely consider the status of 'Principal investigator' to confer more career stability than simply being the last author on three publications (e.g. a post-doc might publish with students in the laboratory they work for, attaining last-author publications without attaining the 'power' inferred by 'PI'). Indeed, the Wikipedia page for 'Principal investigator' says that, in Canada and the US, PI "is also often used as a synonym for "head of the laboratory" or "research group leader."" (PI is also defined this way in <https://doi.org/10.1371/journal.pcbi.1007448>). The citation given (44: van Dijk et al. 2014, <http://doi.org/10.1016/j.cub.2014.04.039>), does not define 'PI' in the main text, and the supplemental methods suggest that this definition was chosen arbitrarily and for convenience, without being validated ("To ensure a robust estimation of whether someone becomes PI, we consider as becoming PI only those authors that have at least three last author publications and measure the time to PI as the time between that person's first publication and the time of the second last-author publication. We note that our results are highly robust to variations on this method, such as using the first two or four last author publications.").

Therefore, the current justification for the definition of PI amounts to "someone else did it this way". To continue using 'PI' throughout the manuscript, I think the authors should validate whether having three last-author publications is strongly associated with being a lab leader. For this historical dataset this task seems, at best, tedious. An easier approach would be to simply replace the use of 'PI' with 'at least three senior-author publications'. While less elegant, this definition won't mislead readers about what the study actually measured. I think this point is important, given the aforementioned research on male authors being more likely to secure that last-author position. The authors could alternatively use the time taken to publish one senior-author paper (and simply define this as "time to senior author").

Thank you for your comment. Fixing on a definition of who qualifies as a PI, and what this term exactly means is tricky, as the reviewer highlights. Indeed, in our personal experience, definitions of PI status differ between countries, and even between institutions within the same country. Nevertheless, a key step towards achieving a position of leadership in academia (i.e. being responsible for other people and your own research ideas, whether or not this is associated with a permanent job position) is being a senior author on more than one peer-reviewed paper. The van Dijk et al. definition of a PI matches with this. We now provide a better explanation of our choice of approach in the methods (L211 onwards) and acknowledge that it is by no means perfect. Please also see our response to Reviewer 1's comment regarding Line 194.

~~Assessing directionality of the effects of sociality~~

The methods paragraph from lines 273-281 says that directionality was assessed by seeing "whether a focal authors' sociality in their early career predicted their future academic survival", where 'early career' is defined as the first ten years of publishing, for the cohort of researchers who had not published long before attending ISBE. The discussion (lines 369-371) says "Importantly, the extent of sociality during a focal authors' early career predicted their future survival, suggesting that establishing collaborate behaviour early drives career success, rather than the other way around." I don't follow this logic. Early career researchers who express traits that are linked to success (e.g. having a specialised and useful skill, being a good networker, etc) could foster more collaborations, but that does not mean that those collaborations are what caused their career success (i.e. the unmeasured researcher traits could be causing the indirect correlation between the network indices and career success).

Our intention with this analysis was to explore whether collaboration behaviour (as measured by our three adjusted network metrics) early in one's career could predict later success. We agree that unmeasured researcher traits may well be linked to success and these unmeasured traits could also be linked to the ability to foster collaborations. Indeed, we did not intend to imply that the co-authorship patterns themselves are directly responsible, but rather we meant that they capture variation in our focal authors' 'behaviour'. We have substantially revised how we interpret these results in the Discussion and so hope that our intended meaning is now clear.

~~'Fixing' the gender gap~~

There is a repeated message throughout the manuscript that the study's results can help address the gender gap in science. I don't think this conclusion is warranted; the authors describe an underlying pattern, but do not provide evidence of the causes. There could be something systematic that prevents women from collaborating more, in which case an individual approach (suggesting women seek out more collaborations) is a bit like suggesting women long-jump athletes, to close the pay gap, simply jump as far as their male counterparts.

Specific examples of this messaging are:

Last sentence of abstract: "Encouraging researchers at all levels to invest in collaborations, particularly with female researchers, will help to close the gender gap in science and academia."

Introduction, Lines 72-75: "Understanding the connections between sociality and career progression and duration, and particularly how this might differ between genders, could shed light on strategies that researchers may wish to follow, and provide greater understanding of the underlying causes of gender disparities in the sciences."

Discussion, lines 413-44 and 417-418 "Our findings suggest that for scientists early in their career, seeking out collaborations has long term-benefits", and "all researchers, regardless of gender, could benefit from seeking out collaborations early on in their career."

Concluding statements, lines 436-441: "In conclusion, our results suggest that all researchers – but particularly those that are women – can enhance their career progression and survival in science by collaborating widely and repeatedly. Creating research environments that encourage collaboration – across disciplines and institutes, among career levels, and especially between genders – will lead to greater and more rapid scientific advances[3], and could assist in reducing the gender gap in science."

Thank you for your comments and the examples given. We have rewritten the Introduction and Discussion entirely with the reviewer's comments in mind. The last sentence of the discussion now reads "Creating research environments that encourage and allow for collaboration, particularly for women, should therefore become a priority."

#--MINOR COMMENTS--#

Computational reproducibility

I thank the authors for making analysis scripts available, in addition to processed data, but have some simple suggestions for improvements.

~~Data availability~~

Pre-processed data has not been made available, with the justification that "For purposes of confidentiality, author names and Scopus Author Identifiers (SAI) will not be made available." Is this protection of author identity necessary, given that SAI's only contain publicly available information? If it is, then perhaps the authors could include a hypothetical dataset of SAI's (e.g. with their own identifiers), in the same style as the real dataset, so that the code 'Social authors code_part1.Rmd' can still be run (currently, it is hard to really tell how the script works, without having real data to play with).

Thank you, we have included a hypothetical dataset of the authors' own Scopus Author Identifiers in the same style as the real dataset, so that the first part of the code can be run.

~~Reproducibility of code~~

In 'Social authors code_part1.Rmd', section 8 currently calls the wrong columns. Columns 17 and 21 are being renamed, but only 13 columns are presented in the available dataset. To make this code more robust, the column names could be called directly with their names, rather than relying on their number (e.g. `names(new.corr)[names(new.corr) == "indegree"] <- "degree"`).

Thank you! Changed, calling column with names directly as suggested.

In 'Social authors code_part2.Rmd', the wrong file name was used for importing the data (i.e. doesn't match the name of the data file that was uploaded).

In 'Social authors code_part2.Rmd', the csv file called (social.authors.data.csv) is correct and matches the data file that was uploaded:

Read in data provided with Script

new = read_csv("social.authors.data.csv")

For plotting results figures, the numbers are manually entered. To minimise mistakes, it would be best to call them directly from the model (e.g. 'plotM6a' numbers were slightly different to what was manually entered). For Figure 4 it was not clear which models were used to extract the plotted numbers. It was also not clear in the code which models were used for the results tables.

Numbers are now called automatically from the model – thanks for this suggestion.

We have also clarified which models were used to extract the plotted numbers, and which models were used for the results tables.

~~Missing version numbers for R packages~~

'rscopus' (line 128)

'bibliometrix' (line 132)

'asnipe', 'igraph', 'itergraph', and 'network' (line 140)

'lmvar' (line 183)

Done; added all version numbers in the reference list.

****Wording of the abstract****

In the opening sentence of the abstract - "Intense competition for limited opportunities means the career path of a scientist is a challenging one, and female scientists in particular are less likely to survive in academia." – the use of 'in particular' implies that gender is the biggest inequity in career length, rather than being one of many factors (e.g. racial identity, socioeconomic status).

"We built authorship social networks from publication records to test how sociality predicts career progression and survival in biologists contributing to three international conferences in the 1990s". Had I only read the abstract, I'd think the study had taken three different conferences, covering different aspects of biology, in the 1990s, rather than three iterations of the same conference. A more accurate summary would be something like "To test how sociality predicted career progression, we built authorship social networks (corrected for productivity) from publication records of behavioral ecologists who had attended a meeting of the International Society of Behavioral Ecology in 1992, 1994, or 1996."

Abstract line 37, "publishing with many diverse co-authors", Discussion lines 368: "Being more collaborative with a diverse set of co-authors". Many readers will associate 'Diverse' with diversity initiatives, e.g. gender, race, field of study, socio-economic background. Perhaps better to use "unique" or "distinct", to convey that the co-authors are different people, rather than the more ambiguous "diverse".

Thanks for these suggestions. We have completely rewritten the abstract to reflect these changes and to be more careful with our wording.

****Other comments****

Lines 62-63: "articles mentioned on social media gain more citations[16]". I think this statement is premature, given the limited evidence on this topic. The paper cited as [16], "Tweeting birds: online mentions predict future citations in ornithology" (<http://doi.org/10.1098/rsos.171371>) looks at only a subset of the scientific literature, and tests for a correlation between Altmetric scores (which includes both traditional and social media) and citation counts from the year 2014. Even if there is a robust correlation between social media attention and citation counts, this does not prove that social media attention increases citations (e.g. more citable papers could be more tweetable – see also <https://doi.org/10.1371/journal.pone.0183551>).

This section has been rewritten and is no longer relevant.

Line 87-88:

"Our cohort consisted of contributors to three consecutive conference of the International Society for Behavioral Ecology in the 1990s" ◊ I misunderstood this to mean that the cohort was people who went to three consecutive ISBEs, not people who had been to any of the three

Changed from 'Our cohort consisted of contributors to three consecutive conferences' to: 'Our cohort consists of contributors to any of three consecutive conferences'

Line 92-93:

"gender bias in publishing is unlikely to be an issue[31,32] (although see[29])" ◇ this claim is too broad to be supported by the given evidence. 'Gender bias in publishing' could refer to any stage in the publication process, from manuscript conception to appearing in a journal. Citations 31(<https://doi.org/10.1371/journal.pone.0201725>) and 32 (<https://doi.org/10.1016/j.biocon.2009.06.021>) do not find gender bias in the editorial decisions at two journals, but this does not discount other types of biases (e.g. citation 31 suggests women are under-represented as corresponding authors).

We have rephrased this to make it clear that gender bias does not apply to the likelihood of a paper appearing in print. We agree that there may be other sources of bias in the publication process (encompassing from study inception through to final manuscript acceptance and publication), but our point here is that within our field of interest, papers that do have female authors are no less likely to be published than those that have male authors.

Misplaced parenthesis lines 169-170: "published with co-authors that were less connected lower (global clustering coefficient; Fog.ESM6.2C). ◇ 'lower' should be inside the parenthesis

Thank you, done.

In the main text, the same description – "published more often with the same co-authors" – is given for interpreting the 'consistency score' and 'connectedness score' (lines 185-189). I think this is a typo, because the two indices must have slightly different interpretations? I found the supplementary Figure ESM5 helpful for understanding the indices, and think it could be usefully included in the main text.

Thank you for pointing this out, this was indeed a typo and has now been updated with the definition of a new and more intuitive consistency measure: adjusted tie strength. Unfortunately, manuscript length constraints mean that we must keep Figure ESM5 as a supplementary figure. However, we now include more comprehensive descriptions throughout the main manuscript.

Discussion, Line 306: "Identifying as male" ◇ given that gender was inferred from "first name and/or online profile", rather than asking authors to identify their gender, perhaps this should instead be "Being identified as male"

We have changed this from "Identifying as male also increased the likelihood that a focal author remained in science" to: "The likelihood that a focal author remained in science was also lower for women"

Discussion, Lines 382-386: "The shorter careers and higher dropout rates of female scientists explain a large proportion of their reduced productivity and impact[7]. However, these factors likely also restrict opportunities for enhancing sociality by establishing new collaborations, thereby further contributing to less favourable career outcomes for female scientists." What is the connection between these two sentences? (And aren't "shorter careers" and "higher dropout rates" the same thing?) The second sentence says "these factors" (i.e. having a shorter career) reduce opportunities for collaborations, but this is trivial: women who stop publishing do not form new collaborations. That pattern should not affect the outcome of this study, though, given that the analysis tried to control for the number of publications (which would be correlated with career length)?

Please see the main text for a rewording and rephrasing of this section to reflect your comments.

Discussion, lines 411-413: "Junior scientists are typically pursuers of collaborations, while senior researchers are net attractors, having new collaborations proposed to them[59]" ◇ Citation 59 offers this interpretation for the pattern that the duration of collaborations decreases with career age ("Quantifying the impact of weak, strong, and super ties in scientific careers": <https://doi.org/10.1073/pnas.1501444112>), but this seems simplistic. Senior researchers can easily propose opportunistic collaborations with ERCs who have the time or motivation for the time-consuming aspect of the project (e.g. students of collaborators, visitors), resulting in the same pattern without those senior researchers being 'pursued'.

We have removed this sentence from the manuscript.

Alternative explanation for why "the effects of author sociality on career progression and length were consistently stronger for women than for men" (Discussion, lines 386-388) ◇ time use data from around the world consistently show that women have less free time than men. Stronger co-author networks, which can keep balls in the air when the focal author drops the ball, could therefore be more important

for women, who simply cannot attend to those balls as often (e.g. look at productivity data during COVID-19).

We have added this alternative explanation to the text: "A second potential explanation is that large co-authorship networks with low connectedness may allow at least some projects to continue during periods of hardship for individual team members[66], and these benefits may be especially important for women[e.g. 64]."

Referee: 3

Comments to the Author(s)

This paper explores collaboration differences between men and women authors relate to PI status and career length. While the question is of great interest to a broad interdisciplinary scientific community, several questions with the chosen methodology suggest a revision is necessary before publication in Royal Society B.

~ I would like to see Fig ESM3 broken down by gender (say using a stacked box plot) and with the appropriate log-binning (Fig ESM3 A,C are heavy-tailed distributions).

Thanks for this suggestion; Fig ESM3 A,B are now broken down per gender to show that these restrictions did not result in a gender bias (as already explained in ESM3). Fig ESM3 A,B are on a logarithmic scale due to heavy-tailed distributions.

~ many effects are driven by the heavy-tailed nature of publications and citations— thus, unless there is reason to suspect name disambiguation errors, the 6 focal authors with more than 400 papers should be included. Were any of these women?

Thank you for this comment. As stated in ESM3, we excluded focal authors with more than 400 papers (n = 6 focal authors): one woman and five men.

~ since the number of publications displays strong gender differences, the correction procedure is very important. It looks like linear and normality assumptions were used to build the adjusted measures of sociality. Please provide justification that the residuals are normally distributed, and this standardization procedure did not actually exacerbate the gender differences in productivity.

*The models now include the interaction term: gender * no. of publications. The assumptions of normality and homoscedasticity for GLMs were assessed by visual inspection of residuals and normal probability plots where appropriate and were found to be met in all cases (now mentioned in the Methods). We also now report McFadden Pseudo-R² as indications of good model fit.*

~ the definition of PI as time to 2 last-author papers is consistent with other measures of seniority and the disciplinary norms of this field.

Please see our responses to referees 1 and 2 regarding our definition of time to PI and PI status, which, given the nebulous nature of what a PI is, and who qualifies as one, is never going to be perfect. Nonetheless, within the field of behavioural ecology, our definition is appropriate and consistent with other measures of seniority and the disciplinary norms in this field.

~ The authors attempt to control for female author name changes is applauded. However, in practice, we have found that contemporary male authors are also likely to change their last names (typically hyphenated), and the rate of real author name changes is much smaller than the rate of mis-identified authors, last-name spelling differences (particularly with latinized eastern-european names and east-asian names), name order errors (last & first names flipped), missing prefixes or suffixes, and missing punctuation — all of which contribute to noise in the reconstruction of publication careers.

We agree, using names is a challenge! During curation of the dataset we manually entered the names of all of our authors and thus hopefully we avoided noise that can arise from automated scripts. We also invested considerable effort in identifying authors with multiple SCOPUS identities, which often arose due to mis-spelled names or missing prefixes, suffixes or punctuation. Therefore, we are confident that the amount of noise in our dataset deriving from mis-identified authors is as low as possible.

~ are the survival results consistent if the dependent variable is switched to publication sequence from real-world time?

Unfortunately, we do not understand what the referee is suggesting here. We do not include the date of publication in any of our analyses, but only consider time differences between publications (e.g. when calculating career length).

~ A similar analysis was conducted in:

Jadidi et al (2018) Gender disparities in science? Dropout, Productivity, Collaborations and Success of Male and Female Computer Scientists. *Advances in Complex Systems*. 21. 1750011

Could the authors put their findings in the context of these results?

Thank you for making us aware of this excellent paper, which we now cite.

~ Can the authors offer a more specific interpretation for these findings in terms of ecology field norms and practices?

We now provide more discussion within the context of ecology and previous efforts to evaluate co-authorship patterns in the field. Please see the penultimate paragraph of the discussion.

Appendix B

Associate Editor Board Member

Comments to Author:

Many thanks for sending us your carefully revised manuscript, which is a substantial improvement over your first submission. In particular, you have done a good job in reframing your analysis (now more in relation to other collaboration analyses). However, there are still some important issues remaining, in particular concerning the statistical analysis and interpretations of causality. We have invited a special reviewer (reviewer #2) to have a look at the statistical analysis. This reviewer really had a careful look at your analysis and made a lot of important suggestions that you should take very seriously when revising your manuscript. In particular, please entirely remove the residuals analysis as it can lead to erroneous conclusions with your data (this was also seen as potentially very problematic by reviewer #1). Furthermore, you should be more careful when describing your results: you should frame them more in terms of statistical relationships and patterns rather than making (potentially erroneous) conclusions about causality. This is also an issue that is pointed out (again) by reviewer #1; in particular for the last discussion section. The data, even without any over-interpretation, are interesting enough to publish. I also agree that it would be premature to generally conclude that women should collaborate more to be more successful in academia. And as reviewer #1 suggested, please be up-front with how you identify a PI already in the abstract. Finally, also please have a look at the other issues that the two reviewers have raised. I do hope that you will not be too disappointed by this review as I realize you have done a lot of work with the resubmitted manuscript already – but as reviewer #2 also pointed out, no new analyses will be needed and it is mostly a matter of rephrasing/toning down your results and the interpretation. I am looking forward to receive your revised paper in due time.

Dear Editor,

Thank you for your positive response to the revised manuscript. We are pleased that you like the reframing of our analysis. We respond to all of the reviewers' comments in detail below, but in particular we have:

- (1) Rerun the survival analyses excluding publication output altogether, which showed similar results as our original analysis. However, we have decided to include these as supplementary material (ESM Supplement 1).*
- (2) After much thought and further reading, we would like to retain our original approach using residual regression. Publication number is an important covariate that we feel would raise questions in readers' minds if it was not included in our models. However, because it is highly correlated with our predictors of interest ('predictor' in the sense of statistical modelling terminology), it cannot be included directly. We now justify our approach with citations in the methods (L 199), and clarify specifically how our approach differs from the much-criticised 'regression of residuals' method that the statistics reviewer referred to (L 199-202).*
- (3) Removed all words and phrases that imply causality throughout the manuscript, and have added a paragraph to the discussion regarding correlation vs. causation to avoid over-interpretation of our results (L 461-477).*
- (4) Addressed the issue of censoring bias in the survival analysis (L 278-282).*

Reviewer(s)' Comments to Author:

Referee: 2

Comments to the Author(s).

I appreciate the author's responses to the reviewer comments, and think the revised manuscript is much improved.

Thank you very much for reviewing our paper for a second time. We appreciate the investment in time this takes, and we are happy you think that the revised manuscript is much improved.

I still think it would be worthwhile having a specialist statistical reviewer comment on the analysis methods. I didn't quite understand the justification for keeping the residual analyses (which do not propagate error, and can lead to erroneous conclusions when model fit is poor), rather than controlling for productivity by including number of papers as a covariate. The authors replied "Number of publications could not be entered into survival models as a covariate or an offset because it was significantly correlated with all metrics". But isn't the correlation the thing we're trying to control? Again, this is not my specialty so I might be missing something obvious.

Otherwise, my comments are minor and can be easily addressed:

Please see our responses to the statistical reviewer.

Abstract line 29-32: recommend revising or splitting this long sentence.

Done.

Abstract line 32-33: "women were less likely, and slower, to become a principal investigator" and line 40-41 "increased likelihood and reduced time to become a PI". One of my concerns about the original manuscript was the unvalidated measure of 'principal investigator' being three last author publications. The authors have chosen to stick with the term PI throughout the manuscript. I would still like the abstract to define how this was measured, because many readers won't look into the details. For example: "women were less likely, and slower, to become a principal investigator (approximated by having three last-author publications)".

Thank you for the suggestion, we have changed as suggested.

Introduction, lines 53-55. The first sentence of the introduction is a bit hard to read.

Recommend splitting into two after 'challenge', or adding a comma after 'insecurity'.

We have added a comma after 'insecurity'.

Introduction, line 177: "if collaboration behaviour impacts on careers"... Should be "impacts careers" or "has an impact on careers".

We have changed this to "has an impact on careers".

Introduction, line 199: "published with more collaborators (i.e. unique co-authors), multiple times"... stray comma

We have removed the stray comma.

Methods, lines 208-211: "...any focal author with at least three last-author publications was classified as being a PI, and ii) we approximated the time taken ('time to become a PI') as the difference in years between their first publication and their second last-author (multi-authored) publication (i.e. requiring at least three publications)". If a PI requires three last-author publications, shouldn't the time taken to reach PI be the time between their first publication and *third* last-author paper, not the second?

We adopted this methodology to make our work comparable to other studies (e.g. van Dijk et al. 2014 Current Biology). We have now rewritten this section to explain that the

measurement is to the 2nd last-author paper, but 3 last-author papers are used as a buffer to estimate this somewhat crude measure of PI-status (L 207-213).

Results, lines 328-332: “female focal authors had significantly fewer co-authors than expected (adjusted network size; Fig.1D), published with the same co-authors less frequently than expected (adjusted tie strength; Fig.1E), and had more connected co-authors than expected (adjusted clustering coefficient; Fig.1F) when compared to male co-authors.” The use of ‘than expected’ in this sentence reinforces the ‘male default’. I recommend moving “when compared to male co-authors” to the start of the sentence and deleting the three “than expected”s.

We have made the change as suggested.

Results, line 387: “publishing with the same co-authors more frequently significantly reduced career length”. Is this meant to say ‘increased’ rather than ‘reduced’? If not, the results are a bit confusing, because lines 399-401 say “Publishing with more unique co-authors, with the same co-authors more frequently, and with less connected co-authors in early career all predicted a longer career overall.”

Thank you for spotting this. Yes, this has now been corrected to ‘increased’.

Discussion, lines 418-419: “Compared to men, women were also more likely to be a PI, and to become one more quickly, if they published repeatedly with their co-authors.” Recommend re-writing this sentence. I think the most likely interpretation is “women who publish repeatedly with their co-authors become a PI more than men”, but instead I think it’s trying to say that the effect of repeated collaborations is higher for women than men.

We have changed the text to make this clearer. The sentence now reads “Furthermore, publishing repeatedly with co-authors corresponded to researchers taking less time to become a PI, and again this was more positive for women.”(L 424-426)

Discussion line 494: “comparable positive effects” ... comparably?

Removed ‘comparable’ from the sentence.

Discussion line 502: “Researchers can enhance their careers in academic science by collaborating widely and repeatedly, and this is likely to be of even greater benefit for women” ... Up until this concluding paragraph of the revised manuscript, the authors have been fairly careful not to overreach the causal implications of the data. I recommend toning this down.

We have toned down the concluding paragraph on the discussion, and made sure not to imply any causality.

Referee: 4

Review of “Collaboration enhances career progression in academic science, especially for female Researchers” by Jessica E.M. van der Wal

Submitted to Proceedings B

Reviewed 2-28-2021

Comments to the authors:

This data set will be of great interest to the readers of Proceeding B, and I hope it will be published with appropriate revisions. I thank the authors for a well-written and cleanly edited manuscript, which makes the job of a reviewer much more pleasant. I have been asked to provide a special statistical review of the manuscript, so I will generally restrict my comments to statistical issues. My approach has been to take the revised manuscripts

at face value, i.e., without first reading the previous reviews and your responses to them. However, after a careful reading of the revised manuscript, I did go back and read the previous reviews in order to avoid (when possible) making requests that conflict with those of the previous reviewers, a situation that can be very frustrating for an author (we have all been there). My first two comments below (and especially the first) are of greatest concern.

1. My major concern is with unfounded attempts to infer causation. Yours is not a designed experiment in which subjects were randomized across potentially confounding variables, so the opportunities for causal inferences are limited (no fault of yours; it would be impossible to design and carry out such an experiment with human subjects). You acknowledge this in your response to the previous reviewers and have apparently made some changes to the text that tone down the causal inferences, but the revised text (especially the Discussion) is still permeated with causal conclusions. You are justified in interpreting gender effects as causal because causation necessarily flows from gender to any of the response variables that have been analyzed, although the path may pass through many intermediate (and unmeasured) variables. However, you are not justified in assuming that causation flows from collaborative behavior to academic success; there may be unmeasured confounding variables (academic environment, home environment, personality traits, etc.) that directly affect both collaborative behavior and academic success, generating a correlation between them. Here are some specific comments and recommendations:

a) Section titles in the Methods and Discussion that begin with “Effects of” are misleading. Although the contributions of predictors to linear statistical models are often referred to as “effects”, the use of this word can be misleading when the data are not from a designed experiment. Classic ANOVA models, from which the notion of “effects” originate, were all motivated by designed experiments. And in the vernacular, “effect” and “affect” are synonyms for “cause”.

Thank you for pointing out this issue. We have changed the section titles accordingly and attempted to remove all wording that might suggest causality. In addition, we now explicitly acknowledge in the discussion that our study is, by necessity, correlational and that it is possible that collaboration behaviour and career progression are not causally linked.

b) When describing statistical results for which causal inference is not possible, replace words and phrases that imply causation (effect, affect, benefits from, has negative consequences, etc.) with words or phrases that simply indicate a statistical association (predicts, correlates, differs from, relates to, etc.). There are abundant examples in the Discussion (lines 413, 414, 418, 420, 423, 436, 438, 439, 468, 474, 479, 483, 485, 489, 494, 500, 502).

Throughout the manuscript we have replaced all words and phrases that imply causality, with phrases simply indicating statistical association.

c) Contrary to your assertion, the newly added analysis (“Assessing directionality of the effects of collaboration behavior”, lines 297ff) does not allow causal inference. The fact that the collaborative behavior quantified in this reduced data set preceded (in time) the measured outcomes is not sufficient to eliminate alternative pathways of causation. Again, there may be unmeasured confounding variables that affect both collaborative behavior (including early in a subject’s career) and the measures of

academic success that you quantified. If you remove all language suggesting that this analysis provides insight into causation, the reader would be perplexed as to why it was done, so it should probably be eliminated from the manuscript.

We have removed all language suggesting that this analysis provides insight into causation. However, similar longitudinal analyses are common in non-human animal studies of sociality and e.g. fitness and longevity, so we would prefer to retain this part of our study for comparison. Furthermore, the result that collaboration behaviour during the first 10 years of publishing is already indicative of overall career progression is a useful insight as it is similar to the findings of van Dijk et al. 2014 Current Biology who showed that early publication history could predict PI status.

d) The significant interaction between gender and collaboration when modelling career length is interesting, and you give it considerable attention in your discussion. Assuming this pattern is real (not a type-1 error), you promote the interpretation that collaboration has a causal effect on career length, with gender influencing this relationship through mechanisms of the sort discussed in lines 442-464. The alternative causal explanation, however, is that there are differences between genders in confounding variables (academic environment, home environment, personality traits) that affect both the propensity/opportunity to collaborate and career length. I doubt that gender differences in personality traits provide a viable explanation since the Evolutionary Psychology literature suggests that, if anything, women are *more* likely to be cooperative and collaborative. However, environmental differences between genders that are confounding cannot be easily dismissed.

Thank you for reminding us that this should be evaluated. We have now included a paragraph in the discussion (L 461-477) where we consider whether our results could be due to gender differences in self-esteem and/or different family obligations and conclude that, regardless, our results indicate interesting associations.

e) The final paragraph of the Discussion clearly promotes a causal path from collaborative behavior to academic success. You need to add appropriate provisos to your conclusions about the value of interventions that increase opportunities for collaboration. The data presented in this manuscript do not support the contention that forcing collaboration on researchers who are not naturally inclined to collaborate will improve their academic progress. I confess to agreeing with you that collaboration is likely to have a causal influence on academic success, but the data presented, in and of themselves, do not demonstrate this.

We agree that we may have worded this too strongly. While still arguing for the value of programmes that foster collaboration, especially for female researchers, and across career stages, our revised text now acknowledges that quantitative data is nonetheless needed to ascertain the true value of such programmes. See L 504-508 (last sentences of discussion).

2. I have concerns about your justification and use of adjusted collaboration metrics. Each adjusted metric was obtained by replacing the raw metric with standardized residuals from a regression of the metric on publication number and gender. These adjusted metrics were then used as response variables when gender was the predictor, and as predictor variables when academic progress metrics were the responses. I have two concerns, the second of which is the more serious:

a) Publication number is one of four measures of academic success that you quantified. All four are correlated with one another, all differ significantly between males and females, and all are correlated with your collaboration metrics, so there appears to be no distinction between them in any of these regards. Why did you attempt to correct for

publication number but not for any of the other measures of academic success? What is special about publication number?

b) I apologize for the length of this comment, but I want to carefully justify my criticism rather than simply stating it as fact. When analyzing a non-designed experiment, adjusting for the effects of one or more variables, in the manner that you did, can lead to nonsense results. This is best demonstrated by an example. First, note that every correlation has a causal explanation. There need not be a direct causal connection (hence the misleading admonition that “correlation does not imply causation”), but if there is no direct causal connection between correlated variables y and z then there must be an indirect one via at least one additional variable that causally affects both y and z . Now imagine two variables y and x_1 that are not causally connected in any way (directly or indirectly) and between which there is necessarily no correlation. Now let’s include y (as the response), x_1 (as a predictor), and a third variable x_2 (as a second predictor) in a non-designed experiment where uncontrolled variable z_1 simultaneously affects x_1 and x_2 (causing a correlation between them) and uncontrolled variable z_2 simultaneously affects x_2 and y (causing a correlation between them). If you regress y on x_1 while controlling x_2 (i.e., using adjusted SS), you will find a correlation between y and x_1 even though there is no causal explanation for such a correlation out in the real world. This is purely a mathematical artifact of taking a method for partitioning causation in designed experiments and applying it in an inappropriate situation; the causal explanation of this new correlation is your mathematical manipulation. How does this apply to your study? The way in which you generated each adjusted collaboration metric is identical to including publication number (and its interaction with gender) in a multiple regression model, with the collaboration metric as the response, and then using the adjusted SS for any other predictor in the model. The adjusted covariance between the collaboration metric and the second predictor is identical to the covariance between your adjusted collaboration metric and that predictor. In other words, the correlation between an adjusted collaboration metric and any other variable in your data set will contain a mathematical artifact, to the extent that the correlation between the unadjusted collaboration metric and publication number is due to unmeasured confounding variables (of which there are likely to be many). This will affect any analysis in which an adjusted metric is either a response variable or a predictor. Another way to see this is from the perspective of path analysis, in which a correlation matrix is used to assign weights to a specified network of causal connections between the variables. Path analysis only works if the causal connections have been correctly specified and all relevant variables are included in the network. When important causal variables are omitted (e.g., the unmeasured confounding variables in your case), the results are nonsense. The bottom line is this: your analyses involving adjusted collaboration metrics have the potential to be highly misleading, and I encourage you to delete them.

Thank you for these detailed and clearly explained comments. We would first clarify that we do not refer to publication number as one of our measures of academic success. Rather, we consider publication number as a confounding factor that is essential to control for when investigating the relationships between collaboration behaviour metrics and career progression. Publication number is intrinsically related to the three measures of collaboration metrics and because it led to multicollinearity in our models, they could not be used in the same model in their raw forms (i.e. by including publication number as a covariate). Instead, we calculated the residuals of a simple linear regression of the interaction between publication number and age on each collaboration metric, and used the residuals from this regression in our final survival

analyses. We interpret the result as the relationship between our collaboration metric of interest, and career progression, having removed the (additional) contribution made through its relationship with publication number. It is important to note that this differs from the criticized use of residuals of the focal predictor (Freckleton 2002 “On the misuse of residuals in ecology: regression of residuals vs. multiple regression”), in that our approach only involves the response variables. We have added an explanation of why this approach is valid to the main text (L 191-202).

We have followed the reviewer’s suggestion of rerunning the analyses excluding publication output altogether (unadjusted metrics). These analyses show very similar results (ESM Supplement 1) to those of our original analysis, although in some cases we find larger effect sizes: especially when looking at the relationship between network size (number of co-authors) and career length. Whether this strong relationship is driven by network size OR also through the effect of number of papers on network size, cannot be assessed in analyses with unadjusted metrics. Therefore, we think that these analyses do not make much sense on their own as they ignore an important factor that is essential to account for in our models. In summary, although we appreciate the pitfalls that the reviewer has carefully outlined here, we stand by our original residual regression approach as we think it is essential to correct for publication number. We provide citations to make clear that we use an appropriate methodology to do so, and specifically clarify that our approach differs from ‘regression of residuals,’ and also that analyses with the unadjusted metrics give very comparable results (L 199-202, and ESM Supplement 1).

3. Please avoid language that suggests you are accepting the null hypothesis for a statistical test. “No significant difference” is standard jargon for “the null hypothesis is not rejected” which, in turn, is jargon for “no conclusion about the null hypothesis.” However, saying there is “no difference” (lines 190, 402, 474; there may be other examples) amounts to claiming that the null hypothesis is true, which is never a justifiable conclusion since the associated error rate is unknown. In lines 475-481 you clearly accept the null hypothesis that there were no gender differences in how collaboration is related to career length, and you propose several causal mechanisms for why this might be so. Notably absent from this list of explanations for your failure to detect a gender effect is the possibility that your analysis simply lacked sufficient statistical power to detect it. Given that a significant gender effect was found when the full data set was analyzed (larger sample size), this is a very viable explanation.

Thank you for your comment. We now refer to ‘no significant difference’ rather than the less clear ‘no difference’. We have also added the possibility that reduced sample size may explain why we find no effect of gender when analysing our early career dataset (L 490-493).

4. You apparently use a type-1 error rate of 0.05 for all of your statistical tests. This needs to be stated explicitly in your statistical methods.

We added to the text that all statistical tests were set to $\alpha = 0.05$.

5. I know it is common to refer to non-significant patterns as “trends” when the p-value is between 0.05 and 0.10 (Lines 367, 373, 385), but this really amounts to using a larger type-1 error rate than you report. This relates to my previous comment in that it reflects a tendency to mentally slide from “fail to reject the null” to “accept the null”. When $p > 0.05$ the correct conclusion is to withhold judgement about the null (NOT accept it), in which care you are in no danger of making a fatal error when $p = 0.06$, say. Sorry, but this is a pet peeve of mine.

We have removed all mentions of trends in the text.

6. Line 197ff: “we regressed each metric against an interaction between focal authors’ gender and number of publications, ...” Taken literally, this is a non-hierarchical model that includes no main effects. Is this correct? It seems that you might also have included at least the main effect of publication number since in the continuation of the sentence quoted above you describe the residuals from this model as “adjusted for publication output”. If you are dead-set on retaining the adjusted metrics (which I advise against), then please clarify the details of this model. Also, I don’t see that you have presented a test for the interaction term, and there is no real justification in controlling for a model term that is not statistically significant.

Here we wanted to account for the relationship between publication number and the social network metric (i.e. β), but not the difference in metric between the genders (i.e. intercept). Therefore, we used a non-hierarchical model with an interaction term that allowed the slope to differ by gender (now stated on L 193). Including gender as a main effect would make it nonsensical to then include it as a variable in models that also included gender (one of our key variables of interest). Although unusual, this is appropriate in this case. The test for the interaction term is presented in ESM 5, as indicated in the text.

7. Line 235ff: if less than 1% of focal authors appear as coauthors in other networks, why not just eliminate them from the analysis to ensure that the data are independent?

We appreciate the suggestion, but to remove the 1% of relevant authors is not a trivial process and would require considerable data wrangling. Given that there is already a precedent, we follow the approach taken in Turner et al. 2018 Behav. Ecol. Sociobiol., in which they reported 7% structure overlap among ego networks, from which was concluded that the networks were almost entirely independent of one another, and parametric statistical analyses were used.

8. For each of your GLMs, please specify the link function (I assume canonical, but this should be explicit) and the model fitting method (I assume maximum likelihood, but again, this should be explicit).

We have added the relevant link functions (Gaussian distribution: identity link; Binomial distribution: logit link; Poisson distribution: log link) and model fitting method (maximum likelihood) to the text.

9. Line 248: why was tie strength log-transformed prior to analysis? Inferences on mean(log(TS)) cannot necessarily be extrapolated to mean(TS). With your large sample sizes, the error variance of mean(TS) will be very nearly Gaussian, so the analysis on non-transformed data will be very robust.

We log-transformed tie strength because it made the residuals more normally distributed. In ecological studies it is common practice to transform the response variable to obtain the curve with the most bell-shaped curve of residuals possible (Zuur et al., 2009; “Mixed Effects Models and Extensions in Ecology” with R. Springer, New York, NY, USA).

10. In your GLMs, why did you use different error distributions for adjusted and unadjusted metrics? If assuming Gaussian errors is reasonable for the adjusted metric then it is also reasonable for the unadjusted scores. The adjustments should not affect the shape of the distribution very much. Choosing different distributions will, however, change the link function, assuming you are using canonical links. With your sample sizes, the choice of error distribution should have little effect on the analyses since the mean responses will be nearly Gaussian in all cases. This is more of a comment than a recommendation; I just found it curious that you chose different error distributions.

As per standard practice in ecological studies (e.g. Bolker et al. 2009, Generalized linear mixed models: a practical guide for ecology and evolution, Trends in Ecology & Evolution), a Gaussian distribution was used when the data was continuous, Poisson distribution was used for count data, and Binomial distribution was used for proportional data.

11. Line 261: “the assumptions of normality and homoscedasticity for GLMs ... were assessed by visual inspection of residuals and normal probability plots ...”: I realize this was added at the request of reviewer #3, but it makes your methodology inconsistent. You assume other distributions for many of your GLMs but express no concern about their distributional assumptions. Again, with your large sample sizes, all of your mean responses are going to be Gaussian to a very good approximation, so the analyses will be very robust even if the individual error deviations differ markedly from the assumed distribution. Furthermore, visual inspection of residuals is not a very objective way to assess assumptions; why even bother?

In our field it is standard practice to use QQ plots and residual vs. predicted value plots to assess normality and the uniformity of model residuals e.g. Buja et al. 2009, Statistical Inference for exploratory data analysis and model diagnostics, Phil. Trans. R. Soc. A.; Crawley 2013, The R Book Hartig 2021; DHARMA: Residual Diagnostics for Hierarchical (Multi-Level / Mixed) Regression Models). We have reworded this part of our methods (L 274-277) to increase clarity and provide the R package name used to produce the diagnostic plots (DHARMA, Hartig 2021, DHARMA: Residual Diagnostics for Hierarchical (Multi-Level / Mixed) Regression Models).

12. Lines 266 and 715: “Estimates from AFT models with CIs overlapping 1.0 are non-significant.” It took me a while to figure out what this even means. Please be more explicit: Estimates of the deceleration factor are significantly different from 1.0, with a type-1 error rate of 0.05, when the 95% CI does not include the value 1.0.

We have changed the text as suggested.

13. Line 274ff: In the AFT models for time to PI status and career length, censoring focal individuals who had not reached PI status by the end of the study, or who were still publishing at the end of the study, may bias the analysis. For example, what if the sex ratio in the censored group differs from that in the non-censored group? This would be simple enough to test for.

Thank you for raising the issue of censoring bias in our survival analyses. In the case of our survival analysis investigating career length, we believe we have taken an entirely standard and valid approach. Indeed, whether the sex ratio in the censored group differs from that in the non-censored group (it does), is exactly the question that we are interested in investigating. That is, do women in our dataset leave science (i.e. stop publishing) more often, and more quickly, than do men. We now report how censoring affects females and males to be entirely transparent on this point.

In the survival analysis where we investigate time to PI, we agree that our original approach could lead to bias. This is because our analysis included all authors in our dataset. Thus, the censored group contained authors who never became a PI during the time period of our study but remained in science, as well as authors who never became a PI but also left science. Censoring that includes this latter group could introduce a bias because once an author leaves science they have no future possibility of becoming a PI, according to our definitions. To avoid this issue, we now exclude from this analysis all authors who left science during the course of our study (i.e. stopped publishing before 2016) and never became a PI before they left science (n = 163 authors). This leaves only

authors who either achieved PI status (regardless of whether they remained in science after this achievement or subsequently left) or who remained actively publishing up to the end of our study and thus always maintained the possibility to become a PI. This analysis now becomes structurally similar to our survival analysis of career length in that, while there may be an unequal sex ratio in terms of who is and is not censored, this is what we are interested in. We have updated the text (L 278-282) to make these changes clear.

14. Line 288ff: These analyses do not address my concerns above. What you have shown here is that changing the censoring criteria does not have a qualitative effect on the resulting AFT models. These analyses make small changes to the censored group, whereas I am concerned about the characteristics of the censored group as a whole.

Please see our response to the previous comment, which we believe addresses your concern. In our study the characteristics of the censored group could be determined by the censoring criteria, if female authors were, for example, more likely to have longer gaps between publications than were men, causing them to be more likely to be censored. We consider the fact that changing the censoring criteria makes only small changes to the censored group is therefore a good thing, since it shows that our censoring is not introducing an unfair bias against one sex over the other.

15. Line 323: Remove the word “consequently”. You imply that the gender differences in your metrics of academic success can be attributed to the differences in publication number, but then you immediately follow by stating that these differences in academic success persist when you adjust for gender differences in publication number.

We have removed the word “consequently”.

16. Line 716: non-overlapping CIs is a conservative test for a nominal type-1 error rate of 0.05. The actual type-1 error rate for this test depends on the two standard errors and their associated degrees of freedom, but it is always less than 0.05.

*Thank you for your comment. We follow the procedure described in Swindell, W. (2010): Accelerated failure time models provide a useful statistical framework for aging research. *Exp. Gerontol.* 44(3):190-200. We have also changed the wording for this legend.*

Finally, I offer the following comments/suggestions of a non-statistical nature:

17. Line 194: there should be a paragraph break here. The first part of this paragraph addresses gender differences in prestige authorship, whereas the second part addresses the effect of publication number and the approach taken to correct for it. The “however” does not make sense here because the results are from a different analysis.

We have introduced a paragraph break, and removed ‘however’ from the sentence.

18. Line 205ff: Here are a couple of caveats to consider:

a) Researchers who collaborate infrequently, or only on dual-authored papers, will never achieve PI status as defined here. This will generate a positive correlation between collaboration and “career progression” that is purely an artifact.

b) When collaborators are mostly undergraduate students, as is often the case at non-R1 institutions, the PI will frequently be the first author rather than the last author. Such researchers are less likely to meet your PI criteria, even though they may have a consistent record of research funding and oversee a large lab of student researchers.

Thank you for these comments. We realise and admit that our definition of PI is not perfect. However, after much discussion and searching for alternatives, we believe it is the best measure available. Furthermore, last authorship usually denotes senior

leadership of an independent project in our field. We have added further details to clarify this in the text (L 210).

19. Despite the usage in the R community, the correct spelling is generalized (with a z), not generalised (with an s) linear model. See the original source: Nelder, John; Wedderburn, Robert (1972). "Generalized Linear Models". *Journal of the Royal Statistical Society. Series A (General)*. Blackwell Publishing. 135 (3): 370–384. This is a trivial point and I make no demands!

We have changed the spelling to 'generalized'.

20. Table 1: In the interest saving journal space, is it necessary to repeat so much of the methods in the table legend?

We have reduced the text in the legend of Table 1.

21. Table 1 and throughout the statistical results: You present a large number of estimated parameters for your models, along with their standard errors. Many of these numbers will be meaningless to your readers, especially those from the GLMs with non-identity link functions. The signs on the test statistics indicate the directionality of significant trends. You could save a lot of space, and make the text more readable, by getting rid of the estimated parameters that are not informative to the reader.

Thank you for your suggestion. However we have decided to retain all estimates for completeness and to aid any future use of our results in e.g. meta-analyses.

22. Line 385: there is something wrong with the sentence that begins “With a trend...”. Perhaps the colon after the parentheses should be a comma? If so, the sentence would read better if you reversed the two clauses.

We have reversed the two clauses, as suggested.

Appendix C

RESPONSE TO REVIEWERS

Associate Editor

Board Member: 1

Comments to Author:

Many thanks for resubmitting your manuscript, which has now been re-evaluated by the two previous reviewers. While both reviewers think the manuscript to be valuable, there are still some issues that need to be carefully addressed before potential publication. I believe a revision will be relatively straightforward because both reviewers provide clear guidance.

Dear Editor,

Thank you for your positive response to the revised manuscript, and for the thoughtful reviews that the two previous reviewers have provided. We respond to all of the reviewers' comments in detail below.

Reviewer(s)' Comments to Author:

Referee: 2

Comments to the Author(s)

I thank the authors for their responses to the reviewer comments, and to the editor for finding a statistical reviewer.

Thank you very much for reviewing our manuscript for a third time.

I still find the definition of PI and time taken to reach PI odd. I now understand that authors were classified as being a PI when they had three last-author publications, but the “time to reach PI status” was taken as the time to have two last-author publications. This is not clear in the abstract, which says (lines 31-34) “Among this cohort, women were slower and less likely to become a principal investigator (PI; approximated by having three last-author publications), and published fewer papers over fewer years (i.e. had shorter academic careers) than men.” Shouldn't it instead say “Among this cohort, women were less likely to become a principal investigator (PI; approximated by having three last-author publications), and were slower to do so (approximated by the time taken to have two last-author publications)”? Or, the analyses could be re-done so that time taken to reach PI is congruent with the stated definition of a PI.

*In previous versions we followed the methods of a published paper (van Dijk et al. 2014 Publication metrics and success on the academic job market. Curr. Biol. 24) but we agree that it was odd that the time taken to reach PI was calculated using a different number of publications than PI status. Therefore, we have redone the analyses so that the same number of papers is used to calculate both time taken to reach PI and PI status (i.e. the difference in years between a focal author's first publication and their **third** last-author (multi-authored) publication). This did not qualitatively change results.*

On the same topic, lines 313-317 read “Following the methods of[48], we approximated the ‘time to become a PI’ as the difference in years between a focal author's first publication and their second last-author (multi-authored) publication. Last authorship position usually denotes senior leadership of an independent project in our field[49] and this approach is robust to variations such as using time to publish two or four last-author publications (e.g. SI in[41]).” Shouldn't this say “robust to variations such as using time to publish three or four last-author publications”, given that time to publish two is presented as the main analysis?

See previous comment: this has now been changed to time to third last-author publication.

Another semantic quibble: “The likelihood that a female focal author left science after becoming a PI was higher than for men”. This could be slightly changed to “left academic science”, in recognition that some of those female authors might have left academic publishing but still be conducting scientific research elsewhere (e.g. industry, government).

This is a very valid point – thank you for highlighting this. Changed as suggested.

Also, please make all code available on the OSF (including for the supporting information – currently it says “The computational scripts used for these Supplements are available upon request.”).

The data and R scripts (including for ESM) are available via OSF:

https://osf.io/b7nfp/?view_only=17c6d4218ab94093b27f8119e2338e08

Finally, I recommend the authors do a careful proof-read before submitting any minor revisions. (I really sympathise with the tedium of doing so on a manuscript that has been so long in the making! I find the MS word “read aloud” feature helpful for this).

Apologies for this – the majority of these mistakes were only present in the track-changes document due to error, but not occur in the ‘clean’ version. We have now carefully proof-read our manuscript and have corrected any further mistakes.

For example, some small things introduced by tracked changes:

“Overall, authors with stronger and less clustered networks were more likely to become a PI, did so more quickly” – missing “and”

Now corrected.

“Persisting in academic science a major challenge” – missing “is”

Now corrected.

“By shaping funding, citation rates, and reputation, collaboration behaviour could have considerable impact on career progression in science” – not sure but I think this is missing “a”, or else “could considerably”

Have added “a”

“Collaboration patterns, however, differ between genders: female researchers generally have smaller[20] (but see[21]) and more clustered networks[22], publish fewer papers as ‘high prestige’ authors (single-author, or in first or last authorship positions)[23], and typically publish fewer papers than their male counterparts [10,20].” – superfluous “typically”, given there is already a “generally” in this sentence

Removed “typically” from sentence.

“Therefore, we conducted a longitudinal analysis to investigate whether a” – typo and missing space

Now corrected.

“Thereofor, we did not include authorship position in further analyses.” – typo

Now corrected.

“Unfortunately, there is little published work on how name changes affect indexing services and citation accuracy (but see[51]) so checked whether author name changes could have generated false negative records of leaving science.” – missing ‘we’

Now corrected.

Overly confusing sentence:

“In i), event = being a PI, and 64 of 772 focal authors (8%; 50% of whom were women) who were still in science at the end of the study period yet did not become a PI during this time, were censored. In ii), event = cessation of publishing, and 642 of 935 focal authors (69%; 27% of whom were women) who published within two years of the end of the study period were censored as still active.”

Changed to make less confusing:

“In i), the event was being a PI, and 64 of 772 focal authors (8%; 50% of whom were women) were censored because they were still in science at the end of the study period yet did not become a PI during this time. In ii), the event was cessation of publishing, and 642 of 935 focal authors (69%; 27% of whom were women) were censored because they were still publishing within two years of the end of the study period.”

Stray parenthesis:

Collaborative working has similarly positive effects on scientific success in computer science[22]), and for cell biologists and physicists[16],

Parenthesis removed.

Referee: 4

Comments to the Author(s)

I am attaching a file with my comments to the authors. The authors are to be commended for carefully revising their manuscript in response to the reviewers, and for their patience in dealing with several rounds of reviews. After the first set of reviews were submitted, I was invited to prepare a review as a statistical specialist, and I have now been asked to evaluate the revised manuscript. In the interest of moving forward I will restrict my comments to the requested changes that were emphasized by the editor.

Thank you once again for your positive and useful feedback.

1. *The previous version was replete with causal inferences despite the correlational nature of the study, and the authors have done a good job of purging most of these. However, some statements suggesting causation remain, and these must be purged before this manuscript can be published. They are few enough to be enumerated: Line 42 (abstract): replace “improve” with “are positively correlated with”. This is particularly critical since many readers of the journal will read only the abstract.*

Changed as suggested.

- *The authors cite a number of other studies that present some correlates of collaboration (lines 66-69, 483) and give causal interpretations to these correlations. I would not be surprised if the authors of these studies jumped to causal conclusions (this particular abuse of statistics is common), but that does not justify repeating them.*

We have adjusted the text in both these places to avoid causal language.

- Line 418: change “influence” to “correlate with”.

Changed as suggested.

- Line 423: change “increased” to “was positively correlated with”.

Changed as suggested.

- Line 425: change “corresponded to” to “was associated with” for clarity.

Changed as suggested.

- Line 427: change “had negative effects on” with “was negatively correlated with”

Changed as suggested.

- Line 428: change “support” to “are consistent with”.

Changed as suggested.

- Line 430: “women may benefit” suggests collaboration is causal. Break this sentence (lines 428-432) into two sentences. First sentence: “Therefore, our results are consistent with ... [13,60].” Start the next sentence with a caveat: “However, if this is the correct causal interpretation, then our results suggest ... [continue with rest of the sentence]”.

Changed as suggested.

- The discussion on lines 435-459 is relevant only if collaborative behavior is causal, so an appropriate proviso is called for. I recommend moving the last sentence in the previous paragraph (lines 432-433) to the beginning of the next paragraph (line 435) and rewriting it as follows: “If collaborative behaviors have causal effects on academic career trajectories, why might the effects be stronger for women than men?”

Changed as suggested.

- Lines 450-451: rewrite this sentence to replace “had especially positive effects” with phrasing that does not imply causation. Here you are talking about your data rather than ideas that are in the literature.

We have deleted this whole sentence, as it no longer applies.

- Line 481: change “may shape” to “is associated with”.

Changed as suggested.

2. The authors were encouraged to eliminate the adjusted collaboration metrics, but they have chosen to retain them. I have two comments about this, and one serious concern, as enumerated below.
 - (a) The authors appear to misunderstand the nature of my concern about the consequences of these adjustments. A regression approach is useful when building predictive models from descriptive data, and this is the focus of the literature they cite [46, 47] to justify their sequential regression approach. Their data are indeed descriptive, and a regression approach is appropriate, but the authors are quite clear that developing a predictive statistical model is not their primary motivation; they want to identify *potentially causal* variables. It is in this context that my concern is relevant. When the variables are adjusted, the experimenter becomes a causal agent and introduces correlations that will confound the correlations attributable to other causes. As the example from my previous review demonstrates, the adjustments can introduce correlations between variables that are uncorrelated in the unadjusted data. This is not a fatal problem if the authors refrain from offering causal interpretations. They are free to calculate any statistics they believe to be useful, such as the adjusted metrics, and to include them in their statistical models, but they must be very tentative when discussing these metrics as potentially causal.

Thank you for your comment. We accept that earlier manuscript drafts may have overemphasised relationships as being causal. We hope that this is no longer the case, and have tried to ensure that, following the reviewer’s careful comments, we do not make any statements that imply causation.

- (b) The authors have clarified their rationale for adjusting the collaboration metrics. Because of how the collaboration metrics are defined, they are intrinsically correlated with publication number. Thus, a correlation between one of these collaboration metrics and career advancement might simply reflect an influence of publication number on career advancement and have nothing to do with collaboration per se. As I understand it, the authors desire to calculate a set of statistics that can be interpreted as measures of collaboration that are not

correlated with publication number, and these are the adjusted metrics. This is fine, so long as the authors are careful to acknowledge that the results are consistent with multiple causal interpretations.

Thank you for your comment. We have tried to make clear in the discussion that our results are consistent with multiple causal interpretations.

(c) I am concerned that the method used for adjusting the collaboration metrics does not achieve the desired goal. When I read the previous draft of this manuscript, I assumed that the description of the statistical model used to generate the adjusted metrics was not complete (see comment #6 in my previous review). The authors have now clarified that this is not the case. As they describe it, each metric was regressed on the interaction between gender and publication number (PN), without any main effects, and the standardized residuals from this model were used as the adjusted metrics. These adjusted metrics will display no gender*PN interaction. But, unless the effect of PN is entirely tied up in the gender*PN interaction term, these adjusted metrics will not be independent of PN. If there was a PN main-effect in the raw data, then there will be a PN main-effect in the residuals. This is a likely explanation for why the authors obtained similar results with adjusted and non-adjusted metrics: both are associated with publication number, and publication number may be driving the patterns that the authors attribute to the collaboration metrics. Thus, the interpretations presented on lines 34-37, 197- 198, and 327-328 are highly suspect. This is a serious problem that must be addressed by the authors before this manuscript can be published. It is a simple matter to determine if this is a problem: if the metrics are properly adjusted then they should not be correlated with publication number. If the adjusted metrics are not independent of publication number, then the problem should be fixed by adding publication number as a main effect to the regression model used to adjust the metrics. My fear is that when all effects of publication number are removed, the correlations between collaboration metrics and academic advancement may largely disappear.

As suggested, we now have added publication number as a main effect to the regression models used to adjust the metrics. Although this does not change what is being estimated (different slopes for male and female focal authors), this approach allows us to report that the slopes are significant, and significantly different by gender, which we have added to ESM5.

3. Here are a few suggestions for improving the readability of the manuscript:
- I believe the modelling details given in lines 287-291 apply to the AFT models only, but this is ambiguous. I recommend breaking this into two paragraphs, one for the GLM's and one for the AFT models.

We have separated this part into two paragraphs per model type as recommended, and did the same in the previous section.

- Line 420: awkward wording

Changed to:

“Compared to men, women published almost half the number of publications, were almost 25% less likely to become a principal investigator (PI), did so 27% more slowly, and had 41% shorter careers.”

- Lines 506-509: I do not understand the point of this final sentence. Surely the data you present will be of interest to all readers, not just those who are inclined to collaborate. I recommend deleting this sentence.

We have removed this part of the sentence “for those researchers already inclined to collaborate”

- Line 624: “regression” is misspelled.

Changed.

- The Results section is difficult to read because of the large number of statistics embedded in the text. I recommend putting these into several tables and presenting only p-values (or the significance status in the case of AFT models) in the text.

We have revised the two existing tables and they now include all the statistics. We believe this now makes the results section easier to read. Thank you for this suggestion.